# Mesoscopic landscape of cortical functions revealed by through-skull wide-field optical imaging in marmoset monkeys

Xindong Song [1✉], Yueqi Guo[1], Hongbo Li[1], Chenggang Chen[1], Jong Hoon Lee[1], Yang Zhang [1], Zachary Schmidt[1] & Xiaoqin Wang [1✉]

The primate cerebral cortex is organized into specialized areas representing different modalities and functions along a continuous surface. The functional maps across the cortex, however, are often investigated a single modality at a time (e.g., audition or vision). To advance our understanding of the complex landscape of primate cortical functions, here we develop a polarization-gated wide-field optical imaging method for measuring cortical functions through the un-thinned intact skull in awake marmoset monkeys (*Callithrix jacchus*), a primate species featuring a smooth cortex. Using this method, adjacent auditory, visual, and somatosensory cortices are noninvasively parcellated in individual subjects with detailed tonotopy, retinotopy, and somatotopy. An additional pure-tone-responsive tonotopic gradient is discovered in auditory cortex and a face-patch sensitive to motion in the lower-center visual field is localized near an auditory region representing frequencies of conspecific vocalizations. This through-skull landscape-mapping approach provides new opportunities for understanding how the primate cortex is organized and coordinated to enable real-world behaviors.

[1] Laboratory of Auditory Neurophysiology, Department of Biomedical Engineering, Johns Hopkins University School of Medicine, Baltimore, MD 21205, USA. ✉email: songxindong@jhmi.edu; xiaoqin.wang@jhu.edu

Evolution has endowed primates with unique features in the organizations of cerebral cortical areas that contribute to our remarkable mental abilities[1]. Information from various sensory modalities is routed through these different areas that are located along a continuum of the cortical surface. The functional organizations of the cortical areas, however, are usually studied one area or one modality at a time. Our ability to efficiently map the primate cortex on its continuous surface across cortical areas and modalities is still limited. Enabling such an ability would thus lay a foundation for further systematic investigations into how different parts of the primate cortical system are organized together to generate our real-world perceptions and behaviors. Here we demonstrate a through-skull wide-field optical imaging strategy to enable mesoscopic observations across somatosensory, auditory, and visual cortices in individual awake marmoset monkeys.

Wide-field optical imaging across multiple modalities in the cortex, although has recently been applied through-skull in awake mice[2,3], has so far been impractical in primates. The imaging view and thus the modality coverage are constrained by the practical size of skull and dura openings, which were considered necessary in primates[4,5]. Across primate species, both skull thickness[6] and the extent of cortical folding[7] vary substantially (Fig. 1a). Interestingly, the common marmoset (*Callithrix jacchus*), an increasingly popular biomedical and neuroscientific primate model[8], features a skull thickness of ~0.5 mm[6] and a cortical folding index of ~1.2[7]. Both numbers are among the lowest of commonly used non-human primate (NHP) models, indicating the potential for imaging through the intact skull and resolving continuous topographic organizations in the cortex. Since the marmoset cortex is mostly smooth without folding (except for the lateral sulcus), many cortical areas are directly accessible under the skull. Noticeably, most of the auditory cortex[9], the middle temporal visual field (MT/V5)[10], and the lateral part of the primary somatosensory area 3b[11] are all located near but outside of the lateral sulcus (Fig. 1b, upper) and together provide an interesting target to image functional organizations across modalities. Therefore, we designed a chronic head-cap implant with a lateral recording chamber targeting these areas over the un-thinned intact skull (Fig. 1b, lower).

Under the skull, changes in cortical activity produce hemodynamic responses that alter tissue's optical properties, including absorption[12,13]. If both skull and dura are surgically removed and the brain is illuminated, this change can be picked up by the intensity of backscattered light and imaged by a camera (referred to as intrinsic signal optical imaging) in a variety of species including NHPs[4,5,12–15] (in some cases, through locally thinned skull[16]). The same imaging approach applied through the intact skull without thinning, although has been feasible in mice with a substantially enlarged field of views[17,18], is technically challenging in other species such as marmosets. Due to the thickness of the marmoset skull, the prominent optical scattering in the bone causes many photons to be backscattered before even reaching the cortex and thus reduces the imaging sensitivity to map cortical functions. Furthermore, as the skull curvature varies across the recording chamber, it is difficult to illuminate a large area homogenously without saturating pixels from surface reflection (Fig. 1c).

To overcome these limitations on utilizing through-skull intrinsic signal for mapping a large proportion of marmoset cortex, we first designed a polarization-gated imaging strategy to emphasize signals from deeper structures. We then applied it to awake marmosets and functionally parcellated adjacent somatosensory, auditory, and visual cortices through the skull. Furthermore, detailed topographic gradients, such as tonotopy, retinotopy, and somatotopy, were revealed within each modality

in the same subjects, and a novel tonotopic gradient was identified. While the retinotopy was measured using moving dots with minimal features in form, a cortical patch selective to the visual form of faces was also mapped out. These mesoscopic maps allow us to efficiently compile a comprehensive landscape across modalities and functions to identify organizational features beyond traditional scopes (Fig. 1d, e). The through-skull mapping approach introduced here can thus open new possibilities for understanding primate cortical organizations.

## Results

**Optical strategy for through-skull large-field imaging in marmosets.** To emphasize the light that has visited the cortex through the marmoset skull, but not the light that has only visited the skull or the surface, we utilized a polarization gating strategy. Polarization gating has been applied in biomedical imaging for selectively probing different depths in tissue[19]. In linearly polarized light, the polarization state is preserved in surface reflections as well as in single scattering events. Generally, more than ten scattering events are required to depolarize light effectively in biological tissues[20,21]. The depolarized light would thus have visited relatively deep in the tissue. We designed a cross-linear polarization enhanced intrinsic imaging setup (termed XINTRINSIC, Fig. 1f). By illuminating the skull surface with linearly polarized light and recording re-emitted light from only the orthogonal (cross-linear) polarization, the surface reflection was effectively eliminated (see also White et al., 2011[17]), and the entire chamber was homogeneously illuminated (Fig. 1d). Moreover, since light has to undergo more than ten scattering events to be significantly depolarized and therefore received by the camera, this strategy rejects the photons that visit only shallower structures in the skull (Fig. 1g).

After testing six wavelengths covering a broad range (470–850 nm, Fig. 1g–i), we found that the shorter wavelengths (e.g., blue, and green) can better detect cortical activities through the marmoset skull in response to stimulations of all three tested sensory modalities (Supplementary Fig. 1, see also "Intrinsic signal polarity" in "Methods"). Moreover, in the auditory cortex, tonotopic maps were obtained consistently and robustly only with the green or blue wavelengths but not with the red, far-red, or amber wavelengths (Supplementary Fig. 2). Similar phenomena have been reported repeatedly by auditory studies across a variety of species[22–28]. Additionally, our green light (530 nm) is near an isosbestic point (Fig. 1i, lower), at which the change in cerebral blood volume can be measured independently from the change in blood oxygenation[13]. The resulting signal would thus be more in line with other NHP neuroimaging approaches that measure cerebral blood volume, such as fMRI enhanced with iron oxide contrast agents[29]. Therefore, we sought to map cortical functions in the following experiments through the intact marmoset skull primarily with green light (but see also Supplementary Figs. 1, 2, and 'Determining the experimental wavelength' in "Methods"). The lateral resolution (full width at half maximum [FWHM]) of XINTRINSIC under this condition was estimated as 0.63 mm (Fig. 1h).

**Through-skull parcellation of functional modalities in marmoset cortex.** Firstly, to functionally parcellate somatosensory, auditory, and visual cortices within our imaging chamber (Fig. 2a), we designed a Fourier-encoded experiment to map these three modalities simultaneously within the same session, in which each modality had its stimuli presentation locked to a distinct repetition period (Fig. 2b). The somatosensory stimuli were mild air-puffs delivered to the subject's mouth and lower face presented at the middle of each 22 s-long cycle (Fig. 2b, left). The

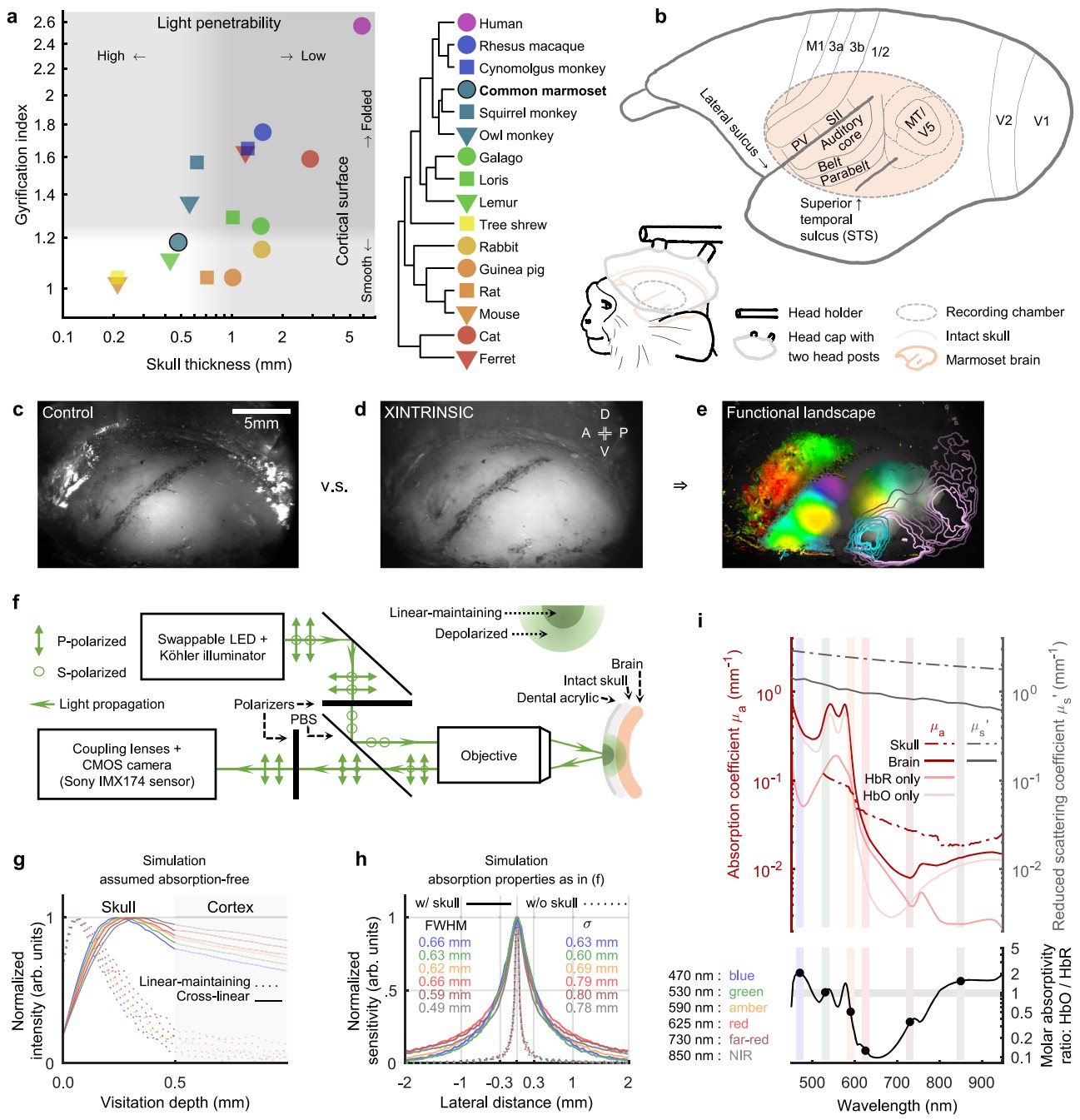

auditory stimuli were 80 dB SPL white noise bursts presented at the middle of each 19.8 s-long cycle (Fig. 2b, middle). The visual stimuli were radially outward moving dots gradually speeding up and then slowing down within each 18 s-long cycle (Fig. 2b, right). Within a 396 s-long session, somatosensory, auditory, and visual stimuli were repeated for 18, 20, and 22 cycles, respectively.

The spectrum of each pixel's response was calculated across the entire session with compensations for estimated delay and polarity (examples shown in Fig. 2c, d). If the signal of a pixel followed a particular type of stimuli well, it would show an evoked amplitude at the corresponding frequency in the spectrum with a phase near the middle of the stimulus cycle[30,31]. For every type of stimuli, we plotted each pixel's response by showing its phase at the corresponding frequency with color and the amplitude of that frequency with intensity (from traces as those shown in Fig. 2d to colormaps in Fig. 2e). Each modality of

sensory stimuli evoked responses in a distinct region within the imaging chamber. The orofacial air-puffs activated the cortical region anterodorsal to the putative lateral sulcus (pLS) (Fig. 2e, left), whereas the white noise sound activated the region posteroventral to it (Fig. 2e, middle). Additionally, moving dots activated the region posterior to the auditory region (Fig. 2e, right). These locations are consistent with previous results using invasive electrophysiology methods[11,32,33] and are also consistent with results that were acquired in each modality separately (Supplementary Fig. 3). Interestingly, there was a low-intensity gap between the auditory and visual regions that could not be effectively activated by any of the current stimuli (Fig. 2f). A large part of this location is consistent with regions suggested for audiovisual integration of motion in macaques and humans[34]. Although the general phenomena described above were robust across all tested subjects, individual variation was also evident

**Fig. 1 Optical strategies for through-skull imaging in marmoset monkeys. a** Skull thickness[6] and gyrification index[7] are compared across species. Light penetrability deteriorates as the skull thickness rises. Increased cortical folding (larger gyrification index) fragmentizes functional maps in the imaging field. **b** Upper: a sketch of key marmoset cortical areas[9–11]. Lower: an illustration of a marmoset with a head-cap implant and a recording chamber. The recording chamber imaged without (**c**) or with (**d**) cross-linear polarization gating under green light. The putative lateral sulcus was marked during implant surgery. A: anterior; P: posterior; D: dorsal; V: ventral. **e** An exemplar functional landscape obtained by XINTRINSIC (see also Fig. 7a). **f** A sketch of the XINTRINSIC optical design. **g** An estimation of photon visitation depths (based on a previous simulation[21]). Visitation depth is defined as the maximum depth that a photon visits in the tissue before being backscattered out of the tissue. A linearly-polarized beam was shed perpendicularly into the tissue. The photons that were depolarized and then collected from the cross-linear polarization channel visited deeper structures than the photons that maintained their initial polarization. These "cross-linear" photons, therefore, are presumably more sensitive to hemodynamic changes in the cortex. Note: this simulation assumes no absorption in the tissue (see also "Methods"). Thus, the traces over the "cortex" proportion are not intended to provide realistic numerical estimates but only for demonstration purposes. Lines are colored according to the LEDs analyzed. The same color scheme also applies in (**h**, **i**). **h** A lateral resolution estimation of through-skull signal sensitivity for picking up cortical hemodynamic changes in marmosets. σ: standard deviation. **i** Upper panel: absorption and scattering properties of the skull and the brain. Red curves: absorption coefficients ($\mu_a$); Gray curves: reduced scattering coefficients ($\mu_s'$); dashed curves: skull; solid curves: brain; pale-red curves: HbR (reduced or deoxygenated hemoglobin) and HbO (oxygenated hemoglobin) components in the brain; vertical color stripes: tested LED wavelengths. In both skull and brain, scattering is the dominant effect (coefficients are higher) over absorption. The absorption of the brain is much stronger at shorter than longer wavelengths. Lower panel: The molar absorptivity ratio between HbO and HbR across wavelengths. The gray horizontal stripe represents a unity molar absorptivity ratio. The green light (530 nm) used in the current study is close to an isosbestic point and thus measures changes in total hemoglobin concentration independently from changes in blood oxygenation[13] (see also Supplementary Figs. 1 and 2).

---

(Supplementary Fig. 4), which demonstrates the importance of parcellating these modalities on an individual subject basis. Data from this simultaneous parcellation experiment thus showed the ability of XINTRINSIC to characterize such a multi-modal functional landscape quickly in each individual subject over a large field of view (up to at least ~ 10 mm by 17 mm).

**Through-skull tonotopy mapping in marmoset auditory cortex.** After functionally parcellating these modalities, we sought to map topographic organizations within each individual modality in the same subject, starting with tonotopic maps in the auditory cortex generated by pure tones. The primate auditory cortex exhibits a hierarchical progression from the core region to the belt and parabelt regions[9,35]. The core region is suggested to have three primary-like areas: A1, R (rostral field), and RT (rostrotemporal field) in a caudal to rostral order (Fig. 3a), all of which are responsive to pure tones[32,35]. Both A1 and R are clearly tonotopically organized and share a low-frequency reversal in between[32,35] (demonstrated in Fig. 3b). Although the tonotopy in RT is less clear due to limited sampling in previous studies, a high-frequency reversal has generally been used to identify the transition from R to RT[15,32,35–37].

To map pure-tone-responsive tonotopic gradients, we utilized a phase-encoded Fourier approach[31,38]. A pure-tone pip sequence, increasing or decreasing in frequency by discrete semitone steps, was played at a moderate sound level (50 dB SPL) for 20 cycles in each recording session (Fig. 3c). The responses from these two sessions (Fig. 3d, e) were then combined to derive each pixel's tone tuning (Fig. 3f) and hemodynamic delay (Fig. 3g). The tone tuning was visualized with color (red: low-frequency, purple: high-frequency), while the tone responsiveness was visualized as the intensity in the map (bright: responsive, dark: non-responsive). Clear tone responsiveness and tonotopic gradients were found abutting the pLS and were largely within the region responsive to loud noise stimuli (Fig. 2e, middle). A zoomed-in view is shown in Fig. 3h (leftmost) together with those of all other tested subjects (see also Supplementary Figs. 5, 2). The known tonotopic gradients of A1-R-RT (Fig. 3b) were evident in every subject. In addition, we observed a low-to-high tonotopic gradient extending laterally and caudally from the low-frequency part of RT (white arrows in Fig. 3h) towards a high-frequency region outside the core region (black arrows in Fig. 3h). The observation of a pure-tone-responsive tonotopic gradient outside the classically described tonotopic maps suggests that the

standard view on the tonotopic organizations of the primate auditory cortex (Fig. 3b) needs to be revised (Fig. 3i).

**Comparison of tonotopic maps acquired through-skull or without the skull.** To validate the cortical responses imaged through-skull, in one of the recorded marmosets (M8E), we used the tonotopic map acquired through-skull (Fig. 4a–c) to guide the implantation of a chronic cranial window that targeted the entire marmoset auditory cortex on the brain surface (Fig. 4d). The cortex was imaged through a transparent artificial dura without the skull while the subject was awake. The same pure-tone tonotopy mapping stimuli (Fig. 3c) generated a tonotopic map (Fig. 4e) that is very similar to the through-skull map (Fig. 4b). The newly described tonotopic gradient is evident on both maps (white and black arrows in Fig. 4b, e). The temporal responses of two exemplar pixels for both through-skull and through-window conditions are shown in Fig. 4c, f. Although the through-skull response amplitudes were ~8 times lower than those with the skull removed, the temporal profiles and the signal-to-noise ratios (or the peak-to-SEM ratios) were comparable between the two conditions. At the population level, through-skull response amplitudes and tuning frequencies were strongly correlated with their through-window counterparts (Fig. 4g, h), suggesting the responses imaged through-skull indeed reflect changes in the cortical intrinsic signal. Moreover, to estimate how the skull diffuses the cortical intrinsic signal, we artificially diffused the through-window map at various scales (Fig. 4i) and estimated how well these diffused maps would explain the through-skull map via a linear model (Fig. 4j). The through-skull tunings and amplitudes can be best explained by through-window maps diffused at 0.60- and 0.70 mm scale (FWHM), respectively. These estimates are close to the simulated resolution of 0.63 mm (Fig. 1i). Together, they show that the resolution of our through-skull imaging is at the scale of ~0.6–0.7 mm.

**Through-skull retinotopy and motion sensitivity mapping in the MT/V5 complex.** Continuing with the subject shown in Figs. 2, 3f, next, we sought to map retinotopic gradients of the moving-dots-sensitive visual region determined in the parcellation experiment. This region presumably includes MT/V5[39], a motion-sensitive visual area[33,40,41] believed to be shared by all primates[1]. In fact, most neurons within MT/V5 are clearly selective to motion direction, largely regardless of other visual features like form[10,33,40]. A complex of satellite areas surrounding MT/V5 (Fig. 5a) is generally

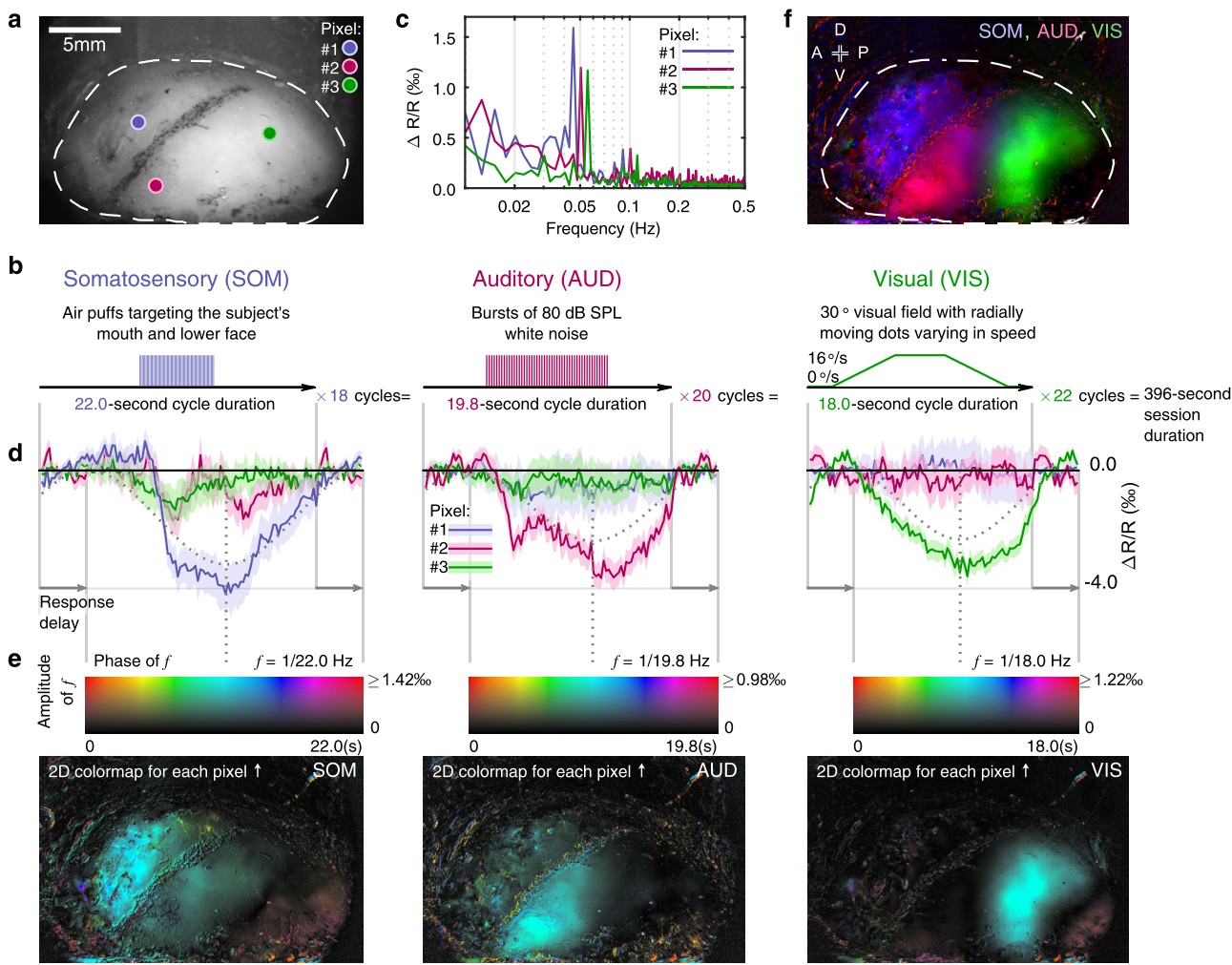

**Fig. 2 Through-skull parcellation of functional modalities in marmoset cortex. a** A view of the recording chamber (subject M126D) with the locations of three exemplar pixels labeled. Dashed contour: the extent of the imageable area over the skull surface. **b** Stimulus design for functionally parcellating somatosensory (SOM), auditory (AUD), and visual (VIS) cortices. **c** The spectra of the exemplar pixels calculated across the entire recording session, following the same color scheme in (**a**). **d** Intrinsic signal responses from the three exemplar pixels labeled in (**a**) to the stimuli shown in (**b**), following the same color scheme in (**a**). Each average response trace is shown as a solid line with a shade representing the standard error mean (SEM, $n = 18$ cycles for SOM; $n = 20$ cycles for AUD; and $n = 22$ cycles for VIS, in the relative change of re-emitted light, or $\Delta R/R$). For each modality, a cosine fitting curve to the trace of the most responsive exemplar pixel is drawn (dotted cosine line) with the response phase indicated by a dotted vertical line. **e** Response maps of the three modalities (left: SOM, middle: AUD, right: VIS). For each pixel, the cosine fitting phase (delay-compensated) and amplitude are encoded respectively as color and intensity in the map, according to the 2D colormap shown above each modality map (for response maps from all tested subjects, see Supplementary Fig. 4). **f** A summary map based on the modality parcellation results in (**e**), with each modality represented in a color channel (red: auditory, green: visual, blue: somatosensory). A: anterior; P: posterior; D: dorsal; V: ventral. Dashed contour: the same as in (**a**).

considered to be involved in motion processing as well[10,33,41,42]. But to what degree the moving-dots sensitive region covers this complex and its surrounding areas, is less established. Nonetheless, some features regarding retinotopic organization are generally consistent across studies in several species including marmosets: (i) MT/V5 is retinotopically organized[33,39,41–44] to represent the entire contralateral visual field in a non-mirrored image fashion[45] and shares a common foveal representation at its ventral side with its neighboring areas; (ii) the region anterior to MT/V5 has a crude retinotopic organization but as a mirrored image with a reversed polar angle gradient[33,43,44]. The previously reported marmoset retinotopic map[33] is illustrated in Fig. 5b. Yet, the spatial relationship between these retinotopic features and moving-dots sensitivity in the marmoset MT/V5 complex is not clear.

Here, we first confirmed the spatial extent of the motion sensitivity to moving dots was robust and replicable, by repeating the visual stimuli used in the parcellation experiment without the stimuli of any

other modality (Fig. 5c, d vs Fig. 2b, d, e). Next, to map retinotopic eccentricity, we varied the range of moving dot stimuli between the center and periphery of the visual field (Fig. 5e). As a result, a periphery-preferring region was located at the anterodorsal part of the motion-sensitive region, while a center-preferring region occupied the posteroventral side of the motion-sensitive region (Fig. 5f). Moreover, to map retinotopic polar-angle tunings (Fig. 5g±j), the dots were permitted to move radially only in a quarter-circle-shaped range in the visual field that was swept either clockwise (Fig. 5g) or counterclockwise (Fig. 5i) for 20 rounds. Similar to the tonotopy experiment, polar angle tunings and hemodynamic delays were calculated by combining these "reversed" sessions together (Fig. 5h, j, k). The resulting lower-upper-lower polar angle gradients for representing the contralateral field were virtually orthogonal to the eccentricity gradient (Fig. 5k vs Fig. 5f). Putting these together, the extent of moving-dots sensitivity occupied two retinotopic regions, and each had a representation of the entire contralateral

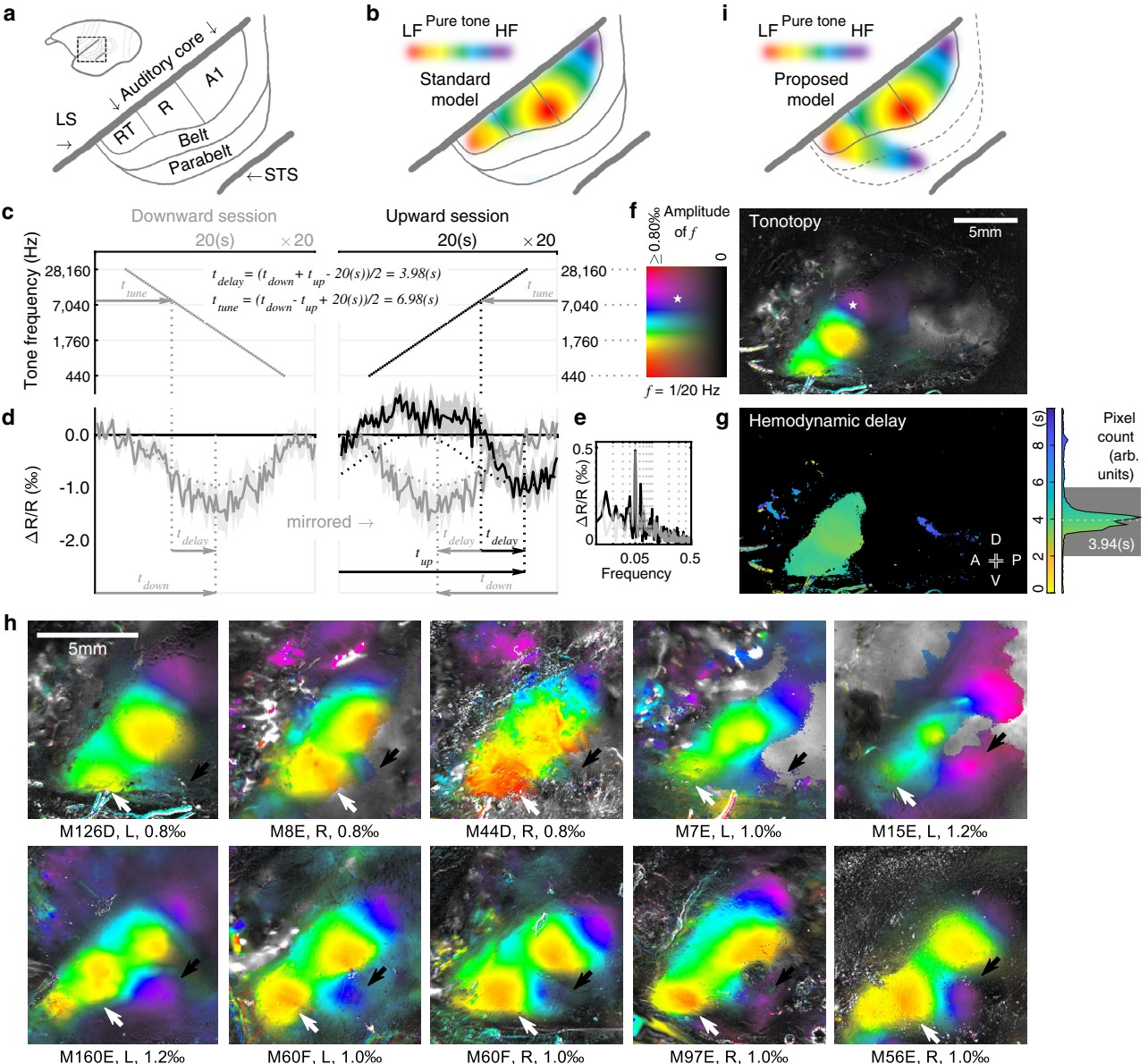

**Fig. 3 Through-skull tonotopy mapping in marmoset auditory cortex. a** A sketch of marmoset auditory cortex[9]. **b** The standard model of pure-tone-responsive tonotopic gradients in marmoset auditory cortex[32,35], LF: low frequency, HF: high frequency. **c** Stimulus design for mapping tonotopy using a downward (left) or an upward (right) pure tone pip sequence in discrete semitone steps. Each sequence was repeated for 20 cycles in a session. **d** Intrinsic signals responding to the stimuli shown in (**c**) from an exemplar pixel, whose location is labeled by a pentagram in (**f**). Average traces (solid lines) and SEMs (shades, $n = 20$ cycles) are shown in relative signal change ($\Delta R/R$). Dotted lines: response phase extraction (same as in Fig. 2d). Arrows and equations: tone tuning and hemodynamic delay estimation process. **e** The spectra of the exemplar pixel calculated across the upward (black) and the downward (gray) sessions. **f** Tonotopy mapped through-skull in subject M126D (the same subject as in Figs. 2, 5, 6, 7). The colormap on the left shows 2D codes for visualizing each pixel's tone-tuning frequency (color) and response amplitude (intensity). **g** Hemodynamic delay map. The hemodynamic delay values are color-coded (for the top 10% of most responsive pixels) and counted in the histogram on the right. Rectangular gray shade: the acceptable delay range. The average delay within this range is 3.94 s. A: anterior; P: posterior; D: dorsal; V: ventral. **h** Tonotopy from ten tested hemispheres in nine subjects (see also Supplementary Fig. 5). White (black) arrows: low- (high-) frequency end of the newly discovered tonotopic gradient. The subject ID, the imaged hemisphere side (L or R), and the upper display limit of response amplitude are listed below each plot. Right hemispheres are mirrored to the left for display purposes. **i** Proposed model for pure-tone-responsive tonotopic gradients in marmoset auditory cortex based on results in (**h**). A newly discovered pure-tone-responsive tonotopic gradient extends from the rostrotemporal field (RT) of the auditory core to the putative parabelt region. The putative area boundaries are indicated in dashed lines.

visual field (Fig. 5l). The more posterodorsal, non-mirrored retinotopic region was consistent with MT/V5 and the anterodorsal side of this region possibly also contained part of MST (Fig. 5m). The other more anteroventral region was retinotopically organized in a mirrored image. This retinotopic region was consistent with the anterior half of MTc/V4t and probably the dorsal half of FST(d) as

well[33]. These observations were generally consistent in all subjects we tested (Supplementary Fig. 6).

It is worth noting that we did not train our subjects to fixate during these sessions. Nevertheless, calibrated eye-tracking on spontaneous fixation showed the subject did keep the gaze to the center, with the median gaze position of any session maximally

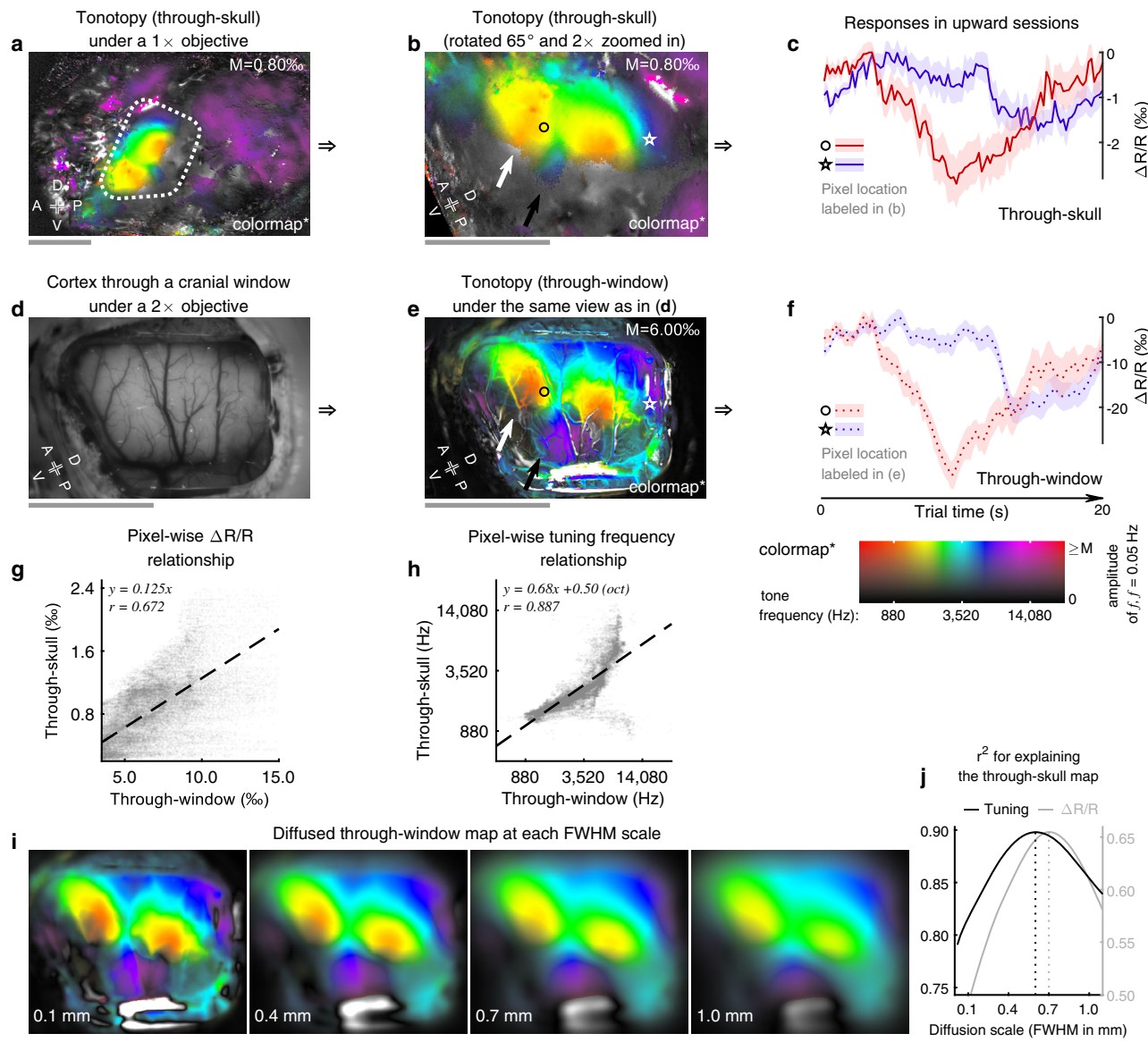

**Fig. 4 Comparison of tonotopic maps acquired through-skull or without the skull. a** The tonotopic map imaged through-skull in subject M8E under a 1×
objective. The dotted contour indicates the location of a subsequently implanted cranial window (**d**) to target the entire auditory cortex on the brain
surface. **b** The 2× zoomed-in map of (**a**). The white (black) arrow indicates the location of the low- (high-) frequency end of the newly described tonotopic
gradient (see also Fig. 3h). The locations of two exemplar pixels are labeled with a circle and a pentagram. **c, f** Trial responses from the exemplar pixels
labeled in (**b** and **e**) during the upward tonotopy mapping session imaged through-skull (**c**) or through-window (**f**). Solid-colored lines: mean responses;
shades: SEMs ($n = 20$ cycles). **d** A cranial window implanted over the location of the tonotopic gradients seen in (**a, b**). Imaging was performed under a 2×
objective. Both the lateral sulcus and the superior temporal sulcus are apparent and match their positions estimated through-skull. **e** The tonotopic map
imaged through the window as in (**d**). The arrows indicate the newly described tonotopic gradient as in (**b**), whereas the circle and the pentagram mark
approximated positions of the two exemplar pixels labeled in (**b**). **g** The scatter plot for comparing pixel-wise response amplitudes through-skull and
through-window. Linear model fitting: $F$-statistic $= 2.94 \times 10^4$, $p$-value $= 0$ ($n = 35679$ pixels). **h** The scatter plot for comparing pixel-wise turning
frequencies through-skull and through-window. Linear model fitting (in octave): $F$-statistic $= 1.32 \times 10^5$, $p$-value $= 0$ ($n = 35679$ pixels). **i** Artificially
diffused through-window maps at various scales. **j** Linear model fitting results across diffusion scales. The goodness-of-fit is given as the r-squared value[2]
between the diffused through-window map and the through-skull map, for both response amplitude and tuning frequency. The upper display limit of
response amplitude "M" is listed at the top right in each map (**a, b, e**). The 2D colormap for these maps is shown below (**f**). Scalebars in (**a, b, d, e**): 5 mm.
A: anterior; P: posterior; D: dorsal; V: ventral. Imaging was performed in the right hemisphere and the results were mirrored for display purposes.

1.1° off the center, while the motion stimuli subtended 30° of
visual angle (Fig. 5n). The oculomotor range of marmosets is also
more limited than macaques[46]. Thus, the effect of fixation was
likely marginal on the data described above. However, one needs
to be cautious when directly comparing retinotopic details of our
results with studies that directly controlled fixation. Moreover,
the retinotopic eccentricity was only mapped as center/periphery

preference under our current stimuli. A future improvement in
stimulus design can be attempted when absolute tuning values are
required, as by some retinotopy analysis[45].

**Face patch mapping and a functional landscape assembled
from all through-skull maps.** Besides encoding motion, the

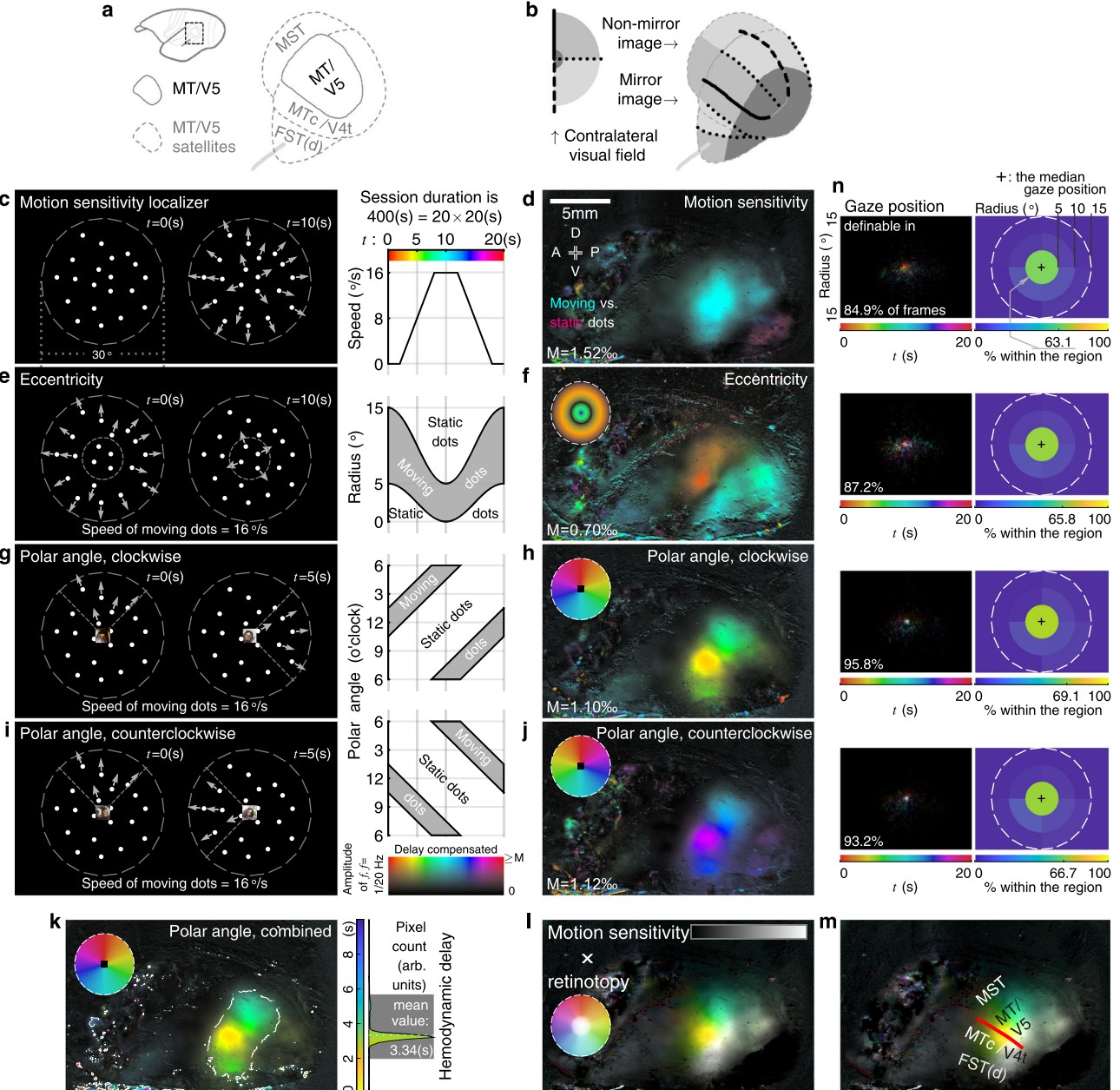

**Fig. 5 Through-skull retinotopy and motion-sensitivity mapping in the MT/V5 complex. a** A sketch of marmoset MT/V5 complex[10,33]. MST: medial superior temporal area, MTc/V4t: MT crescent/transitional V4, FST(d): fundal area of the STS (dorsal). **b** The retinotopic organization of marmoset MT/V5 complex, as in Rosa and Elston 1998[33]. **c, e, g, i** Stimuli for mapping motion sensitivity (**c**), retinotopic eccentricity (**e**), and polar angle tuning with clockwise (**g**) and counterclockwise (**i**) sequences. **d, f, h, j** Resulted maps in subject M126D (the same subject as in Figs. 2, 3f, 6, 7. See also Supplementary Fig. 6 for all tested subjects). A: anterior; P: posterior; D: dorsal; V: ventral. **k** Combined polar angle map with a hemodynamic delay histogram. The top 10% of most responsive pixels are outlined (dashed curves) and counted in the histogram. Rectangular gray shade: the acceptable delay range. **l** The summary map by combining the retinotopic and motion-sensitivity maps. These maps are encoded into separate channels in the HSV color space. The polar angle tuning (**k**) is encoded in the hue (color) channel, whereas the eccentricity (**f**) is encoded in the saturation (chroma) channel. The color wheel incorporates these retinotopic features: a pixel sensitive to the center (< 5° radius) would appear in white, whereas a pixel sensitive to the periphery would appear in a vivid color according to its polar angle tuning. The motion sensitivity (**d**) is encoded in the value (intensity) channel. The more motion sensitivity a pixel exhibits, the brighter it appears. **m** A putative assignment of area labels. The red line: the polar angle tuning reversal. **n** Eye-tracking results of these sessions. The calibrated and definable gaze position of each frame is color-coded for its cycle timestamp and overlaid in a 2D visual field histogram (left panels). A white spot in the histogram implies the position was not preferred by any specific time point in the cycle. For each session, the median gaze position was estimated 1.1°, 0.9°, 0.8°, and 0.9° away from the center. Response maps (**d, f, h, j, k**) are shown with tuning phase (color) and response amplitude (intensity) separately encoded, according to the 2D colormap by the left side of (**j**). The colormap is also transformed into wheel-shaped visual field color codes in (**f, h, j, k**). The upper display limit of response amplitude "M" is listed at the bottom left in (**d, f, h, j**).

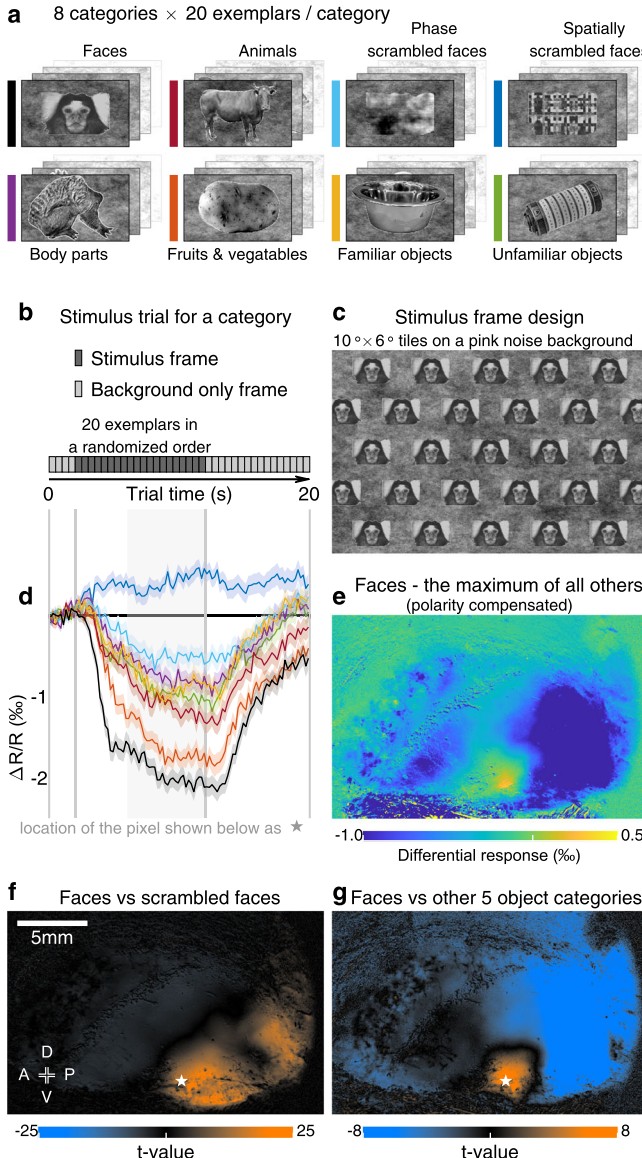

Fig. 6 Through-skull face patch mapping. a Eight categories of visual stimuli used for face patch mapping (with an exemplar from the category shown, see also Supplementary Fig. 7), including faces[52], five other object categories (body parts, animals, fruits & vegetables, familiar objects, and unfamiliar objects), and two types of scrambled faces (phase-scrambled and spatially scrambled faces). b The stimulus trial design for measuring response to a category. c The stimulus frame design for each exemplar (using a face as an example). The exemplar was tiled in the frame as a stretcher bond pattern. Considering the marmoset oculomotor range[46], the subject cannot fully avoid the exemplar within its center 5°-radius visual field. d Response to each category of a face-selective pixel. The location of the pixel is labeled by a pentagram in (f) or (g). The colors indicate categories, as shown in (a). Each average response trace is shown as a solid curve with a shade representing the corresponding SEM (n = 218 trials). The rectangular background gray shade indicates an averaging time window for calculating a response value for each trial. e Differential response map between the mean response to the face category and the maximum of mean responses to all other categories (n = 218 trials for each category). A positive value in the map indicates the pixel responded greater to faces than to any of the other seven stimulus categories. f The t-value map for comparing faces versus scrambled faces. A t-value is calculated for each pixel by comparing its single-trial responses between those to the faces (n = 218 trials) and those to the scrambled faces (n = 436 trials). Preference for unscrambled face in the map (orange color: high t-values) implicates involvement in visual form processing[92]. g The t-value map for comparing the face category (n = 218 trials) versus all other five object categories (n = 1090 trials). The face-over-object preference in the map (orange color: high t-values) defines the location of a face patch. The maps in the current figure are all from subject M126D (the same subject as in Figs. 2, 3f, 5, 7). See also Supplementary Fig. 9 for all tested subjects.

and Supplementary Fig. 7: faces, body parts, animals, fruits & vegetables, familiar objects, unfamiliar objects, spatially scrambled faces, and phase-scrambled faces). A clear patch located at the ventral part of the motion-sensitive region showed a stronger response to faces than to any other control categories (Fig. 6d–g).

Additionally, combining all maps provides evidence for a multi-modal functional landscape (Fig. 7a, see also Supplementary Figs. 8, 9). The region that prefers faces over scrambled faces (indicated by pink contours in Fig. 7a, see also Fig. 6f) mostly co-aligns with the center-preferring retinotopic region (including the white part of the retinotopic patch in Fig. 7a, see also Fig. 5f), with a saddle at the location consistent with the foveal region of MT/V5. This saddle is also consistent with the relative insensitivity to form in MT/V5[10,40]. Furthermore, the face patch (contrasting faces to other objects, indicated by cyan contours in Fig. 7a, see also Fig. 6g) was located within the motion-sensitive retinotopic region ventral to MT/V5, with a preference for the lower center of the visual field (see also Fig. 5f, k, l). These claims hold for all tested animals (Supplementary Figs. 9f, k, p, and 10). Moreover, close to this face patch, in subjects that were also tested for tonotopy, a part of the newly discovered tonotopic gradient (Fig. 3h) was evident to represent acoustic frequencies commonly found in marmoset vocalizations (~4–16 kHz)[53].

**Registration of the assembled functional landscape to the structure-based parcellations**. To relate the assembled functional landscape (Fig. 7a) to the current knowledge of marmoset neuroanatomy, the post-mortem brain of the subject (M126D) was further scanned by structural MRI. These structural data were registered to two neuroanatomy templates (Nencki–Monash template[54] and NIH-MBA v1.1 template[55]). The resulting areal parcellations and their comparison were shown in Fig. 7b–d.

primate visual cortex also processes form and features a specific distributed system for the face (face patches) that has been demonstrated in humans[47–49], in macaques[44,47,50,51], and recently in marmosets[52]. These face patches are located along the ventral occipitotemporal cortex and respond to faces much more strongly than to any other object category. Among these face patches, a functional hierarchy is suggested along the posterior-anterior direction with an increase in identity selectivity and view tolerance[50]. Functional specialization is also suggested along the ventral-dorsal direction with an increase in motion preference[48,51]. One of the face patches described in marmosets, the posterior-dorsal patch (PD), is suggested to locate closely with MT/V5[52]. This phenomenon has also been reported in macaques[44] and in humans[49]. Resolving the relative position of this face patch within the local functional landscape including MT/V5 would thus provide new insights into how the face-processing network is organized and has been transformed through evolution. However, the spatial organization of the marmoset PD patch with respect to (i) the local retinotopic map (if any), and (ii) the extent of motion sensitivity in the MT/V5 complex, remains unclear[52].

We sought to fill these gaps by further imaging and contrasting the responses to eight different categories of images (Fig. 6a–c

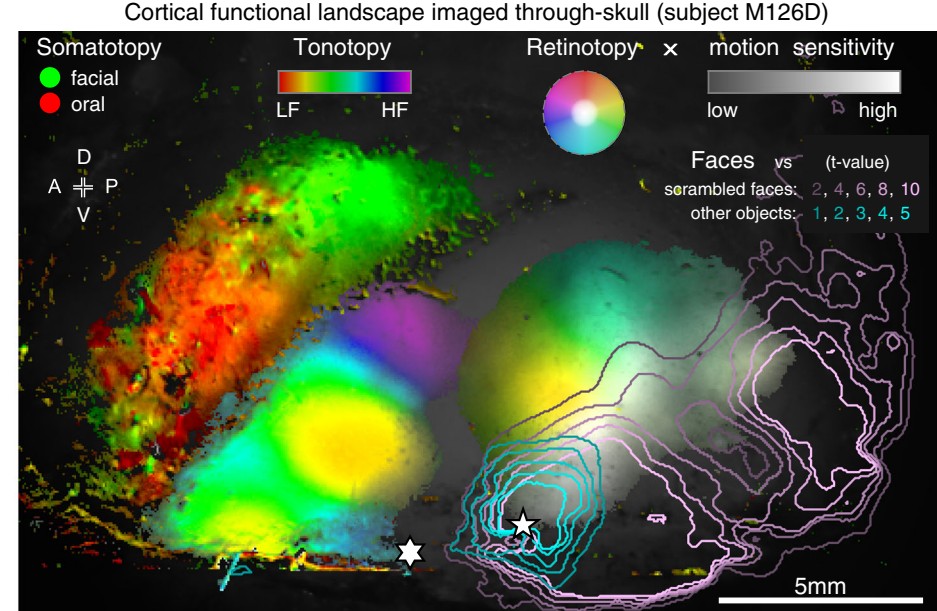

**a** Cortical functional landscape imaged through-skull (subject M126D)

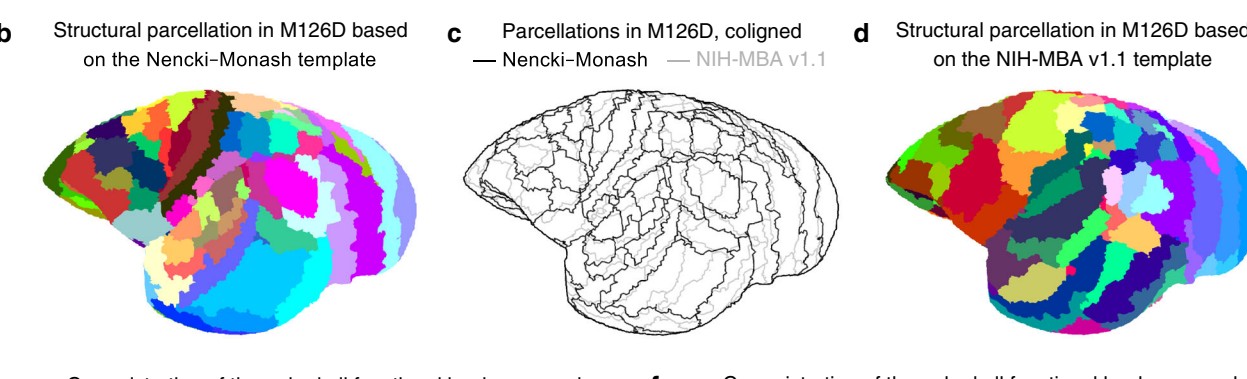

**b** Structural parcellation in M126D based on the Nencki–Monash template

**c** Parcellations in M126D, coligned
— Nencki–Monash  — NIH-MBA v1.1

**d** Structural parcellation in M126D based on the NIH-MBA v1.1 template

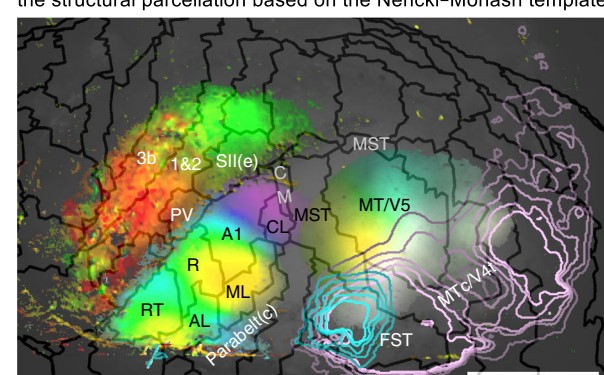

**e** Co-registration of through-skull functional landscape and the structural parcellation based on the Nencki–Monash template

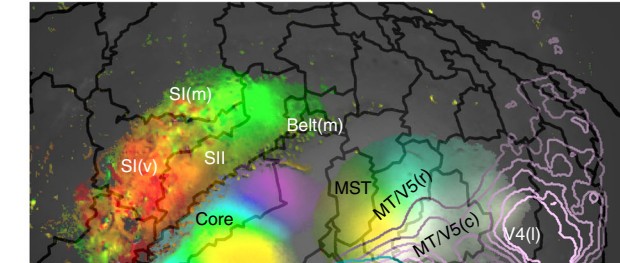

**f** Co-registration of through-skull functional landscape and the structural parcellation based on the NIH-MBA v1.1 template

These two structure-based parcellations are in general agreement but also differ from each other in many details (Fig. 7c). We further registered our functional landscape imaged through-skull to these structural parcellations (Fig. 7e, f). Many functional features in the landscape can be explained by the borders in these parcellations. For example, in Fig. 7e (Nencki–Monash template-based parcellation), the contour of auditory responsiveness in our functional landscape is recaptured by the overall border of the structurally parcellated auditory areas; the low-frequency reversal is coaligned with the border between A1 and R; the polar-angle reversal of the visual motion-sensitive area is also coaligned with the border between MT/V5 and MST. In Fig. 7f (NIH-MBA v1.1

template-based parcellation), the anterior boundary of the visual motion-sensitive region is consistent with the overall anterior border of the MT/V5 complex; and the region with functional insensitivity to face scrambling (the saddle of the pink iso-contours) overlaps with the structurally determined MT/V5 (caudal) area. In addition, we are particularly interested in the locations of two functionally defined regions—the PD face patch, and the newly described tonotopic gradient. The PD face patch sits within the visual area FST in Fig. 7e, and area FST (rostral) in Fig. 7f. These "architectural" registrations, together with the "functional" and "topographic" evidence that this face patch is also motion-sensitive and located within the similar retinotopic region that

**Fig. 7 The cortical functional landscape assembled from all through-skull maps. a** The cortical functional landscape summarized from modality-specific maps. The maps are overlaid on a dimmed image of the recording chamber. The motion-sensitive region is shown with retinotopic tunings together in the "HSV" color space (hue [color]: polar angle; saturation [chroma]: eccentricity; value [intensity]: motion sensitivity (see also Fig. 5l). The t-value maps are shown by the contour lines (see also Fig. 6f, g). Pentagram: a face-sensitive pixel (the same as in Fig. 6f, g). The face patch (outlined by the cyan contours) is largely contained within both the moving-dots-sensitive region and the retinotopic region representing the lower center of the visual field (see also Fig. 5d, f, k). Hexagram: a pixel from the newly discovered tonotopic gradient (see also Fig. 3h) extending from the rostrotemporal field (RT) of the auditory core to the putative parabelt region, tuned to 5.2 kHz, within the marmoset vocalization range of ~4–16 kHz[53]. An additional orofacial somatotopic gradient was measured and is described in Supplementary Fig. 8. All data are from subject M126D (the same subject as in Figs. 2, 3f, 5, 6). See also Supplementary Fig. 9 for all tested subjects. **b–d** Parcellations of cortical areas based on the structural MRI data acquired in the subject (M126D) and registered to the Nencki-Monash template[54] (**b**) or to the NIH-MBA v1.1 template[55] (**d**). A comparison between these parcellations is shown in (**c**). **e** Co-registration of the through-skull functional landscape shown in (**a**) and the structural parcellation based on the Nencki–Monash template[54] shown in (**b**). SII(e): secondary somatosensory area (external); PV: parietal ventral area; AL: anterior lateral auditory belt; ML: middle lateral auditory belt; CL: caudal lateral auditory belt; CM: caudomedial auditory belt; Parabelt(c): parabelt (caudal). **f** Co-registration of the through-skull functional landscape (**a**) and the structural parcellation based on the NIH-MBA v1.1 template[55] (**d**). SI(v): primary somatosensory area (ventral); SI(m): primary somatosensory area (medial); SII: secondary somatosensory area; Belt(l): Belt (lateral); Belt(m): Belt (medial); MT/V5(r): MT/V5 (rostral); MT/V5(c): MT/V5 (caudal); V4(l): V4 lateral; FST(r): FST (rostral); FST(c): FST (caudal).

prefers the lower-center of the visual field in all three tested subjects (Supplementary Figs. 9 and 10), support the idea that this face patch resides in the same area (FST) across subjects. Nevertheless, other possibilities, such as the PD patch may reside in cortical areas other than FST depending on individuals, cannot be excluded. A cortical area is typically distinguished from its neighbors by one or more neurobiological properties from four basic categories: "function", "architecture", "connectivity", and "topographic organization" (also referred to as the "FACT" categories[56]). Evidence from more categories, especially cross-validations by multiple techniques on an individual-subject basis, may help better disambiguate these possibilities in the future. Moreover, the newly described tonotopic gradient has its low-frequency end coaligned with the lateral border of RT in Fig. 7e. The similar area was hypothesized to have a role in interpreting the affective content of vocalizations[57]. This gradient also has its high-frequency end extending beyond the auditory belt regions into the auditory parabelt in both parcellations. Together, these structural parcellations (Fig. 7b–f) further support our conclusions in the previous experiments that were drawn based on functional mapping results.

## Discussion

The through-skull mapping approach described here allowed us to rapidly measure functional coordinates of a cortical landscape in individual marmosets and revealed organizational features across modalities and functions. The PD face patch described above is also sensitive to motion in the lower center part of the visual field and thus may mediate visual perception of mouth movement. Because the nearby auditory area represents frequencies commonly found in marmoset vocalizations[53], these adjacent cortical regions may form a network coordinating audiovisual information for vocal communication. Similar cortical regions and their neighboring areas in humans and macaques were suggested to be involved in perceiving social interactions[58,59]. Since the marmoset is a highly social and vocal primate[8], our approach has the potential to characterize cortical functional landscapes underlying social behaviors in both healthy and diseased individuals[60]. Moreover, our approach allows us to efficiently locate a functional region of interest in the context of its surrounding regions. This region-within-landscape paradigm may offer more replicable comparisons across individuals and studies than traditional approaches.

This functional mapping without thinning the skull was enabled by a wide-field, polarization-gated optical imaging approach. In contrast with traditional through-window approaches in primates that are limited by size and location constraints of the cranial window[4,5], our through-skull imaging approach allows more flexible observations of primate cortical organizations with much broader coverage. The effective size of a single field of view would be ultimately limited by the curvature of the skull. Nevertheless, one can image from multiple angles, stitch different fields of view together, and consequently increase the effective coverage on the surface beyond what a single field of view allows.

Although label-free, the intrinsic signal of the brain is inherently constrained by hemodynamic change, which may not be fully predicted by local neural signals[61]. Any significant vasculature between the brain and the imaging setup may also affect imaging quality, such as large blood vessels in the dura, diploic veins in the skull, or any tissue and vasculature proliferation around these structures. The optical properties of the skull, which are thickness- and composition-dependent and may have variations, would also affect the imaging resolution. Moreover, the condition of the coronal suture may vary across subjects and can affect the imaging quality over the local area (Supplementary Fig. 4). Compared to fMRI, another neuroimaging method that also utilizes the hemodynamic signals to measure brain functions, the through-skull optical imaging described here is at a much lower cost (<$10,000) and with no requirement for day-to-day maintenance. It does not generate any acoustic sounds and thus allows quieter and more controlled auditory experiments to be performed. It can also be used in combinations with other fMRI non-compatible electronics or materials (e.g., cochlear implants). Nevertheless, the through-skull imaging still requires the removal of the scalp and thus can be more invasive than fMRI. It does not resolve any depth information or cover the entire brain in 3D, both of which are well enabled by fMRI. fMRI is also inherently more intuitive to be integrated with other MRI-based structural measurements (e.g., diffusion tensor imaging).

Despite its limitations, the approach developed in our study could provide direct guidance for subsequent experiments that require skull opening (e.g., electrophysiology recordings[62], two-photon imaging[63], or functional perturbations[64]), given that the individual variation on functionally identified regions can be up to ~2 mm in marmosets, ~6% of their brain size (Supplementary Fig. 11). It may yet prove powerful in concert with non-invasive viral gene delivery in primates[65] for potential through-skull calcium imaging[2,3] as well as minimally invasive optogenetic perturbations[66] where targeting of causal manipulations could benefit from our relatively non-invasive imaging approach. We provide open access to our imaging setup design and software code for straightforward duplication (https://x-song-x.github.io/XINTRINSIC/). With its simple architecture, the XINTRINSIC

setup is largely affordable and has potential for future miniaturization to map cortical functions through-skull in freely roaming marmosets as well.

Together, the ability to map through the intact skull across cortical landscapes provides new opportunities for examining how the primate cortical system is organized and coordinated across different modalities to enable real-world perceptions and behaviors. This low-cost, readily replicable, and efficient imaging approach would thus have the promise to push NHP neuroimaging toward democratized, scalable, and reproducible science.

## Methods

**Subjects**. Eleven marmosets were used in the current study. The eleven include six for mapping cross-modality landscape through the intact skull (summarized in Supplementary Table 1) and another five as additional tonotopic mapping subjects (Fig. 3h and Supplementary Fig. 2); four females, seven males; and five left hemispheres, seven right hemispheres. The right hemisphere view was mirrored to match the view of the left hemisphere in the figures for display purposes. The six landscape-mapping subjects' ages ranging from 44 to 81 months old during imaging acquisition. All experimental procedures conformed to local and US National Institutes of Health guidelines and were approved by the Johns Hopkins University Animal Use and Care Committee.

**Through-skull imageability compared across species**. The gyrification index[7,67] (GI), and the skull thickness were compared across species based on the data compiled from multiple sources (Fig. 1a). Human (*Homo sapiens*), rhesus macaque (*Macaca mulatta*), common marmoset (*Callithrix jacchus*), loris (*Nycticebus coucang*), lemur (*Cheirogaleus medius*), rabbit (*Oryctolagus cuniculus*), cat (*Felis catus*), and ferret (*Mustela putorius furo*) were reported to have GI values of 2.56, 1.75, 1.18, 1.29, 1.11, 1.15, 1.58, and 1.63, respectively[7]. Cynomolgus monkey (*Macaca fascicularis*), squirrel monkey (*Saimiri sciureus*), owl monkey (*Aotus trivirgatus*), galago (*Otolemur garnetti*), tree shrew (*Tupaia glis*), guinea pig (*Cavia porcellus*), rat (*Rattus norvegicus*), and mouse (*Mus musculus*) were reported to have GI values of 1.65, 1.57, 1.36, 1.25, 1.04, 1.04, 1.04, and 1.02, respectively[67]. All skull thickness data were taken near the location of the parietal eminence. Human, rhesus macaque, cynomolgus monkey, common marmoset, squirrel monkey, owl monkey, and loris data were acquired from one primate study as 5.87, 1.52, 1.24, 0.48, 0.62, 0.56, and 1.01 mm, respectively[6], whereas rabbit[68], rat[68], mouse[69], cat[70], and ferret[71] data were 1.50, 0.71, 0.21, 2.90, and 1.20 mm, respectively. Galago (https://doi.org/10.17602/M2/M33809), lemur (ark:/87602/m4/M12869), tree shrew (https://doi.org/10.17602/M2/M6393), and guinea pig (https://doi.org/10.17602/M2/M47035) data were assessed based on the cranium CT scan datasets accessed and archived on MorphoSource.org. The resulting values are 1.48, 0.43, 0.21, and 1.01 mm. The common marmoset used in the present study has a relatively flat brain and thin skull with a GI value of 1.18 (considered lissencephalic[8]) and a skull thickness of 0.48 mm (considered as imageable through-skull based on the data presented in the current study). We further confirmed marmoset skull thickness near the location over the auditory cortex by clamping a micrometer onto the edge of acutely prepared bone chips (which may slightly overestimate the thickness due to the skull curvature). The average value is 0.61 mm ($n = 4$ samples), very close to the value reported on the parietal location.

**Marmoset preparation for through-skull imaging**. The basic design of the marmoset chronic head-cap implant has been described previously[72]. Marmosets were adapted to sit calmly in a Plexiglass restraint chair through an acclimation period of two to four weeks. Using sterile techniques, two stainless steel head posts were attached to the skull with dental cement material while the animal was anesthetized with isoflurane (0.5–2.0%, mixed with pure oxygen). During the implant surgery, the lateral part of the skull—presumably over the auditory cortex, the MT complex, and the lateral part of the anterior parietal cortex (Fig. 1b)—was exposed and covered with a thin layer (~1 mm) of dental cement. Before applying the dental cement, the pLS was vaguely visible through-skull when the skull was still wet and was thus marked for later reference (water may partially match the refractive index of the skull). The rest of the exposed skull was covered with a thicker layer of dental cement and a wall was formed around the recording chamber to increase the mechanical stability of the head cap and to protect the chamber.

After the animal fully recovered from the implant surgery, the head of the animal was fixed via head posts (Fig. 1b), and the surface of the recording chamber was further polished with acrylic polishing bits (Dedeco, 7750). The dental cement at the chamber bottom optically concealed the surface texture of the skull, by presumably matching its refractive index. We tested two different dental cement materials (orthodontic resin from DENTSPLY Caulk, and C&B Metabond from Parkell [clear]). We did not observe a significant difference between these two materials in terms of imaging performance. Before each imaging experiment, a minimal amount of petroleum jelly was applied evenly at the chamber bottom to further augment the optical smoothness of the surface. The optical axis of the

imaging setup was aligned as perpendicular as possible to the local surface. The animal was head-fixed and kept awake through all imaging sessions. Outside of imaging experiments, the chamber was protected and sealed with a dental impression material (GC America, EXAMIX NDS injection). When the chamber condition is well maintained, the through-skull imaging can be performed for up to at least several months. Several factors may affect the long-term performance of the through-skull imaging, such as uneven surfaces that compromise the optical homogeneity of the imaged area; tissue and vasculature proliferation in the skull, especially near the coronal suture; potential mechanical deterioration of the headcap over time; and possible age effects on the optical property of the skull.

**Optical properties of the skull and the brain**. To analyze how the imaging light propagates in the tissue (Fig. 1g–i), the optical properties of both the skull and the brain were obtained from previous studies. For scattering in the skull, we followed the equation: $\mu_s'(\lambda) = 1533.01 \cdot \lambda^{-0.65} (\text{mm}^{-1})$, where $\lambda$ is the wavelength in nm, and $\mu_s'(\lambda)$ is the reduced scattering coefficient[73]. For absorption in the skull, coefficients were averaged across three datasets to cover the spectrum[73–75]. For scattering in the brain, the scattering coefficient $\mu_s(\lambda)$ and the anisotropy factor $g$ (the mean cosine of the scattering angle) were obtained across a wide spectrum[76]. $\mu_s'(\lambda)$ were calculated as $\mu_s'(\lambda) = \mu_s(\lambda) \cdot (1 - g)$. For absorption in the brain, we assumed: (i) the major absorber in the brain is blood; (ii) blood composes 3% volume of the brain; (iii) the concentration of hemoglobin in blood is 2 mM (~130 g/L); and (iv) the composition of hemoglobin is 25% HbR and 75% HbO[77]. The absorption coefficients of the brain were thus derived based on the absorption spectra of HbR and HbO (https://omlc.org/spectra/hemoglobin/summary.html). These coefficients are plotted in Fig. 1i.

**Imaging setups**. The general optical design of XINTRINSIC is illustrated in Fig. 1f. A swappable LED (Thorlabs M470L3, M530L4, M590L4, M625L3, M730L5, and M850L3; corresponding to nominal wavelengths [colors] of 470 nm [blue], 530 nm [green], 590 nm [amber], 625 nm [red], 730 nm [far-red], and 850 nm [near inferred, or NIR], respectively) passed light through a Koehler illuminator consisting of 4 lenses (Thorlabs ACL12708U-A/B, Edmund Optics 49–793-INK, 33–923-INK, and 49–372-INK), a field stop, and an aperture stop. The light from the Koehler illuminator was filtered by a polarizer (Thorlabs WP25L-VIS) before being reflected by a polarizing beam splitter (PBS, Thorlabs WPBS254-VIS) to a 1× telecentric objective (Thorlabs TL1X-SAP, without the accessory wave plate). The polarizer was aligned to maximize S-polarized light throughput to the PBS. The light power under the objective was measured as not more than 50 mW before in vivo imaging. The focal plane was lowered to the level that put most of the surface features on the skull in focus. Backscattered light was collected by the same objective, transmitted through and filtered by the same PBS, and then passed to another polarizer placed to allow P-polarized (cross-linearly polarized) light to go to the coupling relay lenses (Thorlabs, AC508-080-AB-ML, ×2). The light was eventually focused by a camera lens (Navitar, NMV-6X16, at the maximal zoom) onto a high-saturation-capacity, high-frame-rate, mass-produced CMOS camera (FLIR, grasshopper 3, GS3-U3-23S6M-C, with a SONY IMX174 sensor). The imaging acquisition of the camera was controlled by an 80 Hz trigger pulse train, generated on a data acquisition (DAQ) card (NI, PCIe-6323). Recordings were acquired with custom code in MATLAB (Mathworks, R2018a and later), at a resolution of 1920 × 1200 pixels and a speed of 80 frames/second (1920 × 1200@80fps). The data were binned and saved in a format of 480 × 300@5fps. Based on a Poisson process assumption for incoherent photon arrivals, the theoretical shot noise floor under the current condition over 20 repetitions was estimated as 0.08‰.

Besides the Koehler illuminator built in the XINTRINSIC setup, another independent illuminator, coupled with fiber bundles as output ports but without polarization control, was built for control purposes (used in Fig. 1c). The light from one of the swappable LEDs was passed through four coupling lenses with a diffuser in the middle (Thorlabs ACL12708U-A/B, AC254-045-A-ML, ED1-S20, ACL5040U-A, and ACL50832U-A) to fulfill both the input aperture (4.52 mm) and the numerical aperture (NA = 0.57) of a gooseneck Y-bundle light guide (Thorlabs, OSL2YFB). The output ports were pointed from the side of the objective to illuminate the chamber as homogenously (to the best of our ability).

Our through-skull imaging targets a field of view (FOV) that is large (>20 mm) and curved over the skull surface. This requires a thick depth of field of the setup to keep a consistent lateral resolution across the entire FOV. By utilizing an objective (Thorlabs TL1X-SAP) with a small numerical aperture (NA = 0.03), our setup features a total depth of field estimated as ~0.98 mm (following equation [10] in Oldenbourg and Shriback 2009[78]). With this relatively thick depth of field, we were able to capture images with sharp small features throughout the curved skull surface in the FOV. These features can serve as reference points for aligning sessions acquired across different days. Moreover, our estimated resolution on functional measurements is ~0.6–0.7 mm (Fig. 4). To produce a lateral blurring more than half of this scale, a target needs to be placed at least 5 mm away from the focal plane ($h \geq D \cdot n/(2 \cdot \text{NA})$, where $h$ is the focal plane depth from the nominal plane, $D$ is the maximal blurring diameter, $n$ is the refractive index, and NA is the numerical aperture). This depth tolerance on functional measures further facilitates functional mapping over a large FOV.

**Determining the experimental wavelength.** To determine the wavelength effect for through-skull imaging of cortical intrinsic signals across multiple sensory modalities in marmosets, the same experiment described in Fig. 2 was repeated with six different wavelengths in subject M126D. Response patterns following somatosensory, auditory, and visual stimuli were functionally identified (Supplementary Fig. 1).

The shorter wavelengths (blue and green) had superior response amplitudes over other wavelengths (amber, red, far-red, and NIR) for all three modalities (Supplementary Fig. 1b–d). This is consistent with the absorption spectra—the blood in the brain absorbs the shorter wavelengths (blue and green) much more efficiently than other wavelengths (Fig. 1i, upper). Although a longer wavelength (e.g., NIR) would have weaker scattering and absorption and thus penetrate deeper brain structures, the signal amplitudes were much smaller.

The amber light shows ambiguity in the signal (the third column in Supplementary Fig. 1). For each direction (positive or negative) we assume a ΔR/R change would respond to a stimulus, the resulting response patterns are not consistent with those measured with other wavelengths. This ambiguity can be further explained by how much a wavelength emphasizes the HbO increase over the HbR decrease following a stimulus (Supplementary Fig. 1f, g, see also the "intrinsic signal polarity" section later).

The auditory response patterns measured with green, blue, and NIR lights are generally similar but are different from the patterns measured with red and far-red lights (Supplementary Fig. 1c, e). This is consistent with the previously reported difficulty of mapping the auditory cortex with red and its neighboring wavelengths[22–28]. Indeed, to date, the auditory cortex in primates or carnivores has only been successfully mapped with green lights[15,23–28], but not with any red light (wavelength: ~600–700 nm), a popular wavelength choice for visual and somatosensory studies.

To further test the wavelength effect for mapping the auditory cortex in marmosets, we measured tonotopy in three additional subjects with six wavelengths (Supplementary Fig. 2). In subject M160E (left column in Supplementary Fig. 2), imaging was performed through the un-thinned intact skull. Despite a gross agreement in the coarse tonotopy, the activation pattern varies across different wavelengths. Blue, green, and NIR lights all produced a well-contrasted activated area right below the pLS with a good consistency among themselves. In contrast, amber, red, and far-red lights generated patterns with more variations and these patterns are dominated by "vessel-like" stripes running away from the pLS and extending more posteroventrally than the activated area in the blue, green, and NIR maps. The scales of these stripes and the gaps between them are comparable to our estimated resolution of through-skull imaging with the green light (~0.6–0.7 mm). These stripes can be potentially due to blood vessel artifacts being emphasized by these wavelengths (e.g., vein draining), or alternatively, attributed to a true stripe-like organization in the cortex that is only differentiable when resolution is high enough. To tease these possibilities apart, we mapped tonotopy in another two subjects, M97E through a partially thinned skull (middle column in Supplementary Fig. 2), and M102D with a cranial window targeting the entire auditory cortex (right column in Supplementary Fig. 2). The observations described in the first subject are also robust in these two subjects. Additionally, the "vessel-like" stripes under amber, red, and far-red illuminations become more and more evident in these two subjects. These stripes are indeed dominated by artifacts on blood vessels leaving the auditory cortex towards the STS and are less contributed by hemodynamic responses in brain parenchyma to reflect local neuronal activities. It is thus most likely that the "auditory" patterns measured with amber, red, and far-red lights in the first subject (left panels in Supplementary Fig. 2d–f) are "artificially" extended and biased towards the STS by a similar blood vessel pattern. Together, for the reasons that are still not fully understood, we were able to acquire consistent and robust auditory maps with green and blue light but encountered difficulty in doing so with amber, red, and far-red lights in the marmoset auditory cortex.

Considering the signal amplitude, the ambiguity in signal polarity, the consistency, and robustness in mapping the auditory cortex, the shorter wavelengths (blue and green) are more suitable for mapping cortical hemodynamic responses across multiple sensory modalities through the intact skull in awake marmosets. Additionally, our green wavelength (530 nm) is near an isosbestic point. Measurements taken with this wavelength would be more in line with other approaches that emphasize cerebral blood volume (e.g., fMRI with iron oxide contrast agents[29]). We chose the green light as the major illumination source in our measurements. Interestingly, this wavelength choice leaves room for further penetrability improvement with a longer wavelength (e.g., NIR) and for potential application in other species with thicker skulls (e.g., macaques).

**Estimating the photon visitation depth.** Utilizing Monte Carlo simulations, a previous computational study has shown how different polarization gating strategies can be used to probe different depths in the tissue. The study simulated and analyzed the photon visitation depth of polarized light backscattered from a scattering media[21]. A semi-infinite homogeneous medium was assumed to be absorption-free and comprised of Mie scatterers with the anisotropy factor $g$ of 0.92 and the scatterer size of 2.21 wavelengths. Illumination was simulated as a linearly polarized beam placed perpendicular to the medium surface. Photons were collected either through the co-linearly polarized channel or through the cross-

linearly polarized channel (polarization direction is relative to illumination). Photons that undergo relatively few scattering events and maintain their initial polarization contribute only to the co-linear channel, whereas photons that have random polarization through multiple (~ more than 10) scattering events contribute equally to the co-linear channel and cross-linear channel. The "linear-maintaining" channel was thus defined as the intensity difference between the "co-linear" channel and the "cross-linear" channel. We borrowed the estimated visitation depths of the "linear-maintaining" and the "cross-linear" channels from this study[21]. These visitation depths were expressed in the unit of the mean free path (MFP). To translate these results into a model consisting of two layers (the skull on top of the brain) with absolute thicknesses, we calculated MFPs [MFP = $1/\mu_s(\lambda)$] for both the skull and the brain, based on the parameters mentioned above (absolute visitation depths shown in Fig. 1g). It is worth noting that there are several assumptions in the current model that may affect the degree it resembles the real condition: (i) the refractive indices of the skull and the brain were assumed equal; (ii) both layers were assumed absorption-free; and (iii) no other structures were assumed between the skull and the brain, such as the dura mater. All these factors may contribute to a bias in estimating an absolute visitation depth. For example, as the real refractive index of the brain is lower than that of the skull, total reflection may happen when photons in the brain try to re-enter the skull and be eventually received. This effect would produce a sharp decrease in visitation depth at the border between the skull and the brain. The weak but definitive absorption in the skull could also shift the peak of the visitation depth to a reduced depth for both channels. Nevertheless, the relative scale of the visitation depths estimated here may provide a reference to guide practice.

**Estimating the lateral resolution.** We used Monte Carlo (MC) simulation to estimate the lateral resolution of our imaging approach. In the simulation model, the medium consisted of three stacking layers, each mimicking the skull, the gray matter, and the white matter. For the backscattered light collected from a point on the surface of the skull, its intensity would be affected if there is an absorption change in the gray matter underneath the skull. The more laterally displaced the change is from the point, the less sensitive the response is to the change. The lateral resolution of the signal recorded on this point was described as the normalized lateral profile of its sensitivity to the absorption change in the gray matter. We did not differentiate photons by their polarization, since the assumed thickness of the skull, 0.50 mm[6], is thicker than the travel distance needed for linearly polarized light to be completely depolarized [$1/\mu_s(\lambda)$][20], estimated ~0.4 mm for the green light. For photons that eventually contribute to the sensitivity of the absorption change, they must travel through the skull to reach the gray matter and are thus already depolarized. Therefore, differentiating these photons by their polarization states would not result in much difference in estimating the lateral resolution here.

The reciprocity rule is used in our simulation; that is, the trajectory of light propagation in one direction is equivalent to the reversed direction. To simulate collecting photons that exit the medium from a specific point on the skull surface, we reversely launched photons into the medium at this point. The launched photons may eventually leave the medium from the top surface and be collected, which reversely simulates the illumination process. In our live experiments, the epi-illumination was applied through the same objective (NA = 0.03) that collected backscattered light (the back focal aperture of the objective was fulfilled in illumination). We thus limited the photon launching and collection angles in the simulation within this range (NA ≤ 0.03).

The simulation was performed in MATLAB (Mathworks) with a GPU-based simulation package (MCXLAB[79]). The medium was set to be a finite cube with a dimension of $10 \times 10 \times 10$ mm³. The three layers of the medium (skull, gray matter, and white matter) were configured according to the parameters listed in the Table 1.

The optical properties of the skull and the brain were derived from the same data sources and assumptions mentioned in the previous section. $\mu_a(\lambda)$ is the absorption coefficient. $\mu_s(\lambda)$ is the scattering coefficient. $g$ is the anisotropy factor and was further assumed as 0.92 for the skull[21]. The thickness of each layer was assumed as 0.5 mm for the skull[6], 1.3 mm for the gray matter[80], and 8.2 mm for the white matter. The refractive indices were assumed as 1.56 for the skull[81] and 1.37 for the gray and white matters[79].

In the initial photon trajectory simulation, all layers of the medium were first set free of any absorption. $1 \times 10^9$ photons were launched into the medium at the center point of the surface. The simulated trajectory of each launched photon was collected if its exiting angle from the medium was within the collection angle range (NA ≤ 0.03). To put absorption back into consideration, the weight ($W$) of each photon was determined by the Beer's Law based on its simulated trajectory:

$$W = e^{-\sum_{i=1}^{n} \mu_a(i) \cdot x(i)}, \tag{1}$$

where $x(i)$ is the pathlength of a photon traveled in the $i$th medium, and $\mu_a(i)$ is the absorption coefficient of that medium. Therefore, the sum of these detected photons' weights would reflect the light intensity detected by the camera.

To estimate the system's lateral resolution defined above, we divided the gray matter layer into a 2D grid of column-shaped voxels, each with a size of 0.033 mm $\times$ 0.033 mm $\times$ 1.300 mm (lateral width, lateral width, depth). To apply a perturbation MC method[82], we increased the absorption coefficient by 10% in a single gray matter voxel and recalculated each photon's weight based on its originally simulated trajectory. The resulting change in the total light intensity (ΔR)

**Table 1. Optical and physical properties of the skull and the brain.**

| Wavelength (nm) | 470 | 530 | 590 | 625 | 730 | 850 |
|---|---|---|---|---|---|---|
| Skull absorption $\mu_a(\lambda)$ (mm$^{-1}$) | 0.1137 | 0.1137 | 0.0750 | 0.0424 | 0.0274 | 0.0172 |
| Gray matter absorption $\mu_a(\lambda)$ (mm$^{-1}$) | 0.465 | 0.638 | 0.287 | 0.032 | 0.009 | 0.016 |
| White matter absorption $\mu_a(\lambda)$ (mm$^{-1}$) | 0.465 | 0.638 | 0.287 | 0.032 | 0.009 | 0.016 |
| Skull scattering $\mu_s(\lambda)$ (mm$^{-1}$) | 28.10 | 25.99 | 24.24 | 23.35 | 21.11 | 19.12 |
| Gray matter scattering $\mu_s(\lambda)$ (mm$^{-1}$) | 11.63 | 10.53 | 9.62 | 9.15 | 8.12 | 7.24 |
| White matter scattering $\mu_s(\lambda)$ (mm$^{-1}$) | 42.85 | 41.94 | 40.70 | 40.28 | 38.61 | 35.31 |
| Skull anisotropy factor $g$ | 0.920 | 0.920 | 0.920 | 0.920 | 0.920 | 0.920 |
| Gray matter anisotropy factor $g$ | 0.882 | 0.887 | 0.892 | 0.895 | 0.899 | 0.898 |
| White matter anisotropy factor $g$ | 0.790 | 0.810 | 0.830 | 0.836 | 0.858 | 0.871 |

thus reflects the imaging point's sensitivity to the voxel's absorption change. The same process was repeated for the center row of voxels that passes the imaging point. A normalized $\Delta R$ curve was plotted against the lateral distance of these voxels from the imaging point (Fig. 1h). This curve reflects the lateral profile of the imaging point's sensitivity to the absorption changes in these gray matter voxels. We used FWHM and standard deviation (σ) of the lateral profile as measurements of the lateral resolution.

**Stimulus delivery**. For auditory stimulation, digital waveforms (16-bit resolution, 100,000 samples/second conversion rate) were converted into analog signals on the same DAQ card that controlled the imaging acquisition. The analog signal was attenuated by a programmable attenuator (Tucker-Davis Technology, PA5) before feeding into an amplifier (Crown, D75) that powered a loudspeaker (KEF, LS50), placed ~1 m away in front of the subject. Sound levels were calibrated by placing a microphone (Brüel & Kjær, type 4191) at the animal's typical head position.

For visual stimulation, the display content was controlled by custom code in MATLAB, using the Psychophysics Toolbox extension[83]. An LCD monitor (LG, 32GK850F, 32-inch nominal size, 144 Hz refreshing rate, 2560 × 1440 resolution, 400 cd/m² maximal brightness) was placed ~75 cm away in front of the subject.

For somatosensory stimulation, air puffs were powered by a pneumatic air supply to a set of solenoid valves (SMC, S070M-6DC-40) with a 0.38 MPa pressure at the valve inlet. The outlets of the valves were further routed through 2 mm-inner-diameter 1.5 m long tubing and 10–30 cm long adjustable hoses (Loc-Line, 1/4″) before blowing air out of nozzles (Loc-Line, 1/16″ round nozzle). The airflow intensity out of the nozzle was tested on the experimenter's skin and confirmed to be appropriately mild before pointing onto the subject. An extra nozzle was placed about 3 cm away from the animal's ear (contralateral to the imaging chamber), with the airflow directed away from the subject. Air puffs at the same repetition rate were delivered through this nozzle whenever there were no somatosensory puffs delivered from the other nozzles to counterbalance the acoustic noise generated from the other somatosensory stimulating nozzles. All solenoid valves were switched by control signals generated on the same DAQ card that controlled the imaging acquisition.

The monitor, the loudspeaker, and the air nozzles, together with the XINTRINSIC imaging setup and the subject were positioned inside a double-walled soundproof chamber (Industrial Acoustics, custom model), whose interior was covered by 3-inch acoustic absorption foam (Pinta Acoustic, Sonex). Stimulations of different modalities were all synchronized with the imaging acquisition. The somatosensory and auditory stimulations shared the same imaging hardware clock and start trigger on the DAQ card, whereas the DAQ card reset the trial time clock used in MATLAB for visual stimuli generation at the beginning of every trial or cycle.

**Experimental design: modality parcellation**. For the parcellation experiment, the stimulating air nozzle was placed facing the subject and ~5–8 mm away from the subject's mouth. The airflow direction was slightly tilted upward to target the regions of the face and the mouth below the nasal and ocular areas. The LCD monitor for visual stimulation was placed between the loudspeaker and the subject. The sound level was re-calibrated to compensate for the effect of the monitor placed in between. A 396 s-long recording session simultaneously consisted of 18 cycles of a 22 s-long somatosensory stimulation trial, 20 cycles of a 19.8 s-long auditory stimulation trial, and 22 cycles of an 18 s-long visual stimulation trial (Fig. 2b). The somatosensory stimuli were a 6 s-long, 10 Hz-repetition-rate air-puff train, presented at the center of each somatosensory trial cycle (Fig. 2b, left). To generate each air puff, the solenoid valve was switched on for 0.05 s and then off (50% duty cycle). The auditory stimuli were a 9.8 s-long, 5 Hz-repetition-rate white noise burst sequence, presented at the center of each auditory trial cycle (Fig. 2b, middle). Each white noise burst was 0.1 s in duration, with 20 ms sine ramps at the onset and the offset. The burst was 80 dB SPL loud when it was on. The visual stimuli were radially outward moving dots varying in speed (Fig. 2b, right). The display background was black, and the dots were white. Each dot was 0.4° in viewing diameter. Initially, 180 dots were randomized in position within a 15°-radius display field. Whenever a dot moved out of this field range, another replacing dot was generated in the very center of the field with a randomized

moving direction. During each 18 s-long visual trial cycle, the instantaneous speed of all dots started at 0°/s for the first 2 s, then linearly increased to 16°/s during the next 5 s, stayed at 16°/s for the next 4 s, linearly decreased to 0°/s during the next 5 s, and stayed at 0°/s for the last 2 s of the cycle. Simultaneous eye-tracking was performed to make sure the subject was alert and generally kept their gaze in the center. In an additional control experiment (Supplementary Fig. 3), this simultaneous parcellation paradigm was compared with the results that were obtained in each individual modality separately. The similar session described above was repeated three more times in the subject M126D, each time with the stimulation only of a single modality, but not the other two modalities.

**Experimental design: tonotopy mapping**. For the tonotopy mapping experiment, two sessions were recorded to derive the tonotopic map in each subject. Each session was 400 s long and consisted of 20 cycles of a repeating 20 s-long trial (Fig. 3c). The trial started with 2.7 s of silence, followed by a sequence of 73 pure tone pips, and ended with another 2.7 s of silence. Each pure tone pip was 0.2 s in duration and had 20-millisecond sine ramps at both the onset and the offset. All pure tone pips were played at 50 dB SPL. The "downward" session (Fig. 3c, left) had a trial cycle with the tone pip sequence decreasing in frequency for 6 octaves (72 semitones), starting with a pure tone pip of 28,160 Hz, continuing with each of the following pips descending one semitone in frequency from the previous pip, and ending with a pure tone pip of 440 Hz (standard tuning frequency A440 [ISO 16]). The "upward" session (Fig. 3c, right) was the opposite and had a trial cycle with the pure tone pip sequence starting with a pure tone pip of 440 Hz and ending with a pure tone pip of 28,160 Hz.

**Experimental design: motion and retinotopy mapping**. For the motion sensitivity and retinotopy mapping experiment, four sessions were recorded in each subject. All sessions were designed based on 0.4°-diameter white dots moving radially outward within a 15°-radius display field and on a black background[84]. Each session was 400 s long and consisted of 20 cycles, each 20 s long. The first session was to re-map the spatial extent of motion sensitivity to moving dots (Fig. 5c), and the design was very similar to the visual part of the parcellation experiment. During each cycle, the instantaneous speed of all dots started at 0°/s for the first 2 s, then linearly increased to 16°/s during the next 6 s, stayed at 16°/s for the next 4 s, linearly decreased to 0°/s during the next 6 s, and stayed at 0°/s for the last 2 s of the cycle (Fig. 5c, right). The 16°/s speed of moving dots is very close to the median of preferred speeds to moving dots by marmoset MT neurons[85]. The second session was to map the retinotopic eccentricity (Fig. 5e). The dots moved outward at a speed of 16°/s, within a range determined by eccentricity, and otherwise stayed static. The inner and the outer limits of the allowed motion range were both sinusoidally modulated and synchronized with the 20 s-long cycles at a cosine phase (Fig. 5e, right). The outer limit started at a 15°-radius at the beginning of each cycle and dropped to 5° at the middle of each cycle, whereas the inner limit started at a 5°-radius at the beginning of each cycle and dropped to 0° at the middle of each cycle. Thus, at the beginning of each cycle, only the periphery of the field (5°–15° radius) contained moving dots, leaving the center of the field (0°–5° radius) with static dots, whereas at the middle of each cycle, only the center contained moving dots, leaving the periphery with static dots (Fig. 5e, left). The third (Fig. 5g) and the fourth (Fig. 5i) sessions were to map the retinotopic polar angle tuning. The dots moved outward at a speed of 16°/s, only within a range determined by polar angle, and otherwise stayed static. This motion range was a quarter circle in shape, with its middle line pointing at the 12 o'clock polar angle at the beginning of each cycle. The shape of this range persisted while the angle at which the range middle line pointed was swept either clockwise (Fig. 5g) or counterclockwise (Fig. 5i) for each session. The sweeping speed was 0.05 rounds/s and was thus synchronized with the 20 s-long cycles. Different from the first two sessions, in which the motion range patterns at any moment were always rotationally symmetric and thus did not carry a natural bias to drive the subject's attention to any specific polar angle at any moment, the motion range pattern of these polar angle testing stimuli was not rotationally symmetric at a given moment and may carry such an attention bias. To draw the subject's attention to the center, we displayed random small marmoset pictures (~3° × 3°) in the center of the field on the top of

the dots during these polar angle testing sessions. Each marmoset picture lasted for 2 s. These pictures were acquired through Google Images and further cut and scaled to the desired size. Simultaneous eye-tracking was performed during all sessions (see below for details). Indeed, the eye-tracking results showed the subject's gaze was kept predominantly to the center of the field in these polar angle sessions (Fig. 5n, lower two panels), with 69.1 and 66.7% of the total detected gaze positions falling in the center 5°-radius range, compared to 63.1 and 65.8% in the first two sessions (Fig. 5n, upper two panels).

**Experimental design: somatotopy mapping.** For the somatotopy mapping experiment, two stimulating nozzles were configured. The oral nozzle was placed at but not touching the subject's incisors, whereas the facial nozzle was placed to target the cheek contralateral to the imaging side but kept ~5 to 8 mm away from the subject (Supplementary Fig. 8b). A 437 s-long recording session simultaneously consisted of 19 cycles of a 23 s-long oral stimulation trial and 23 cycles of a 19 s-long facial stimulation trial (Supplementary Fig. 8d). The oral stimuli were a 4 s-long, 10 Hz-repetition-rate air-puff train presented at the center of each oral trial cycle (Supplementary Fig. 8d, left), whereas the facial stimuli were another 4 s-long 10 Hz-repetition-rate air-puff train but presented at the center of each facial trial cycle (Supplementary Fig. 8d, right). Two parallel solenoid valves were routed together to power the facial nozzle. To generate each air puff, the corresponding solenoid valves were switched on for 0.03 s and then off (30% duty cycle).

**Experimental design: face patch mapping.** For the face-patch mapping experiment, six categories of visual objects were tested: marmoset faces, marmoset body parts, animals, fruits and vegetables, familiar objects, and unfamiliar objects. The images of marmoset faces were obtained from the authors of the pioneer marmoset face-patch mapping study[52]. The images of familiar objects were taken in our colony and laboratory at Johns Hopkins. The images of other categories were searched and selected using Google Images. Visual objects were extracted from the original images, converted to grayscale, and further equalized for contrast and average luminance[86]. The objects were scaled and rotated to minimize the variation in aspect ratio and to minimize the variation across categories for the mean value and the standard deviation of object size (defined as the area each object covers) within each category. Two additional categories of scrambled faces were generated. For both spatially scrambled and phase-scrambled faces, the same contour mask of the corresponding face was applied back to the scrambled faces so that the original contour edges remained. Together, 20 exemplar objects were generated for each of the 8 categories (Fig. 6a and Supplementary Fig. 7). To synthesize a stimulus frame for each object, the object was repeated and tiled as a stretcher bond pattern in the frame (Fig. 6c). Each "tile" was about 10° × 6° in size (480 × 288 in display pixel). Considering the range of marmoset gaze is limited and is no larger than the stimulus frame displayed[46], no matter where the subject gazed, it could not fully avoid the object within the 5°-radius center of its visual field. The background of the frame was set as a randomly generated pink noise image that matched the average luminance of the objects (115 on 8-bit grayscale). To test the cortical response to a category, a 20 s-long trial was generated online with 40 display frames (Fig. 6b). Each frame lasted for 0.5 s. The first 4 frames were with pink noise background only, followed by stimulus frames of all 20 exemplar objects of the category in a randomized order, and then ending with another 16 pink noise background only frames. Stimulus trials of these 8 categories were presented in a randomized order for each testing cycle. A testing block contained two testing cycles and was therefore 320 s long. A random music selection from spotify.com was played through the loudspeaker during testing blocks to keep the subject alert. The subjects rested between testing blocks. On the first day of the experiment, a reference image of the imaging field of view was taken. On each following day, this reference image was displayed simultaneously with the real-time image (through different color channels) to align the images and maintain a consistent field of view. Testing blocks acquired on different days were then pooled together for analysis. Simultaneous eye-tracking was performed with the imaging acquisition.

**Eye-tracking.** To track the subject's gaze position during spontaneous eye fixation, an eye-tracking camera (FLIR, FMVU-03MTM-CS; Edmund Optics, 86-410), and a collimated NIR LED (Thorlabs, M850L3, and ACL5040U-DG15-B) were placed right above the monitor to track the animal's eyes during visual experiments. The eye-tracking camera was controlled by a trigger sequence at 10 fps generated on the same DAQ card that controls imaging acquisition and stimulus delivery. To detect fixed gaze positions, we implemented an algorithm that utilizes circle Hough transform to search for a stable circular shape based on the pupil edges in each frame. any significant saccadic transition during a frame would smear the pupil image and make the pupil position "undefinable" by our algorithm.

To determine the absolute position of the subject's gaze, calibration sessions were performed in addition to other visual experiments. During each day of visual experiments. 80 marmoset images were acquired from Google Images. Each image was cut and scaled to a size of 160 × 160 pixels. The screen was divided into a 15 × 9 location grid to display these images without overlap and with one right in the center. Among these 15 × 9 locations, 37 locations had their centers within the center 23° diameter range of visual angle. An image was displayed at one of these 37 locations for 1.5 s, with the rest of the monitor displaying a gray

background. 80 images were displayed in a pseudorandomized order. Eye-tracking recording from this session (120 s in duration) was later used for calibrating the absolute gaze positions in the other experimental sessions. This calibration grid generally covers the entire marmoset oculomotor range, which is more limited than that of macaques[46].

**Optical imaging through a cranial window.** The basic procedures for chronically implanting an artificial dura-based cranial window in NHPs have been previously described[5]. After through-skull imaging, the mapping results were used to mark a target position for a subsequent craniotomy within the imaging chamber. To protect the planned craniotomy, a rubber piston cover was taken from the plunger of a standard 10 ml disposable syringe and was put on top of the targeted position. Additional dental cement material was added for ~1 mm in height around the rubber cover to form a socket and prevent the cover from sliding. The window implantation surgery was performed under sterile conditions when the animal was anesthetized with isoflurane (0.5–2.0%, mixed with pure oxygen). The rubber cover was taken away and the craniotomy and durotomy over the planned position were performed. An artificial dura pre-molded with silicone in a hat-like shape[5] was then implanted. The gap between the craniotomy edge and the sidewall of the artificial dura was filled and sealed with surgical silicone adhesive (WPI, Kwik-Sil). The rubber cover was put back into the pre-built socket to protect the cranial window and sealed to the original recording chamber wall by a dental impression material (GC America, EXAMIX NDS injection).

After the animal fully recovered from the window implantation surgery, intrinsic signal optical imaging was performed through the cranial window (Fig. 4d). Since the cranial window was ~9.5 mm × 6.4 mm in size (with one corner retracted) to target the entire auditory cortex on the brain surface, we swapped out the 1× objective (used for all through-skull experiments) and replaced it with a 2× objective (Thorlabs, TL2X-SAP). The same tonotopy mapping experiment (Fig. 3c) was performed through the cranial window. The results were analyzed and visualized (Fig. 4e) in the same way as in the analysis of through-skull results.

**General processing of the imaging data.** The through-skull intrinsic signal on each pixel was quantified and expressed as normalized intensity change to a baseline intensity in re-emitted back-scattered light (or $\Delta R/R$, to be consistent with the literature in which the recorded intensity is also called "reflectance"). The baseline was loosely defined as the average intensity across the session (Fourier analysis), or the mean intensity over the pre-stimulus interval of a trial (Figs. 2d, 3d, 6d and Supplementary Fig. 9b, g, l, q), or the maximal intensity over a trial (Fig. 4c, f). Since the intensity change is relatively very small to the baseline ($\Delta R/R$ is typically at single-digit permille [‰] scale), these variants on baseline definition result in an unnoticeable difference in quantifying the relative change in $\Delta R/R$.

Temporal traces over multiple repetitions of a cycle or a trial were quantified in mean values and standard errors of the mean (SEMs) of $\Delta R/R$. SEM was defined as $\sqrt{(E[X^2] - (E[X])^2)/n}$, where $X$ is a single simple of $\Delta R/R$, and $n$ is the total repetition number.

Additional motion correction was performed on the modality parcellation recording data (480 × 300@80fps) and the results are shown in Supplementary Fig. 4f–i. The first frame of each session was used as a template for registration. All other frames were aligned with the template through rigid registration with NoRMCorre[87]. The displacements in individual frames are at sub-pixel scales in each of the subjects. Although motion correction may help remove some high-spatial-frequency noise (< 0.1 mm), the low-spatial-frequency features (> 0.1 mm) are well preserved regardless of motion correction.

**Intrinsic signal polarity.** Signal polarity, defined as whether an overall positive or negative change in $\Delta R/R$ indicates a greater response to a stimulus, varies across wavelengths and was empirically determined for each wavelength from the activation patterns of all three tested modalities (Supplementary Fig. 1b–d) as negative for 470 nm (blue), negative for 530 nm (green), ambiguous but presumably positive for 590 nm (amber), positive for 625 nm (red), positive for 730 nm (far-red), and negative for 850 nm (NIR). The observed signal polarity difference among wavelengths is correlated with and can be explained by the HbO/HbR molar absorptivity ratio (Supplementary Fig. 1f). Any wavelength with this ratio higher than 0.5 (blue, green, and NIR) showed a negative signal polarity, whereas any wavelength with this ratio lower than 0.5 (red and far-red) showed a positive signal polarity. The amber color with a ratio of ~0.5 showed ambiguity in signal polarity. It was suggested the major hemodynamic response following a stimulus consists of an HbO increase and a HbR decrease[88]. Assuming the HbO increase is ~twice as the HbR decrease (demonstrated in Supplementary Fig. 1g), the wavelengths with higher ratios (blue, green, and NIR) would pick up the HbO increase more over the HbR decrease and are thus more absorbed during the hemodynamic response (negative $\Delta R/R$), whereas the wavelengths with lower ratios (red and far-red) would pick up the HbR decrease more over the HbO increase and are thus less absorbed during the hemodynamic response (positive $\Delta R/R$). The amber light may pick up these opposite changes in a more balanced way and is thus ambiguous in signal polarity.

It is worth noting that, although the assumed negative signal polarity for the lower-ratio wavelengths (red and far-red) resulted in activations in the putative

auditory cortex, the response patterns are substantially different from the auditory response patterns measured with the higher-ratio wavelengths (green, blue, and NIR) (Supplementary Fig. 1c). This is different from the somatosensory and the visual response patterns, which are consistent across all measurement wavelengths (except for the amber light). Many previous attempts using intrinsic signals to map the auditory cortex in a variety of species (chinchillas[22], cats[23,26], ferrets[24,25,27,28]) only succeeded with green wavelengths but not with red wavelengths. These studies, together with our data here, imply that the HbR hemodynamic response may behave differently in the auditory cortex compared to other cortical areas. Further investigations are needed to reveal more detailed hemodynamic similarities and dissimilarities in different cortical areas and species.

**Imaging analysis for phase-encoded experiments**. A phase-encoding strategy[30,31,38] was utilized in designing all these experiments except for the face-patch experiment. To analyze phase-encoding experiments, the trace of each pixel across the entire recording session was Fourier transformed (examples see Figs. 2c, 3e). Signal polarity was compensated to calculate each pixel's phase response to the stimulus at the corresponding frequency.

Research has suggested that hemodynamic response begins within ~0.5 s of stimulus onset and peaks at 3–5 s after the onset[88]. To decode the preferred stimulus phase (or tuning phase) from the raw phase in response, it is necessary to compensate for this hemodynamic delay. For both the tonotopy mapping and the retinotopy polar angle mapping experiments, two sessions were performed with stimuli that were reversely identical to each other. The "upward" pure tone pip sequence was the time-reversal of the "downward" pure tone pip sequence, whereas the clockwise polar angle sweep was the time-reversal of the counterclockwise polar angle sweep. Thus, by assuming the hemodynamic delays of a pixel in these two temporally "mirrored" sessions are identical[31], this delay can be numerically derived or canceled out by reversing the response phase of one session and combining it with the response phase of the other session (Fig. 3c–g). The equations in Fig. 3c show how the tuning phase and the hemodynamic delay for the exemplar pixel labeled in Fig. 3f. were derived from the raw response phases shown in Fig. 3d. To quantify the distribution of hemodynamic delays across pixels in these sessions, we empirically considered the top 10% of pixels, ranked by their mean response amplitudes at the corresponding frequency in these two sessions, as candidate pixels to estimate the hemodynamic delay. These pixels were color-coded for the derived hemodynamic delay in each subject's delay map (Fig. 3g and Supplementary Figs. 5g, 6d). The delay values of these pixels were counted and shown in a histogram with the same color code (Figs. 3g, 5k and Supplementary Figs. 5h, 6e). These delay values largely fell within a range of 2.0–5.7 s. A peak of this range in the histogram was located within a narrower range of 3.2–4.4 s. By assuming 2.0–5.7 s as the empirically acceptable delay range, we further derived the average hemodynamic delay in each subject among candidate pixels. For the tonotopy experiment, five tested subjects each had an average delay of 3.94, 3.61, 3.47, 4.46, and 4.12 s, with a group average of 3.9 s (Supplementary Fig. 5h). For the retinotopy polar angle experiment, five tested subjects each had an average delay of 3.34, 3.42, 3.72, 2.96, and 3.89 s, with a group average of 3.5 s (Supplementary Fig. 6e). Thus, to decode the tuning phase from the raw response phase for a single session, we compensated 3.9 s for the hemodynamic delay in tonotopy tuning phase analysis (Supplementary Fig. 5), 3.5 s in retinotopy tuning phase analysis (Supplementary Fig. 6), and 3.7 s in all other single-session tuning phase analyses.

The tuning maps were drawn after compensating for signal polarity and delay at the corresponding frequency for each pixel. The HSV (hue-saturation-value) color space was utilized to visualize both responsiveness and tuning for each pixel. The tuning phase was encoded in the hue (color) channel, whereas the response amplitude of the corresponding frequency was encoded in the saturation (chroma) and value (intensity) channels. Since the saturation and value channels can only have values between 0 and 1, for most of the maps, we encoded the response amplitude at the 99th percentile of all pixels as the upper limit value 1 in both saturation and value channels (Figs. 2e, 5d, f, h, j, k and Supplementary Figs. 1, 3, 4, 6, 8), except for the amber light in Supplementary Fig. 1 (97th percentile), the "null-stimulated" components in Supplementary Fig. 3 (following the 99th percentile of the corresponding components of the simultaneously mapped session), Supplementary Fig. 10b, d, f (95th percentile), and the subject M117B in Supplementary Figs. 4, 5, and 10 (97th, 95th, and 93rd percentile, respectively). The upper display limit of response amplitude was set at absolute values as indicated in Figs. 3f, h, 4a, b, e and Supplementary Fig. 2. For tonotopy visualization, pixels that had their tuning phases outside the time range of the tone pip sequence or their delay values outside the acceptable range were set to 0 in their saturation channel. To visualize the retinotopy tuning together with the motion sensitivity, we encoded retinotopy polar angle tuning in the hue (color) channel, retinotopy eccentricity tuning in the saturation (chroma) channel, and motion sensitivity in the value (intensity) channel (Figs. 5l, m, 7a, e, f and Supplementary Figs. 6i, 9f, k, p, 10h, 11a). To generate a summary map for a session designed with multiple stimulation frequencies (e.g., parcellation, somatotopy experiments), the RGB (red–green–blue) color space was utilized to display maps of different frequencies together (Fig. 2f and Supplementary Figs. 1e, 3e, 4e, 8f, 8j). For the map of each stimulation frequency, a mask was first generated with pixels that had tuning phases falling into the trial time range corresponding to the most intense stimuli with 1 s tolerance at either side. Pixels included in this mask had their amplitudes in the original HSV

"value" channel further shown in the designated RGB color channel (red, green, or blue).

**Imaging analysis for trial-based experiments**. For the face-patch experiment, 218, 134, and 234 testing cycles were recorded in the subjects M126D, M15E, and M117B, respectively. The pixel response trace of each trial was normalized to the pre-stimulus baseline, defined as the average signal value of the first 2 s in the trial (Fig. 6d and Supplementary Fig. 9b, g, l). To quantify the overall response in each single trial, a $\Delta R/R$ value was calculated for each pixel by averaging the normalized and polarity-compensated response within a time window of 4–10 s after the first stimulus frame onset. Two t-value maps were drawn for each subject. One map was to compare response values between those to the face category and those to all other five object categories combined (Fig. 6g and Supplementary Fig. 9e, j, o). The other map was to compare response values between those to the face category and those to the two scrambled face categories combined (Fig. 6f and Supplementary Fig. 9d, i, n). The definition of the t-value follows the $t$-statistic in Welch's t-test, as $t = (\bar{X}_1 - \bar{X}_2)/\sqrt{s_1^2/N_1 + s_2^2/N_2}$, where $\bar{X}_j, s_j, N_j$ are the $j$th sample mean, sample standard deviation, and sample size, respectively ($j \in \{1, 2\}$). At our sample sizes ($n > 100$), a t-value $>2.0$ or $>3.4$ corresponds to a $p$-value $< 0.05$ or $<0.001$ in a two-tailed Welch's t-test.

**Functional landscape assembly**. To put maps of different experiments together (Fig. 7a and Supplementary Figs. 9f, k, p), a static surface image was obtained from each experiment as a co-registration reference. The reference image from face patch mapping was used as a template and the reference images from all other experiments were affinely transformed to match this template. A manual sanity check was further performed by checking the co-alignments of small features on the skull surface or chamber bottom, to make sure there was no misalignment at any scale comparable to our estimated imaging resolution (~0.6–0.7 mm). The derived transformations were subsequently applied to the corresponding functional map to co-register these maps together on top of the reference image from face patch mapping (also dimmed and served as the display background). Maps of t-values were also co-registered to the same map but shown as iso-t-value contour lines. Together, these maps formed a cortical functional landscape in each subject.

**Through-skull & through-window comparison**. To compare through-skull and through-window maps, linear regression models were fitted in MATLAB (function "fitlm", with the "RobustOpts" option on). The two maps were manually aligned first (Fig. 4b, e). A loose criterion was setup for selecting candidate pixels to be analyzed by the linear models. A pixel has to show moderate response amplitudes (> 0.2‰ for through-skull, > 1.5‰ for through-window) and tunings to tones (not to silence) in both maps to be considered as a candidate pixel for later analysis. The candidate pixels were further constrained to those within the cranial window and on the superior temporal gyrus. In Fig. 4g, a linear model without a constant term was fit between through-window amplitudes (the independent variable $x$) and through-skull amplitudes (the dependent variable $y$). The resulted equation is $y = 0.125x$, with a $t$-statistic of 528.43 for the coefficient ($p$-value $= 0$). Another linear model (Fig. 4h) was fit on the octave scale (starting at 440 Hz) between through-window tunings (the independent variable $x$) and through-skull tunings (the dependent variable $y$). The resulted equation is $y = 0.675x + 0.502$, with a $t$-statistic for the slope and intercept are 363.63 ($p$-value $= 0$) and 92.416 ($p$-value $= 0$). For generating an artificially diffused through-window map at a certain FWHM scale (Fig. 4i), a convolution kernel was scaled from the lateral profile simulated in Fig. 1h (green light, original FWHM is 0.63 mm). The original through-window map was convoluted with the kernel (in a complex plane with polar coordinates composed of response amplitude and response phase), the amplitudes and tunings of the diffused map were then used as the dependent variable for the similar linear models described above to estimate the coefficient of determination $r^2$ for both response amplitude and tuning (Fig. 4j).

**Structural MRI parcellations**. After euthanasia, one subject (M126D) was perfused transcardially with 4% paraformaldehyde (PFA). Before MRI scanning, the brain sample was immersed in gadolinium MR contrast agent solution (1:200 dilution, 0.5 mmol/ml) mixed with Phosphate Buffer Saline (PBS) solution for 7 days[55] to reduce the T1 relaxation time. Ex-vivo MRI T2w images were acquired on an 11.7 T MRI platform (Bruker Biospin) with an 8-channel surface coil. The T2w images were collected using a 3D FLASH sequence (TE = 3.5 ms, TR = 25 ms, flip angle = 20°, resolution = 0.15 mm isotropic). The 3D cortical surface of the marmoset brain was reconstructed using Freesurfer (http://surfer.nmr.mgh. harvard.edu/). To localize the anatomical labels of the marmoset brain, the Nencki-Monash template[54], and the NIH-MBA v1.1 template[55] were registered to the T2w images using affine transformations implemented in Freesurfer.

The surface of the extracted brain was further optically imaged from the angle that mimicked the view of through-skull functional imaging. Before euthanasia, a cranial window similar to the one shown in Fig. 4 was implanted in this subject. Through-window tonotopic gradients were thus also acquired in this subject. The registration of the structural MRI-based parcellation to the functional landscape imaged through-skull was accomplished in the following steps. First, the MRI structural parcellations were rotated in 3D to match the view with and be registered

to the surface image of the extracted brain, using anatomical markers of the brain contour and sulci. Second, the brain surface image was registered with the through-window tonotopic map using anatomical markers of the cortical vasculature pattern and the lateral sulcus. Third, the through-window tonotopic map was registered with the through-skull tonotopic map following the similar fashion shown in Fig. 4. All three registration steps were performed using rigid transformations.

**Additional resources**. Additional information about the XINTRINSIC setup, full part list, design notes, assembly instruction and program code can be found https://x-song-x.github.io/XINTRINSIC/.

**Reporting summary**. Further information on research design is available in the Nature Research Reporting Summary linked to this article.

## Data availability
The raw data supporting the current study have not been deposited in a public repository because of the large size of the dataset but are available from the corresponding authors upon request.

## Code availability
The data collection[89,90] and analysis[91] routine was performed with custom MATLAB code. The code for these routines is publicly available[89–91].

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

## Acknowledgements

This research was supported by National Institutes of Health grants DC003180 and DC005808, DARPA Cooperative Agreement N66001-17-2-4008 to X.W. X.S. was supported by a fellowship from the Kavli Neuroscience Discovery Insitute at JHU. We thank S. H. Park and D. Leopold for providing marmoset face images, J. Lynch, K. Schonvisky, S. Miller, E. Easter, and J. Izzi for assistance with surgeries and animal care, Q. Fang for technical advice on the MCXLAB simulation toolbox, A. W. Roe for sharing cranial window related protocols, and E. Issa, X. Liu, L. Zhao, and M. Osmanski for their comments on the earlier versions of the manuscript. Creation of datasets accessed on MorphoSource was made possible by the following funders and grant numbers: NERC NE/G001952/1 to N.S. Jeffery, NSF BCS 1304045 and a research grant from Trinity College of Arts and Sciences to C. Wall, and NSF BCS 1540421 to G.S. Yapuncich and D.M. Boyer.

## Author contributions

X.S. and X.W. designed the study. X.S., Y.G., C.C., and X.W. developed the marmoset optical imaging preparation. X.S. developed the XINTRINSIC setup. Y.G. performed the Monte Carlo simulation. C.C. and X.S. designed the pilot through-skull testing. H.L. and X.S. designed the eye-tracking system and the stimuli in the face-patch mapping experiment. X.S. designed the mapping experiments and analyzed the data. X.S., Z.S., J.H.L., and Y.G. performed the mapping experiments. Y.Z. performed structural MRI scan and analysis. X.S., Y.G., and X.W. wrote the manuscript with input from all authors. X.W. supervised the study.

## Competing interests

The authors declare no competing interests.
