## [Peer review file · Nature Communications]

REVIEWER COMMENTS

Reviewer #1 (Remarks to the Author):

In this study, Song et al. applied a polarization-gated wide-field optical imaging method to detect intrinsic signals over adjacent auditory, visual, and somatosensory cortices in awake marmosets. The authors used this method to clearly show that tonotopy, retinotopy, and somatotopy are distributed in each corresponding area. In addition, face patches were included in the motion-sensitive region. The methodology used for the measurements and analyses is described in detail. The basic features of the mapping were reproduced over five animals and the basic optical properties (lateral resolution and depth penetration) of the method were well estimated. Although the lateral resolution of 0.3–0.7 mm is lower than those of intrinsic imaging and two-photon imaging through a cranial window, this method can resolve the frequency gradient in the tonotopy, and the angle and eccentricity in the motion sensitivity. The color map of the functional landscape in Fig. 7 is very impressive. Thus, I agree that the present method is very useful for noninvasive mapping of the neural responses to multiple modalities over the cerebral cortex in the marmoset.

However, my major concern is that the authors do not make full use of the advantages of the wide area that can be simultaneously imaged. For example, if the face of a marmoset moves, is a stronger response evoked than the linear sum of the responses when the static image of the marmosets and moving dots are presented? If the face and vocalization of the marmoset are presented at the same time, do the signals in the auditory cortex and MT/V5 differ compared with when each is presented separately? To fully demonstrate the usefulness of this method, it is essential to show not only that the different functional modalities in different sessions can be parcellated in the same animal, but also that there are things that can be simultaneously seen for the first time there.

Minor points:

The authors should quantitatively estimate the maximum size of the cortical area that can be imaged simultaneously. It is important to show the extent to which the through-skull imaging outperforms through-window imaging in respect to the area imaged because spatial resolution is higher in the latter imaging. For how long were the authors able to keep looking at the same area? The edge of the area where the skull was exposed should be clearly shown in the figures. I am not sure what the white and gray pixels in figures such as Fig. 3I indicate.

I'm not convinced that there is a newly discovered pure-tone-responsive tonotopic gradient. In the through-window imaging in Fig. 4E, there are certainly blue and purple regions in the parabelt, but there is no clear gradient band from RT to the parabelt. Blue and purple regions are also observed in the opposite site near the lateral sulcus. What is the difference between them? In addition, Fig. 3I indicates that this gradient was ambiguous in many animals. The conclusion regarding this tonotopic gradient should therefore be weakened.

Is it possible to quantify the variability of the response regions among animals? If it is large, this mapping could be used to accurately determine the target area before two-photon imaging or electrical recording. Please discuss this issue.

W118B was the oldest animal and the mapping in this animal appears to have the poorest resolution. Please discuss this matter and give possible reasons.

The equations shown in Fig. 3D are essential to calculate the amplitude of f . It would be helpful for many readers if the authors explained these equations in the Methods.

Was there any motion artifact in the awake condition?

There are many typos (for example, eah in line 822 and emmitted in line 823). Please carefully check for them.

Reviewer #2 (Remarks to the Author):

The technique of optical imaging used for brain mapping is usually invasive, requiring the implantation of a cranial window. This creates challenges, for example, for long-term studies. Although earlier studies in mice have shown the feasibility of obtaining some signals through the skull, this has not been possible in primates. This paper by Song et al. provides a breakthrough in this respect, allowed by innovative application of a cross-linear polarization enhanced imaging. This results in the ability to image the marmoset cerebral cortex through the intact skull, at a relatively affordable cost. Compared to conventional optical imaging, this method allows a nearly four times larger field of view.

To validate this technique the authors confirmed several functional signatures of the mesoscale organization, previously derived from electrophysiology, such as tonotopy and face-sensitive patches. It could be argued that not much has been shown in terms of what we know about the marmoset cortex, but this is not (in my view) the point of the paper – the key message is that something that has been thought to be impossible can be done, and evidence is provided that this can be done reliably. I am

convinced of the value of this technique as a scouting tool for placement of electrode arrays, tracer injections, or optogenetic vectors, and it may also be useful for studies of brain plasticity. Its general applicability will be significant for the field, and the paper is likely to become well cited. However, there are several technical points that should be rectified or clarified before I can recommend the paper for publication in Nature Communications.

1. Some clarifications needed: one weakness of the present paper is that some of the putative new findings (new face-motion patch, and new tonotopic gradient) are not related to current knowledge of the marmoset cortex neuroanatomy. Have the brains of these animals been sectioned for histology, or was a MR obtained? I am having trouble relating these findings to currently recognized areas. If a MR was obtained (or can be obtained) it would be important to register the location of the patches to one of the current computational templates (e.g. Majka et al. Neuroimage 2021). If histological sections were obtained, then correlation with the Paxinos atlas should be possible. For example, if the face patch was located in V4T, this would correlate nicely with the observation that V4T contains islands of cells that are motion-sensitive, and cells that are purely orientation-selective (Rosa and Elston 1998). If the patch is in TEO, this would be interesting in showing that this cytoarchitectural area may contain subdivisions. Likewise, could the new tonotopic gradient be related to the observation that RT, but not R or A1 have connections with the medial prefrontal cortex, which has been hypothesized to have a role in interpreting affective content of vocalizations (Reser et al. 2009)?

2. Quantification of the functional landscape: The authors reported many interesting observations that could be novel, important findings on the functional landscape of the dorsal areas.

a. For one, it is stated on page 13: "the face patch (contrasting faces to other objects, indicated by cyan contours in Fig. 7, see also Fig. 6G) was located within the motion-sensitive retinotopic region anteroventral to MT/V5, with a preference for the lower-center of the visual field ... These claims hold for all tested animals." It is hard to judge if the above statement applies to all the animals investigated. In animal M15E (Supp Fig 8K), the face patch appears to be located posterior-ventral, rather than anterior-ventral to the yellow region (MT/V5 - MTc/V4t border?). In M117B, the face patch appears to sit on the part of MT/V5, preferring upper-center rather than lower-center of the visual field (Reviewer figure 1B).

b. Another potentially novel observation is "close to this face patch, a part of the newly discovered tonotopic gradient (Fig. 3I) represents acoustic frequencies commonly found in marmoset vocalizations⁵³ (~3-11 kHz)." (page 13) Again, this is not clear in one of the three animals (M117B) as no tonotopy map is superimposed in Fig7P.

c. It is also stated that "This region-within-landscape paradigm may offer more replicable comparisons across individuals and studies than traditional approaches." on page 14.

These exciting observations and claims need to be consolidated by objective quantification of the imaging data. For this purpose, the following analysis of the acquired data is recommendable:

I. Obtaining areal borders in an algorithmic manner in each animal. A standard technique to determine the border is the visual field sign map (Serenio et al., 1994; 1995).

II. Alignment of the functional data to a common reference across animals. The lateral sulcus could be a potential candidate for the reference landmark. On the aligned brain image, please show functional results of each animal, such as locations of face patches. This would be a less desirable option if the analyses suggested under point 1 cannot be performed.

3. Source of artifactual signals: In Supplementary figure 3, at least in three (M15E, M8E and M7E) out of six marmosets investigated, clear artifactual signals appear systematically in the anterior part of the ROI. In these regions, the temporal delay in response to somatosensory stimuli is far from the expected one (cyan) with some non-random spatial distribution.

A similar but less obvious glitch is also seen in the other two animals (M44D, M117B), and in Supplementary Figure 4. These artifactual signals can be problematic when aiming at delineating smaller functional structures such as subregions encoding lower or upper lips somatosensation within the primary somatosensory cortex. Thus, a bit more investigation would be desirable as to why and how this artifact happened in these animals but not in the remaining one (M126D). Maybe the difference lies in smoothness of the surface of the imaging window, or composition of applied transparent glue.

4. Apart from the above major issues, clearance of the following minor issues would enhance the value of the paper:

a. The source of the signal enhancement. The reported signal is generally much larger than previously reported ones. For example, the signal through the cranial window in Figure 4F can reach $\sim 20\%$ dR/R. This response amplitude is nearly 20 times larger than that was reported in Tani et al (ref 15), who exploited a comparable auditory stimulation and green light (535nm) for marmosets. It is not clear whether this signal enhancement is due to some physiological conditions or the optical settings developed in this study. If the latter is the case, which exact component contributes to the signal enhancement is not readily penetrable to readers. For instance, the Koehler illumination appears to contribute to homogeneous illumination across the large field of view, but not to enhancing S/N.

b. Fig1B, For probing visual response, light from the screen must be isolated from the reflected light by the Koehler illumination. How was this achieved? An illustration or a picture depicting a light-isolation device would be helpful. Was this picture obtained through the cross-polarization or not?

c. Fig1H, It would be informative for the readers if the panel includes traces for the conventional optical condition for widefield imaging, where polarization was not used for both illumination and light collection. The peak of the normalized intensity is at around 0.3-0.4mm visitation depth. Please add a brief discussion of how this value is impacted by optical parameters other than the wavelength. Please extend the x-axis to include the visitation depth at which the light intensity returns to 0. This information will provide critical information to understand what depth of the brain the recorded optical signal stems from.

d. Fig1I, it is recommendable to include the Monte-Carlo simulation results where no polarization was used for both illumination and light collection. This will highlight how the light-polarization improves the lateral resolution. Also, it would be informative if the panel includes traces when there is no skull above the brain. This will enable readers to better interpret the imaging results without the skull, presented in Figure 4.

e. e. Fig2. How was $dR/R=0$ defined? In page 39, "The baseline was loosely defined as the average intensity across the session, or the mean intensity over the pre-stimulus interval of a trial, or the maximal intensity over a trial. ". Please describe which definition was used for Fig2.

f. p.7, "These locations are consistent with previous results using invasive electrophysiology methods" and Fig2D-F. To demonstrate this in an objective manner, it is advisable to superimpose areal borders already published (e.g., <https://marmosetbrainmapping.org>) onto the images of the cortical surfaces used in this study. This will help readers to understand how well the imaging technique can delineate brain areas, which appears to be one of the main scopes of this study. Also it could lead to scientific discovery of individual differences between marmosets.

g. Fig4G. The dotted abscissa seems to be a fitted line, but it may be misunderstood as an identity line. The panel would look more intuitively clear if the x and y axes were on the same scale as in panel H.

h. Fig4B, E, H, The difference between the two window conditions is not only the spatial resolution but also the shift in preferred frequency. For example, the preferred frequency of the two peaks in panel B is clearly higher than that in panel E. The spatial filtering (panel I) does not explain the shift but instead exaggerates the size of the shift. A possible explanation of this shift would be helpful.

i. Fig4J, It is not apparent why the best diffusion scales are considerably different between dR/R and tuning. A discussion regarding this would be useful.

j. Fig6, Was there a difference in behavioural response (eg. Pupil diameter) between the stimulus categories? Fig6D. What does dR/R mean in this panel? Considering 20s-long trial, signals at $t=0s$ and $t=20s$ must be identical, but they are not in this panel.

k. Fig7 also Fig6E-G, There are two regions that respond preferentially to face over scrambled faces (purple contours). The posterior one is interesting in that it responds less to the five other object

categories. Traces of this region, in the same format as Fig6D, would be helpful for readers to understand the response of this region.

l. Supplementary Fig4B,C,G,H,5D,E,8F,K,P, the color bars are missing.

m. Fig7 & Supplementary Fig8, To show how replicable the functional landscape across animals is, it would be informative to show positions of each functional areas (e.g., local peak motion-sensitive regions, face patch) on the brain across all the marmosets investigated. As a reference, the lateral sulcus could be helpful.

n. p.14, “...our through-skull imaging approach allows more flexible observations of primate cortical organizations with a much broader coverage. The effective size of a single field of view would be ultimately limited by the curvature of the skull.” A guideline of maximum allowed distance from the focal plane would be helpful for future users who aim to target different brain regions and ROI sizes.

o. p.26, “the refractive index of the skull and the brain were assumed equal. The refractive indexes of the skull and the brain are not equal, as shown in Methods Table 1 (1.56 and 1.37). It would be informative for readers whether and how these values impact the simulation results.

p. p.30, “air buffs” - This phrase appears to be miss-spelled.

q. p.38, “The same tonotopy mapping experiment (Fig. 3D) was performed through the cranial window, this time with 100 cycles instead of 20 cycles.” For a fairer comparison between the imaging conditions, please show the result using the same number of cycles of the data in Fig4E-H.

Reviewer #3 (Remarks to the Author):

Song et al developed a novel polarization-gated intrinsic signal optical imaging system to use as a mesoscopic cortical mapping system through the intact skull of marmosets. The paper was exciting and fully described the polarized-light imaging method as well as the authors' reasonable approach to testing multiple cortical regions and sensory systems in the same animal. It is an excellent paper and the only major request I have is that the authors should add a discussion of the pros and cons of using this method over the less-invasive method fMRI, which utilizes the same fundamental blood-based signal used here as an intrinsic optical signal.

Responses to Reviewers' Comments (MS# NCOMMS-21-27211)

We thank the three reviewers for their constructive and helpful comments. In the revised manuscript, we have carefully addressed all concerns raised by the reviewers. We have also provided additional data and analyses in light of the comments by the reviewers and included these results in the revised manuscript. The revised or newly added text is highlighted by the **RED font**. We hope our responses and revision have adequately addressed all concerns by the reviewers and would be happy to address any remaining questions. The following is a point-by-point list of all changes made in response to the comments by the three reviewers. All page and figure numbers referred to are those of the revised manuscript. The reviewers' comments are cited (marked by **BLUE font**), followed by our response.

List of new figures or panels:

Fig. 3I: Data from five more hemispheres have been added to this panel to further strengthen our conclusion.

Fig. 7B-F: We have added new neuroanatomical parcellation data (based on structural MRI) to relate our functional mapping results to anatomical structures.

Supplementary Fig. 3: The effect of simultaneous paradigm on modality parcellation.

Supplementary Fig. 9Q, R: More quantifications of the functional landscapes.

Supplementary Fig. 10: Quantification of Individual variation in the functional landscape.

List of changes in figure numbers:

The original **Supplementary Fig. 3-8** have been renumbered as **Supplementary Fig. 4-9**, respectively.

Reviewer #1 (Remarks to the Author):

In this study, Song et al. applied a polarization-gated wide-field optical imaging method to detect intrinsic signals over adjacent auditory, visual, and somatosensory cortices in awake marmosets. The authors used this method to clearly show that tonotopy, retinotopy, and somatotopy are distributed in each corresponding area. In addition, face patches were included in the motion-sensitive region. The methodology used for the measurements and analyses is described in detail. The basic features of the mapping were reproduced over five animals and the basic optical properties (lateral resolution and depth penetration) of the method were well estimated. Although the lateral resolution of 0.3–0.7 mm is lower than those of intrinsic imaging and two-photon imaging through a cranial window, this method can resolve the frequency gradient in the tonotopy, and the angle and eccentricity in the motion sensitivity. The color map of the functional landscape in Fig. 7 is very impressive. Thus, I agree that the present method is very useful for noninvasive mapping of the neural responses to multiple modalities over the cerebral cortex in the marmoset.

We thank the reviewer for the appreciation of our newly described through-skull imaging methodology and its applications in awake marmosets.

However, my major concern is that the authors do not make full use of the advantages of the wide area that can be simultaneously imaged. For example, if the face of a marmoset moves, is a stronger response evoked than the linear sum of the responses when the static image of the marmosets and moving dots are presented? If the face and vocalization of the marmoset are presented at the same time, do the signals in the auditory cortex and MT/V5 differ compared with when each is presented separately? To fully demonstrate the usefulness of this method, it is essential to show not only that the different functional modalities in different sessions can be parcellated in the same animal, but also that there are things that can be simultaneously seen for the first time there.

We thank the reviewer for the comments and agree that the experiments (motion + face, face + vocalizations) suggested by the reviewer address interesting scientific questions. However, a cautious answer to any of these important questions requires carefully designed new experiments with controls and validations that are beyond the scope of this manuscript. The scope of our current manuscript is to introduce a novel through-skull imaging method validated by topographic mapping in each of the three major sensory modalities. Researchers can make use of this new methodology to conduct further studies to address questions such as those suggested by the reviewer. It has also been pointed out by reviewer #2 and quoted here “... *It could be argued that not much has been shown in terms of what we know about the marmoset cortex, but this is not (in my view) the point of the paper – the key message is that something that has been thought to be impossible can be done, and evidence is provided that this can be done reliably...*”.

The reviewer also mentioned that “*To fully demonstrate the usefulness of this method, it is essential to show **not** only that the different functional modalities **in different sessions** can be parcellated in the same animal, but also that there are things that can be **simultaneously seen for the first time there**.”. Indeed, our experiment in Fig. 2 did exactly that (see also Supplementary Fig. 1 and 4). The experiment demonstrated different functional modalities (somatosensory, auditory, and visual cortices) can be **simultaneously seen** and parcellated within a **single** recording session (~7 min). We believe that the data shown in Fig. 2 is the first demonstration that these sensory modalities can be simultaneously mapped within a single experimental session in any NHP species.*

A related issue is whether these results parcellated simultaneously within a single session would be different if parcellated individually in different sessions. To clarify this matter, we further compared the maps acquired simultaneously with those acquired in each modality individually. These additional results are shown in the new Supplementary Fig. 3 (also see below). The individually mapped results show clear activations when the corresponding stimulation was presented and little activation if the stimulation was missing. Moreover, the individually mapped activation patterns are very similar to their simultaneously mapped counterparts, thus verifying the patterns we have measured in Fig. 2 were not affected by the simultaneous through-skull mapping paradigm. We believe this verification would provide an essential baseline for comparing simultaneously versus individually mapped patterns and thus facilitate further, more advanced experiments such as the ones the reviewers suggested.

Minor points:

The authors should quantitatively estimate the maximum size of the cortical area that can be imaged simultaneously. It is important to show the extent to which the through-skull imaging outperforms through-window imaging in respect to the area imaged because spatial resolution is higher in the latter imaging. For how long were the authors able to keep looking at the same area? The edge of the area where the skull was exposed should be clearly shown in the figures. I am not sure what the white and gray pixels in figures such as Fig. 3I indicate.

As we have demonstrated in Fig. 2, Supplementary Fig. 1, 4, and further verified in the newly added Supplementary Fig. 3, the extent of the cortical area that can be imaged simultaneously is at least $\sim 10\text{mm} \times 17\text{mm}$. This extent size, as reviewer #2 also pointed out, is $\sim 4\text{x}$ larger than the large cranial window we designed to cover the entire auditory cortex on the brain surface (Fig. 4). To describe this size explicitly, we have modified the last sentence of the corresponding paragraph (p8) as “Data from this **simultaneous parcellation** experiment thus showed the ability of XINTRINSIC to characterize such a multi-modal functional landscape quickly in each individual subject **over a large field of view (up to at least ~ 10 mm by 17 mm)**”. Moreover, we followed the reviewer’s suggestion to include an indicator for chamber margin edge in the figure.

An example is given below for showing how long we were able to keep looking at the same area. Two motion sensitivity sessions were recorded ***197 days apart***. The earlier session was imaged with a different objective and thus with a **smaller FOV**. The response patterns in these two sessions were very similar, suggesting the same area can be functionally imaged through-skull across several months. We have added the following sentence in the manuscript (p22): “**When the chamber condition is well maintained, the through-skull imaging can be performed for up to at least several months.**”

For the white and gray pixels in Fig. 3I, as we have stated in the methods section p46 “... For tonotopy visualization, pixels that had their tuning phases outside the time range of the tone pip sequence or their delay values outside the acceptable range were set to

0 in their saturation channel.” This resetting of saturation to 0 resulted in white and gray pixels.

I'm not convinced that there is a newly discovered pure-tone-responsive tonotopic gradient. In the through-window imaging in Fig. 4E, there are certainly blue and purple regions in the parabelt, but there is no clear gradient band from RT to the parabelt. Blue and purple regions are also observed in the opposite site near the lateral sulcus. What is the difference between them? In addition, Fig. 3I indicates that this gradient was ambiguous in many animals. The conclusion regarding this tonotopic gradient should therefore be weakened.

The standard model of tonotopic gradients in the primate auditory cortex was described in Kaas & Hackett 2000 (and copied below as panel A). As shown in the figure, tonotopic gradients are a defining feature for functional areas RT, R, and A1 in the auditory core. Similar tonotopic gradients were observed in our through-window map (Fig. 4E and copied below as panel C). These gradients are labeled as double-arrow lines below with their corresponding area labels RT, R, and A1 according to the convention in the standard model. As the reviewer has agreed, there are certainly blue and purple (high frequency) regions in the parabelt outside the auditory core. To quantify whether there are iso-frequency bands forming a gradient from low-frequency RT to this high-frequency region in parabelt, five lines (#1 to #5) were drawn in parallel between these regions. Tuning frequencies were quantified along each line (at ten equally spaced points covering each line) and shown in panel D. The tuning frequencies along each line remain relatively constant. The tuning frequencies across adjacent lines are distinctly separable. Together, these lines form a group of “iso-frequency bands” along a tonotopic gradient extending from RT to parabelt. We illustrate this gradient in supplement to the standard model in panel B below.

The reviewer also mentioned the other “blue and purple region in the opposite side near the lateral sulcus”. We believe the reviewer meant the more medial, high-frequency region between RT and R. This high-frequency region is also consistent with the region labeled with two “H” in the standard model (panel A). The tonotopic gradient between this region and the more rostral low-frequency region is a defining feature for the area RT in primates. In contrast, the gradient extending from the low-frequency RT to the high-frequency region in parabelt, to the best of our knowledge, has not been described by previous publications.

To further investigate how this gradient can be generalized in more subjects, we have expanded Fig. 3I with five more hemispheres from four additional subjects. The similar tonotopic gradient extending from RT to parabelt is also evident in each of these maps, as labeled by the white arrows (low-frequency) and black arrows (high-frequency).

Nevertheless, we agree with the reviewer that we should be more cautious about the conclusion regarding this tonotopic gradient. We have revised the last sentence of the corresponding paragraph in the main text (p9) as “...The observation of a pure-tone-responsive tonotopic gradient outside the classically described tonotopic maps suggests that the standard view on the tonotopic organizations of the primate auditory cortex (Fig. 3B) needs to be revised (Fig. 3C).”

Is it possible to quantify the variability of the response regions among animals? If it is large, this mapping could be used to accurately determine the target area before two-photon imaging or electrical recording. Please discuss this issue.

To quantify the individual variation in the functional landscape, we aligned functional landmarks across individual subjects and showed the results in the newly added Supplementary Fig. 10. Based on the mapping results, we determined the locations for the following landmarks as labeled in panel A: the lateral sulcus; the somatotopic orofacial transition; the retinotopic eccentricity transition; the retinotopic polar-angle reversal; the centroid of the face patch; and the low-frequency tonotopic reversal. A retinotopic reference point was further defined as the crossing point between the two marked retinotopic lines.

A Functional landmarks extracted from the landscape (M126D as the example)

B Functional landmarks aligned to the lateral sulcus and the retinotopic polar angle reversal

C Functional landmarks aligned to the lateral sulcus and the tonotopic low-frequency reversal

Three face patch tested subjects were co-registered together in panel B, by first aligning their lateral sulci together and then aligning their retinotopic reference points in a line perpendicular to the co-registered lateral sulcus. The subjects M126D and M15E had their centroids of the PD face patch separated by 2.3 mm. Moreover, the cores of the face patch in these subjects (defined by the inner-most iso-t-value contour) were largely non-overlap with each other. These results demonstrate the individual variation in the location of the PD face patch is at a scale that is comparable to the size of the face patch itself.

Five tonotopy tested subjects were co-registered and further shown in panel C, by first aligning their lateral sulci together and then aligning their low-frequency tonotopic reversals in a line perpendicular to the co-registered lateral sulcus. The distance of the low-frequency tonotopic reversal from the lateral sulcus varied between 1.6 mm to 3.2 mm among different subjects, suggesting a difficulty for localizing the functional reversal using the reference of the lateral sulcus. Furthermore, the retinotopic reference points also varied among subjects in a range of ~ 2 mm in the directions of both parallel and perpendicular to the lateral sulcus. These results further demonstrate the scale of individual variations in the functionally defined areas can be up to ~2 mm in marmosets, ~6% of their brain size.

We have followed the reviewer's suggestion and modified the corresponding sentence in the text (p16) as "... the approach developed in our study could provide direct guidance for subsequent experiments that require skull opening (e.g., electrophysiology recordings, two-photon imaging, or functional perturbations), given the individual variation on functionally identified regions can be up to ~2mm in marmosets, ~6% of their brain size (Supplementary Fig. 10)."

W118B was the oldest animal and the mapping in this animal appears to have the poorest resolution. Please discuss this matter and give possible reasons.

We think the reviewer meant the subject M117B. Among six subjects used in the landscape mapping experiments, M117B was both the longest implanted animal before imaging (26 months, versus 1-16 months) and the oldest subject (80 months, versus 44-54 months). Although the results in this subject demonstrate that through-skull imaging is possible in a subject that has been head-capped for more than 2 years and is relatively old (6, near 7 years old), the imaging quality in this subject is generally inferior to other tested subjects. This subject had been used in other experiments, including one that required drilling and thinning over part of the auditory cortex, before being adopted for the through-skull imaging experiments. Therefore, the compromise in imaging resolution might be attributed to several factors, including but not limited to, uneven pilot drilling and thinning over part of the auditory cortex that compromised the optical homogeneity of the imaged area; the tissue and vasculature proliferation over the top of or within the diploic layer of the skull; potential mechanical deterioration of the aged head-cap; possible age effect on the optical property of the skull. It is also worth noting that, another subject we have supplemented in Fig. 3I, M56E, was imaged at a comparable age (6 years). The results in this subject showed that the age effect alone cannot account for the poorer resolution in subject M117B.

The equations shown in Fig. 3D are essential to calculate the amplitude of f . It would be helpful for many readers if the authors explained these equations in the Methods.

We have followed the reviewer's suggestion and added the following text in the Methods section to explain the calculation (p44) "*Thus, by assuming the hemodynamic delays of a pixel in these two temporally "mirrored" sessions are identical, this delay can be numerically derived or canceled out by reversing the response phase of one session and combining it with the response phase of the other session (Fig. 3D-H). The equations in Fig. 3D show how the tuning phase and the hemodynamic delay for the exemplar pixel labeled in Fig. 3G were derived from the raw response phases shown in Fig. 3E.*"

Was there any motion artifact in the awake condition?

We typically experienced little to none motion artifact under the awake condition following our previously described protocols for marmoset chair adaptation and head-cap implantation (Lu et al *J Neurophysiol* 2001, Gao and Wang *Nat Protoc* 2020). These procedures for adaptation and head-cap implantation have been successfully applied to >100 marmosets in our laboratory in the past 20 years. And the stability of the preparation has allowed delicate experiments such as intracellular recordings to be

performed in awake marmosets (Gao et al *Neuron* 2016, Gao and Wang *Nat Protoc* 2020).

In rare cases, we observed the FOV slightly moved under the imaging camera. These moments are well correlated with the occasions when the subject made clear movements in the chair to adjust the body position (observed through a monitoring camera). When these movements were observed during recording, the session was subsequently canceled and restarted.

Another thing related to the motion artifact is the animal's hair waving into the FOV. Such artifacts occasionally happened near the ventral border of the head cap. An example can be found near the anteroventral corner in Fig. 3G. The white-colored artifacts are due to several hairs protruding into the FOV. We typically shaved the hairs near the head-cap margin to eliminate this type of artifact.

There are many typos (for example, eah in line 822 and emmitted in line 823). Please carefully check for them.

We thank the reviewer for pointing these typos out and have corrected them accordingly.

Reviewer #2 (Remarks to the Author):

The technique of optical imaging used for brain mapping is usually invasive, requiring the implantation of a cranial window. This creates challenges, for example, for long-term studies. Although earlier studies in mice have shown the feasibility of obtaining some signals through the skull, this has not been possible in primates. This paper by Song et al. provides a breakthrough in this respect, allowed by innovative application of a cross-linear polarization enhanced imaging. This results in the ability to image the marmoset cerebral cortex through the intact skull, at a relatively affordable cost. Compared to conventional optical imaging, this method allows a nearly four times larger field of view.

To validate this technique the authors confirmed several functional signatures of the mesoscale organization, previously derived from electrophysiology, such as tonotopy and face-sensitive patches. It could be argued that not much has been shown in terms of what we know about the marmoset cortex, but this is not (in my view) the point of the paper – the key message is that something that has been thought to be impossible can be done, and evidence is provided that this can be done reliably. I am convinced of the value of this technique as a scouting tool for placement of electrode arrays, tracer injections, or optogenetic vectors, and it may also be useful for studies of brain plasticity. Its general applicability will be significant for the field, and the paper is likely to become well cited. However, there are several technical points that should be rectified or clarified before I can recommend the paper for publication in Nature Communications.

1. Some clarifications needed: one weakness of the present paper is that some of the putative new findings (new face-motion patch, and new tonotopic gradient) are not related to current knowledge of the marmoset cortex neuroanatomy. Have the brains of these animals been sectioned for histology, or was a MR obtained? I am having trouble relating these findings to currently recognized areas. If a MR was obtained (or can be obtained) it would be important to register the location of the patches to one of the current computational templates (e.g. Majka et al. Neuroimage 2021). If histological sections were obtained, then correlation with the Paxinos atlas should be possible. For example, if the face patch was located in V4T, this would correlate nicely with the observation that V4T contains islands of cells that are motion-sensitive, and cells that are purely orientation-selective (Rosa and Elston 1998). If the patch is in TEO, this would be interesting in showing that this cytoarchitectural area may contain subdivisions. Likewise, could the new tonotopic gradient be related to the observation that RT, but not R or A1 have connections with the medial prefrontal cortex, which has been hypothesized to have a role in interpreting affective content of vocalizations (Reser et al. 2009)?

To relate our functional results to current knowledge of the marmoset cortex neuroanatomy, we have followed the reviewer's suggestion to further obtain a structural MRI scan on the post-mortem brain of the subject M126D. We registered our scanned data to two marmoset atlas templates for parcellating cortical areas (Majka et al. Neuroimage 2021, Liu et al. Neuroimage 2018). These newly obtained parcellation results are presented in Fig. 7B-F and discussed in an additional paragraph (p14).

“To relate the assembled functional landscape (Fig. 7A) to the current knowledge of marmoset neuroanatomy, the post-mortem brain of the subject (M126D) was further scanned by structural MRI. These structural data were registered to two neuroanatomy templates (Nencki-Monash template⁵⁴ and NIH-MBA v1.1 template⁵⁵). The resulting areal parcellations and their comparison were shown in Fig. 7B-D. These two structure-based parcellations are in general agreement but also differ from each other in many details (Fig. 7C). We further registered our functional landscape imaged through-skull to these structural parcellations (Fig. 7E, F). Many functional features in the landscape can be explained by the borders in these parcellations. For example, in Fig. 7E (Nencki-Monash template-based parcellation), the contour of auditory responsiveness in our functional landscape is recaptured by the overall border of the structurally parcellated auditory areas; the low-frequency reversal is coaligned with the border between A1 and

R; the polar-angle reversal of the visual motion-sensitive area is also coaligned with the border between MT/V5 and MST. In Fig. 7F (NIH-MBA v1.1 template-based parcellation), the anterior boundary of the visual motion-sensitive region is consistent with the overall anterior border of the MT/V5 complex; and the region with functional insensitivity to face scrambling (The saddle of the pink iso-contours) overlaps with the structurally determined MT/V5 (caudal) area. In addition, we are particularly interested in the locations of two functionally defined regions – the PD face patch, and the newly described tonotopic gradient. The PD face patch sits within the visual area FST in Fig. 7E, and area FST (rostral) in Fig. 7F. Moreover, the newly described tonotopic gradient has its low-frequency end coaligned with the lateral border of RT in Fig. 7E. The similar area was hypothesized to have a role in interpreting the affective content of vocalizations⁵⁶. This gradient also has its high-frequency end extending beyond the auditory belt regions into the auditory parabelt in both parcellations. Together, these structural parcellations further support our conclusions in the previous experiments that were drawn based on functional mapping results.”

To check whether the newly described tonotopic gradient can be related to the previous observation that RT, but nor R or A1 have connections with the mPFC (Reser et al, 2009), we further calculated probabilistic fiber tracking from auditory areas shown in Fig 7E and searched for their inferred connections in the PFC (multi-shell diffusion-weighted MRI scans, resolution: 0.15 mm isotropic; 3 shells: b= 1000, 3000, and 5000; 40 directions per shell. FSL toolbox for probabilistic fiber tracking). Seed regions for fiber tracking were defined based on the anatomical parcellation using the Nencki-Monash template as shown in Fig. 7E, including the core (A1, R, RT), the belt (RTL, AL, ML, CL, CM), and the parabelt (CPB, RPB). As shown in the figure below, anterior auditory regions (RT, RTL, AL, CPB, and RPB) exhibited similar connectivity patterns to the frontal areas, including connections to the mPFC. The same connectivity to the mPFC from the more posterior regions (e.g., R, A1, ML, and CL), was generally lacking. The new tonotopic gradient described in the current study was in the anterior regions (parabelt, AL, and probably RTL and RT as well) based on the parcellation in Fig. 7E. These results are consistent with the hypothesis that RT, but nor R or A1 have connections with the mPFC that may play a role in interpreting the affective content of vocalizations (Reser et al 2009).

2. Quantification of the functional landscape: The authors reported many interesting observations that could be novel, important findings on the functional landscape of the dorsal areas.

a. For one, it is stated on page 13: "the face patch (contrasting faces to other objects, indicated by cyan contours in Fig. 7, see also Fig. 6G) was located within the motion-sensitive retinotopic region anteroventral to MT/V5, with a preference for the lower-center of the visual field ... These claims hold for all tested animals." It is hard to judge if the above statement applies to all the animals investigated. In animal M15E (Supp Fig 8K), the face patch appears to be located posterior-ventral, rather than anterior-ventral to the yellow region (MT/V5 - MTc/V4t border?). In M117B, the face patch appears to sit on the part of MT/V5, preferring upper-center rather than lower-center of the visual field (Reviewer figure 1B).

We have changed the corresponding sentence (p13) into "Furthermore, the face patch (contrasting faces to other objects, indicated by cyan contours in Fig. 7A, see also Fig. 6G) was located within the motion-sensitive retinotopic region *ventral* to MT/V5, with a preference for the lower- center of the visual field (see also Fig. 5F, L, M)." to be consistent with what the reviewer pointed out in subject M15E.

To quantitatively determine whether the PD face patch prefers the lower- rather than the upper-center of the visual field in each subject, we calculated the polar-angle tuning of each pixel within the face patch (defined by the inner-most cyan circle in Supplementary Fig. 9F, K, P). These polar-angle tunings of face patch pixels were counted in the histograms shown in the newly added Supplementary Fig. 9R. In each of the three subjects, the polar-angle tuning of the face patch is significantly below the horizontal meridian and sits within the quarter of the contralateral lower visual field ($p < 10^{-12}$, Wilcoxon signed-rank test, single-sided, $n=861, 369, 349$, respectively), confirming our claim is true in each of the three tested subjects.

We believe this newly added panel would help quantify the polar-angle tunings in addition to our original landscape panels (Supplementary Fig. 9F, K, P), in which the center-preferring regions are shown in white color (through the saturation channel in HSV color space) and thus may reduce the visibility of the polar-angle tuning (through hue channel in HSV color space). Additionally, a polar-angle tuning-only map is demonstrated below for the subject M117B (left panel). Face patch contours are further overlaid on top of the polar-angle map in the right panel.

b. Another potentially novel observation is “close to this face patch, a part of the newly discovered tonotopic gradient (Fig. 3I) represents acoustic frequencies commonly found in marmoset vocalizations⁵³ (~3-11 kHz).” (page 13) Again, this is not clear in one of the three animals (M117B) as no tonotopy map is superimposed in Fig7P.

We have changed the sentence (p13) to “Moreover, close to this face patch, in subjects that were also tested for tonotopy, a part of the newly discovered tonotopic gradient (Fig. 3I) was evident to represent acoustic frequencies commonly found in marmoset vocalizations⁵³ (~3-11 kHz).”

Through-skull tonotopy map, as shown in Supplementary Fig. 5, was not obtained in subject M117B due to the technical reason described in Supplementary Table 1: “*The skull over part of the auditory cortex in this subject (M117B) was thinned in an earlier pilot experiment. Thus, tonotopy mapping through the intact unthinned skull could not be performed anymore in this subject*”.

c. It is also stated that “This region-within-landscape paradigm may offer more replicable comparisons across individuals and studies than traditional approaches.”. on page 14.

These exciting observations and claims need to be consolidated by objective quantification of the imaging data. For this purpose, the following analysis of the acquired data is recommendable:

I. Obtaining areal borders in an algorithmic manner in each animal. A standard technique to determine the border is the visual field sign map (Sereno et al., 1994; 1995).

II. Alignment of the functional data to a common reference across animals. The lateral sulcus could be a potential candidate for the reference landmark. On the aligned brain image, please show functional results of each animal, such as locations of face patches. This would be a less desirable option if the analyses suggested under point 1 cannot be performed.

The suggested analysis (I) requires continuous gradients in both polar-angle and eccentricity to be derived for each pixel first. Although polar-angle gradients could be calculated from our experiment, eccentricity tuning, in contrast, can only be estimated in a fashion of center vs. periphery preference from our experiment. We thus cannot directly apply the algorithmic method in Sereno et al., 1994 to our data and have to follow the suggested analysis (II).

To quantify the individual variation in the functional landscape, we aligned functional landmarks across individual subjects and showed the results in the newly added Supplementary Fig. 10. Based on the mapping results, we determined the locations for the following landmarks as labeled in panel A: the lateral sulcus; the retinotopic eccentricity transition; the retinotopic polar-angle reversal; the centroid of the face patch; the low-frequency tonotopic reversal, and the somatotopic orofacial transition. A retinotopic reference point was further defined as the crossing point between the two marked retinotopic lines.

A Functional landmarks extracted from the landscape (M126D as the example)

B Functional landmarks aligned to the lateral sulcus and the retinotopic polar angle reversal

C Functional landmarks aligned to the lateral sulcus and the tonotopic low-frequency reversal

Three face patch tested subjects were co-registered together in panel B, by first aligning their lateral sulci together and then aligning their retinotopic reference points in a line perpendicular to the co-registered lateral sulcus. The subjects M126D and M15E had their centroids of the PD face patch separated by 2.3 mm. Moreover, the cores of the face patch in these subjects (defined by the inner-most iso-t-value contour) were largely non-overlap with each other. These results demonstrate the individual variation in the location of the PD face patch is at a scale that is comparable to the size of the face patch itself.

Five tonotopy tested subjects were co-registered and further shown in panel C, by first aligning their lateral sulci together and then aligning their low-frequency tonotopic reversals in a line perpendicular to the co-registered lateral sulcus. The distance of the low-frequency tonotopic reversal from the lateral sulcus varied between 1.6 mm to 3.2 mm among different subjects, suggesting a difficulty for localizing the functional reversal using the reference of the lateral sulcus. Furthermore, the retinotopic reference points also varied among subjects in a range of ~ 2 mm in the directions of both parallel and perpendicular to the lateral sulcus. These results further demonstrate the scale of individual variations in the functionally defined areas can be up to ~2 mm in marmosets, ~6% of their brain size.

3. Source of artifactual signals: In Supplementary figure 3, at least in three (M15E, M8E and M7E) out six marmosets investigated, clear artifactual signals appear systematically in the anterior part of the ROI. In these regions, the temporal delay in response to somatosensory stimuli is far from the expected one (cyan) with some non-random spatial distribution.

A similar but less obvious glitch is also seen in the other two animals (M44D, M117B), and in Supplementary Figure 4. These artifactual signals can be problematic when aiming at delineating smaller functional structures such as subregions encoding lower or upper lips somatosensation within the primary somatosensory cortex. Thus, a bit more investigation would be desirable as to why and how this artifact happened in these animals but not in the remaining one (M126D). Maybe the difference lies in smoothness of the surface of the imaging window, or composition of applied transparent glue.

These regions are located anterior to the lateral sulcus and are consistent with where the somatosensory cortices are. We suspect the unconstrained oral motion of the subjects might drive somatosensation that produced activities in these areas. In particular, the modality parcellation experiment in Supplementary Fig. 4 (original Supplementary Fig. 3) involved air puffs pointing to the subject's orofacial areas, tested in parallel with loud white noise and moving dots. The subjects might move more during these sessions at moments even outside the stimulation delivery window and thus produce artifacts at other frequencies in the somatosensory regions. Subjects also vary in degree of movements during sessions with somatosensory stimuli. The subject M126D is among the calmest whereas other subjects were observed moving more (e.g., 8E).

These regions are also near to the coronal suture in the skull, where the frontal bone and parietal bone meet. The optical properties along this suture line could differ from other areas that are fully covered by a single piece of cranial bone. This factor might further contribute to the "non-random spatial distribution" of these "artifactual signals". It is possible that individual difference in coronal suture condition also contributes to the phenomenon here. Nevertheless, the chamber preparation of subject M126D was very similar to subject M44D for both the surface of the imaging window and the composition of the dental cement material. These preparational factors alone could not explain the phenomenon described here.

4. Apart from the above major issues, clearance of the following minor issues would enhance the value of the paper:

a. The source of the signal enhancement. The reported signal is generally much larger than previously reported ones. For example, the signal through the cranial window in Figure 4F can reach ~20% dR/R. This response amplitude is nearly 20 times larger than that was reported in Tani et al (ref 15), who exploited a comparable auditory stimulation and green light (535nm) for marmosets. It is not clear whether this signal enhancement is due to some physiological conditions or the optical settings developed in this study. If the latter is the case, which exact component contributes to the signal enhancement is not readily penetrable to readers. For instance, the Koehler illumination appears to contribute to homogeneous illumination across the large field of view, but not to enhancing S/N.

We use “per mille” (‰) rather than “percent” (%) for quantifying $\Delta R/R$ through our manuscript. The signal through the cranial window in Fig. 4F reached $\sim 20\text{‰}$ $\Delta R/R$ or $\sim 2\%$ $\Delta R/R$. This is consistent with the results in Tani et al. Fig. 3D, in which the intrinsic signal also reached $\sim 2\%$ $\Delta R/R$. It is also worth noting that our subjects were fully awake while subjects in Tani et al were anesthetized. This preparation difference, as well as the difference in acoustic stimuli, may also contribute to the slight difference in intrinsic signal amplitude. In terms of enhancing S/N, the optical settings may not contribute much to the through-window condition here.

b. Fig1B, For probing visual response, light from the screen must be isolated from the reflected light by the Koehler illumination. How was this achieved? An illustration or a picture depicting a light-isolation device would be helpful. Was this picture obtained through the cross-polarization or not?

There are three factors related to this question: the light-attenuating structure around the recording chamber; the very low light power from the screen relative to that from the Koehler illuminator; and the constant overall intensity on the screen attributed to our stimuli design.

Firstly, as we described in the Methods section (p22) “*During the implant surgery, the lateral part of the skull, presumably over the auditory cortex, the MT complex, and the lateral part of anterior parietal cortex (Fig. 1B) were exposed and covered with a thin layer (~ 1 mm) of dental cement. ... The rest of the exposed skull was covered with a thicker layer of dental cement and a wall was formed around the recording chamber to increase the mechanical stability of the head-cap and to protect the chamber.” This recording chamber wall of the head-cap naturally serves as a light-attenuating barrier for the light coming from the direction of the display screen.*

To be cautious about the light power from the screen, we further measured the light power by a flat slide power meter head (Thorlabs S170C) placed at the location of the subject’s recording chamber (thus without the recording chamber wall as a screen light barrier). The screen produced $\sim 4\mu\text{W}$ power at the location of the subject’s imaging chamber, whereas the Koehler illuminator produced 30mW light at the same location. Therefore, the screen, if modulated at 100% amplitude depth, would produce up to 0.13‰ change relative to the baseline power of the Koehler illuminator. This relative level, at its most, is very close to the theoretical shot noise floor of our imaging setup (0.08‰).

In practice, our visual stimuli were designed to keep the “amplitude modulation depth” as low as possible across any recording session. For the visual motion experiments, each moving dot getting out of the range was immediately replaced by another dot. For the face patch experiment, both foregrounds and backgrounds were balanced for total luminance, as shown in Fig. 6A and Supplementary Fig. 8, and as described in the Methods section (p38) “Visual objects were ... equalized for contrast and average luminance for all frames. ... The background of the frame was set as a randomly generated pink noise image that matched the average luminance of the objects (115 on 8-bit grayscale).” These visual stimuli designs ensured the overall intensity on-screen was maintained as constant as possible in each of the visual sessions.

To summarize, our recording chamber wall naturally attenuates the light coming from the direction of the screen. Even without any structure for light attenuation, the relative amplitude of screen change can only produce illumination change up to 0.13‰ relative to the Koehler illumination. This level, even at its maximal modulation depth, is very close to the theoretical noise floor of our imaging setup and is much lower than our imaged through-skull intrinsic signal amplitude (~1-8‰). Furthermore, our visual stimuli were designed to keep the luminance level as constant as possible across any visual recording session to minimize the modulation depth from the screen. The light received by the imaging camera from the screen, if any, would most likely add a small constant to the baseline. Therefore, we consider our results are not significantly affected by the light directly from the screen.

c. Fig1H, It would be informative for the readers if the panel includes traces for the conventional optical condition for widefield imaging, where polarization was not used for both illumination and light collection. The peak of the normalized intensity is at around 0.3-0.4mm visitation depth. Please add a brief discussion of how this value is impacted by optical parameters other than the wavelength. Please extend the x-axis to include the visitation depth at which the light intensity returns to 0. This information will provide critical information to understand what depth of the brain the recorded optical signal stems from.

This visitation depth estimation was based on the simulation data from Stockford et al 2002, in which the medium was assumed to be absorption-free. This is a relatively reasonable assumption for the skull, as in which the scattering is ~20x stronger than absorption (shown in Fig. 1F). The same assumption in the brain can significantly overestimate the visitation depth since the scattering in the brain is only ~2x stronger than the absorption.

For this reason, this simulation aims to demonstrate the effect of polarization control on relative visitation depth enhancement in the superficial, scattering-dominant layer (i.e., the skull) rather than a realistic estimation of visitation depth in the deeper absorption-heavy layer (i.e., the brain). In the latter case, absorption would introduce a quicker decay on visitation depth. To reduce this possible confusion, we have spelled the word “Simulation, assumed absorption-free” on the title of this panel and added a masking effect over the “cortex” part of the traces. We also modified the figure legend as “... Note: this simulation assumes no absorption in both the skull and the brain (see also methods). Thus, the traces over the “cortex” proportion are not intended to provide realistic numerical estimates but only for demonstration purposes.” Since the marmoset skull is thicker than the depth needed for the light to be fully depolarized (~0.4 mm for a green light in the skull), the further depth propagation in the brain proportion would be similar as in the conventional intrinsic imaging (within the top ~500um depth of the brain tissue, e.g., Fig. 8 in Tian et al 2011).

We have followed the reviewer’s suggestion and added a sentence (p29) on how the peak of the visitation depth in the skull would be impacted by optical parameters other than the wavelength. “... (ii) both layers were assumed absorption-free; ... The weak but factual absorption in the skull could also shift the peak of the visitation depth to a reduced depth for both channels”.

The visitation depth of conventional widefield imaging without polarization control would be between the linear-maintaining and cross-linear conditions shown in the panel and may have a bias towards the linear-maintaining condition. Unfortunately, we do not have the access to the original code of this visitation depth simulation to perform a directly comparable simulation for the suggested condition.

d. Fig11, it is recommendable to include the Monte-Carlo simulation results where no polarization was used for both illumination and light collection. This will highlight how the light-polarization improves the lateral resolution. Also, it would be informative if the panel includes traces when there is no skull above the brain. This will enable readers to better interpret the imaging results without the skull, presented in Figure 4.

The polarization-based strategy was designed for reducing surface reflections across the curved skull and for emphasizing photons that visit deeper structures. For the effect of polarization regarding on the lateral resolution, we did not expect any improvement and discussed this in the method session (p29-30) “... (for collected photons) We did not differentiate photons by their polarization, since the assumed thickness of the skull, 0.50 mm, is thicker than the travel distance needed for linearly polarized light to be completely depolarized [$1/\mu'_s(\lambda)$], estimated ~ 0.4 mm for the green light. For photons that eventually contribute to the sensitivity of the absorption change, they must travel through the skull to reach the gray matter and are thus already depolarized. Therefore, differentiating these photons by their polarization states would not result in much difference in estimating the lateral resolution here.”

We have followed the reviewer’s other suggestion to include traces when there is no skull above the brain. The newly added traces are shown in dotted lines.

e. e. Fig2. How was $dR/R=0$ defined? In page 39, “The baseline was loosely defined as the average intensity across the session, or the mean intensity over the pre-stimulus interval of a trial, or the maximal intensity over a trial. ”. Please describe which definition was used for Fig2.

We have modified the corresponding method section (p42) as the following “*The baseline was loosely defined as the average intensity across the session (Fourier analysis), or the mean intensity over the pre-stimulus interval of a trial (Fig. 2E, 3E, 6D, Supplementary Fig. 9B, 9G, 9L, 9Q), or the maximal intensity over a trial (Fig. 4C, 4F). Since the intensity change is relatively very small to the baseline ($\Delta R/R$ is typically at single-digit permille [‰] scale), these variants on baseline definition result in an unnoticeable difference in quantifying the relative change in $\Delta R/R$.*”

f. p.7, “These locations are consistent with previous results using invasive electrophysiology methods” and Fig2D-F. To demonstrate this in an objective manner, it is advisable to superimpose areal borders already published (e.g., <https://marmosetbrainmapping.org>) onto the images of the cortical surfaces used in this study. This will help readers to understand how well the imaging technique can delineate brain areas, which appears to be one of the main scopes of this study. Also it could lead to scientific discovery of individual differences between marmosets.

We believe this question is related to the reviewer’s first major concern. We have reconstructed areal borders based on structural MRI data in subject M126D using two published templates and showed the overlaid new results in Fig. 7E and F. We think these newly added panels would also answer the questions here.

g. Fig4G. The dotted abscissa seems to be a fitted line, but it may be misunderstood as an identity line. The panel would look more intuitively clear if the x and y axes were on the same scale as in panel H.

We have adjusted the coordinates on Fig. 4G to disambiguate the dotted line from a 1:1 identity line. The axes on Fig. 4G are for $\Delta R/R$ (‰) and the axes on Fig 4G are for tuning (Hz). We thus cannot use the same scale for panel G as in panel H directly.

h. Fig4B, E, H, The difference between the two window conditions is not only the spatial resolution but also the shift in preferred frequency. For example, the preferred frequency of the two peaks in panel B is clearly higher than that in panel E. The spatial filtering (panel I) does not explain the shift but instead exaggerates the size of the shift. A possible explanation of this shift would be helpful.

We have followed the reviewer’s other suggestion in the point “q” below and have updated panels in Fig. 4 (to use 20 cycles of data for both through- window/skull conditions). After this update, the frequency shifts as the reviewer described in the current point seemed to be reduced. The R-square for frequency tunings between the through-skull and the through-window conditions is 0.79. The spatial diffusion as shown in Fig. 4I would increase this R-square value to 0.90 at the diffusion scale of 0.60 mm (Fig 4J), suggesting the increased spatial diffusion (until 0.6 mm) further helps explain the shifts or discrepancies in frequency tuning between these two window conditions. As the R-squared reached a high level of 0.90 at the peak, only 10% variance was left between the conditions that cannot be explained by either inherent functional similarity in the brain or further spatial diffusion by the skull. It might be also worth mentioning that

the through-window data were acquired 233 days after the through-skull data acquisition, Thus, a slight variation in the tonotopic map over such a long period of time may happen and further contribute to the 10% unexplained variance here.

i. Fig4J, It is not apparent why the best diffusion scales are considerably different between dR/R and tuning. A discussion regarding this would be useful.

Similar to the last question, we have updated Fig. 4 following the reviewer’s suggestion in the point “q”. Interestingly, the difference in best diffusion scales has been reduced after this update. The updated Fig. 4J shows the best diffusion scale based on tuning is 0.60 mm, whereas the best diffusion scale based on $\Delta R/R$ is 0.70 mm. Both numbers are very close to our Monte-Carlo simulated resolution scale, 0.63 mm. We thank the reviewer for the suggestion and consider the earlier difference might be due to some effect of prolonged recording sessions (A 100 cycles session lasted more than half an hour).

j. Fig6, Was there a difference in behavioural response (eg. Pupil diameter) between the stimulus categories? Fig6D. What does dR/R mean in this panel? Considering 20s-long trial, signals at $t=0s$ and $t=20s$ must be identical, but they are not in this panel.

We did not notice much difference in behavioral responses among categories. In particular, the pupil diameters of the frames during which the visual stimuli were delivered were pooled in the figure to the right. The “face” category is only significantly different (smaller) from the “animals” category (multiple comparisons for one-way ANOVA), we would thus consider the pupil diameter alone could not explain the observed difference in neural responses.

The $\Delta R/R$ in Fig. 6D has the same definition as in Fig. 2E and 3E (the baseline intensity was defined as the mean intensity over the pre-stimulus interval of each trial). We have followed the reviewer’s earlier suggestion to include this in the text (p41).

For experiments that contain only a single trial within each cycle (e.g., Fig. 2 and 3), the cycle-averaged signal at the end of the trial must return to the original level at the beginning of the trial (assuming the baseline is relatively stable). This face-patch mapping experiment, however, contained eight different trials (categories) delivered in a pseudorandom order within each experiment cycle. It is only guaranteed that the signals at $t=0s$ and $t=20s$ would be identical when the signals were averaged across both cycles and categories. For signals that were only averaged across cycles but within the same individual category, the signal at the end of the trial is NOT theoretically guaranteed to fully return to the level at the beginning of the trial. The absolute level of the next trial’s

baseline can thus be slightly dependent on the residual signal from the previous trial. Nevertheless, since our total cycle (repetition) numbers ($n=218, 134, 234$ for each subject) are much larger than the total category number ($n=8$), we believe this possible dependence on the previously presented category, if any, would be efficiently averaged out.

k. Fig7 also Fig6E-G, There are two regions that respond preferentially to face over scrambled faces (purple contours). The posterior one is interesting in that it responds less to the five other object categories. Traces of this region, in the same format as Fig6D, would be helpful for readers to understand the response of this region.

We have provided these data for all tested subjects. These results have been integrated as Supplementary Fig. 9Q and are also shown below.

I. Supplementary Fig4B,C,G,H,5D,E,8F,K,P, the color bars are missing.

The color bars of Supplementary Fig. 5B, C, G, H (as 4B, C, G, H in the earlier version), were shown on the right side of the plots. The same for Supplementary Fig. 6D, E (as 5D, E in the earlier version) as well.

The color bars in Supplementary Fig. 9F, K, P (as 8F, K, P in the earlier version) are shown as integrated insets in these panels. The purpose of these panels is to show the relative positions of functionally mapped regions that have already been demonstrated in the

previous figures, we thus dedicate color bars in these panels for illustrating functional tunings.

m. Fig7 & Supplementary Fig8, To show how replicable the functional landscape across animals is, it would be informative to show positions of each functional areas (e.g., local peak motion-sensitive regions, face patch) on the brain across all the marmosets investigated. As a reference, the lateral sulcus could be helpful.

This is related to the earlier comment (2.c.). We have followed the suggestion and shown the newly added results regarding the individual variation in Supplementary Fig. 10.

n. p.14, “...our through-skull imaging approach allows more flexible observations of primate cortical organizations with a much broader coverage. The effective size of a single field of view would be ultimately limited by the curvature of the skull.” A guideline of maximum allowed distance from the focal plane would be helpful for future users who aim to target different brain regions and ROI sizes.

We followed the suggestion and have added the following paragraph (p25): “Our through-skull imaging targets a field of view (FOV) that is large (>20mm) and curved over the skull surface. This requires a thick depth of field of the setup to keep a consistent lateral resolution across the entire FOV. By utilizing an objective (Thorlabs TL1X-SAP) with a small numerical aperture (NA=0.03), our setup features a total depth of field estimated as ~0.98mm (following equation [10] in Oldenbourg and Shrivack 2009⁷⁵). With this relatively thick depth of field, we were able to capture images with sharp small features throughout the curved skull surface in the FOV. These features can serve as reference points for aligning sessions acquired across different days. Moreover, our estimated resolution on functional measurements is ~0.6-0.7mm (Fig. 4). To produce a lateral blurring more than half of this scale, a target needs to be placed at least more than 5 mm away from the focal plane. This depth tolerance on functional measures further facilitates functional mapping over a large FOV.”

o. p.26, “the refractive index of the skull and the brain were assumed equal. The refractive indexes of the skull and the brain are not equal, as shown in Methods Table 1 (1.56 and 1.37). It would be informative for readers whether and how these values impact the simulation results.

We have added the following sentences into the corresponding text (p29). “..., as the real refractive index of the brain is lower than that of the skull, total reflection may happen when photons in the brain try to re-enter the skull and be eventually received. This effect would produce a sharp decrease in visitation depth at the border between the skull and the brain.”

p. p.30, “air buffs” - This phrase appears to be miss-spelled.

We have corrected this miss-spelled word accordingly.

q. p.38, "The same tonotopy mapping experiment (Fig. 3D) was performed through the cranial window, this time with 100 cycles instead of 20 cycles." For a fairer comparison between the imaging conditions, please show the result using the same number of cycles of the data in Fig4E-H.

We followed the suggestion and have changed the through-window condition by using the first 20 cycles of the data and revised the results in Fig. 4C, E-J accordingly. The main conclusions still hold for the revised results. Interestingly, the two best diffusion scales estimated for through-skull resolution are closer to each other after this revision (0.60 mm for tuning-based estimation, 0.70 for amplitude-based estimation). We thank the reviewer for the suggestion.

Reviewer #3 (Remarks to the Author):

Song et al developed a novel polarization-gated intrinsic signal optical imaging system to use as a mesoscopic cortical mapping system through the intact skull of marmosets. The paper was exciting and fully described the polarized-light imaging method as well as the authors' reasonable approach to testing multiple cortical regions and sensory systems in the same animal. It is an excellent paper and the only major request I have is that the authors should add a discussion of the pros and cons of using this method over the less-invasive method fMRI, which utilizes the same fundamental blood-based signal used here as an intrinsic optical signal.

We thank the reviewer for his/her comments! We have followed the suggestion and added the following discussion for comparing between through-skull optical imaging and fMRI (p16):

“.... Compared to fMRI, another neuroimaging method that also utilizes the hemodynamic signals to measure brain functions, the through-skull optical imaging described here is at a much lower cost (<\$10,000) and with no requirement for day-to-day maintenance. It does not generate any acoustic sounds and thus allows quieter and more controlled auditory experiments to be performed. It can also be used in combinations of other fMRI non-compatible electronics or materials (e.g., cochlear implants). Nevertheless, the through-skull imaging still requires the removal of the scalp and thus can be more invasive than fMRI. It does not resolve any depth information or cover the entire brain in 3D, both of which are well enabled by fMRI. fMRI is also inherently more intuitive to be integrated with other MRI-based structural measurements (e.g., diffusion tensor imaging).”

REVIEWER COMMENTS

Reviewer #1 (Remarks to the Author):

I have read through the point-by-point responses and the revised manuscript. The authors did a good job responding to the comments. In particular, I appreciate Supplementary Figure 10. I have no additional major concern, but please describe in Methods those factors that might cause the image degradation in M117B.

Reviewer #2 (Remarks to the Author):

The paper is greatly improved, but there are still a number of technical points that deserve clarification, and/ or incorporation to the text as part of the discussion.

(p.13 in rebuttal referring to FigS9)

The newly added Fig9I appears to be contradicting what is shown in panels F, K and P. Panel I indicates that the face patch is in the contra-lower visual field, whereas panels F, K and P indicate they are either in the center-preferring region or outside of the motion-sensitive region. Perhaps the apparent inconsistency stems from the altered definition of a pixel's preference to the visual field. In panel I, the pixels' preference is judged solely on the polar angle map, whereas in panels F, K and P, the preference is judged both on the polar angle map and the response amplitude to the polar angle stimulation. To avoid confusion of the readers, it is advisable to use a consistent definition between the panels. Also, in revised FigS9F, K, P, colors in the color bars are not filled.

(p.14 in rebuttal) "eccentricity tuning ... can only be estimated in a fashion of center vs periphery preference from our experiment. We thus cannot directly apply the algorithmic method in Sereno et al., 1994"

From the explanation provided, it is not clear why the method in Sereno et al. cannot be applied. The eccentricity stimulus employed by the current study appears, from the description provided, to be ideally suited for this method. Perhaps the imaging result was not adequate for this method? For example, the spatial gradient for the eccentricity (Fig5F) might not be smooth, which is necessary for Sereno's method. If this was the case, please explain why the gradient was not continuous in space despite the eccentricity stimulus changing smoothly in time. In any case, the paper should at least acknowledge this as something to be attempted in future studies, and what conditions need to be improved to allow this (relative to the methods used in the present data collection).

(p.15 in rebuttal) “The subjects M126D and M15E had their centroids of the PD face patch separated by 2.3mm ...the cores of the face patch in these subjects were largely non-overlap with each other”

This raises a concern about the reliability of the location of the core of the face patch, including the possibility that they might reside in the different cortical areas depending on the subject. Moreover, the spatial distribution of the face patch depends on the subject, particularly when compared with other object categories (FigS9E, J, O): it is one quasi-gaussian patch in one animal (panel E), two patches in another animal (panel J) and yet one elongated region in the other animal (panel O). With the currently provided data, readers are left unclear whether this level of inter-subject variability is actual or incurring from the imaging technique employed. As the author speculated (rebuttal page 16), the success of XINTRINSIC could be affected by the optical properties of the skull surface. To disambiguate these possibilities, it is advisable that the location and the distribution of the face patch be confirmed with other measurement techniques, such as single-unit electrophysiology or functional MRI. It is also advisable how the functional signal would be impacted by optical properties of the skull, such as inhomogeneous thickness or different chemical compositions across the imaging field. The paper needs to better acknowledge these uncertainties, and discuss what could be done to disambiguate among these possibilities.

Related to the point above, the actual size of the face patch remains elusive as well. In Fig6, the face patch diameter is ~5mm (against scrambled faces) or ~2mm (against other objects). Considering the spatial resolution of functional measures of XINTRINSIC (Fig4), it seems likely that the actual size of the face patch is smaller than this. To better interpret the XINTRINSIC signal, a reader will find an estimate of the actual size of the face patch to be informative. This is important when the XINTRINSIC result is used to guide further studies at cellular resolution. One way to do this could be done with spatial deconvolution or filtering – an idea similar to the one used for Fig4J. A similar concern holds for the facial- and oral-coding regions in the somatosensory cortex (figs7). Should a reader interpret the imaging result showing the region encoding both the face and the mouth (a yellow region in Fig7F)? Or these two modalities are entirely segregated, as postulated in Fig7C?

(p.16, reply to our comment 3 in rebuttal)

Considering the spatial resolution of functional measures of XINTRINSIC to be 0.6-0.7mm (Fig4), structures smaller than this resolution are deemed an artefactual signal. Such artefactual signals are pervasive anterior to the lateral sulcus (figs4, figs7), and this needs to be resolved. Otherwise, interpretation of the imaging result would be confusing. The authors reasoned this could be due to unconstrained orofacial movements, which potentially drove somatosensation. However, this explanation does not itself explain why there are artefactual signals smaller than 0.6-0.7mm. We speculate the unconstrained orofacial motion could cause brain or skull motion, which in turn caused the artefactual signals below the resolution limit. Yet, this explanation would be somewhat surprising considering the fact that the same head fixation scheme is used for patch recording, which requires extreme stability. To further clarify the potential source of the artefactual signals, we suggest the following analysis: 1) Relate orofacial motion to imaging signals. If the orofacial motion causes the artefacts, then the artefacts are pronounced during the orofacial motion. One way to demonstrate this

is to align the imaging signal by the time of orofacial motion events. 2) Register each imaging frame to a reference frame. If the orofacial motion incurs brain or skull motion as we suspect, this procedure should reduce the magnitude of the artefactual signals smaller than 0.6-0.7mm. Please report the size of the deviation in each experiment to show whether the brain/skull motion is pronounced in the somatosensory stimulation experiments.

(p.16 in rebuttal) "The optical properties along this (coronal) suture line could differ from other areas ..."

Whether to see if this is the case, please superimpose the trace of the coronal suture in each animal of FigS4. This explanation appears to be more persuasive than the other, as the artefact appears not only in response to somatosensory stimulation but also in response to auditory stimulation (FigS4B), and sometimes even response to visual stimulation (M8E and M117B of FigS4C), suggesting that the optical property of the skull can be a critical determinant of for the success of INTRINSIC. This may impose a constraint on the location of the imaging window.

(p.21 in rebuttal) "The difference in best diffusion scales has been reduced after this update"

This is an interesting observation, but its interpretation needs attention. It is not clear whether the convergence is due to using the same number of trials or due to innate variability of the measured signal across trials. Please show how variable this estimate is. With 20 trials sampled from the total 100 trials, it should be possible to draw at least five lines.

(p.23 in rebuttal) "to produce a lateral blurring more than half of this scale, a target needs to be placed at least more than 5mm away from the focal plane"

Please explain how this number (5mm) was estimated.

(FigS6I) Colors in the color bar are not filled.

Responses to Reviewers' Comments (MS# NCOMMS-21-27211A)

We thank the two reviewers for their constructive and helpful comments. In the revised manuscript, we have carefully addressed all concerns raised by the reviewers. We have also provided additional data and analyses in light of the comments by the reviewers and included these results in the revised manuscript. The revised or newly added text is highlighted by the **RED font**. We hope our responses and revision have adequately addressed all concerns by the reviewers and would be happy to address any remaining questions. The following is a point-by-point list of all changes made in response to the comments by the three reviewers. All page and figure numbers referred to are those of the revised manuscript. The reviewers' comments are cited (marked by **BLUE font**), followed by our response.

List of new figures or panels:

Supplementary Fig. 11: Registrations of the PD face patch to the retinotopic and motion sensitivity maps

Reviewer #1 (Remarks to the Author):

I have read through the point-by-point responses and the revised manuscript. The authors did a good job responding to the comments. In particular, I appreciate Supplementary Figure 10. I have no additional major concern, but please describe in Methods those factors that might cause the image degradation in M117B.

We thank the reviewer's comments. We have added the description of the potential factors that might cause the image degradation in M117B in Methods (p23-24) "**Several factors may affect the long-term performance of the through-skull imaging, such as uneven surface that compromises the optical homogeneity of the imaged area; tissue and vasculature proliferation in the skull, especially near the coronal suture; potential mechanical deterioration of the head-cap over time; and possible age effect on the optical property of the skull.**".

Reviewer #2 (Remarks to the Author):

The paper is greatly improved, but there are still a number of technical points that deserve clarification, and/ or incorporation to the text as part of the discussion.

We thank the reviewer for the comments. We are glad to further clarify the technical concerns in the following point-to-point responses.

(p.13 in rebuttal referring to FigS9)

The newly added Fig9I appears to be contradicting what is shown in panels F, K and P. Panel I indicates that the face patch is in the contra-lower visual field, whereas panels F,

K and P indicate they are either in the center-preferring region or outside of the motion-sensitive region. Perhaps the apparent inconsistency stems from the altered definition of a pixel's preference to the visual field. In panel I, the pixels' preference is judged solely on the polar angle map, whereas in panels F, K and P, the preference is judged both on the polar angle map and the response amplitude to the polar angle stimulation. To avoid confusion of the readers, it is advisable to use a consistent definition between the panels. Also, in revised FigS9F, K, P, colors in the color bars are not filled.

We assume the reviewer meant Fig S9R instead of Fig S9I since Fig S9I is not newly added and is unrelated to panels F and P.

The panel S9R was added to address the reviewer's earlier concern on whether every subject had its PD face patch preferring the lower but not the upper visual field. It is thus dedicated to showing the statistics of polar angle tuning of face patch pixels. The original panels F, K, P in Supplementary Figure 9, in contrast, were to visualize each subject's functional landscape assembled from all measured functional contrasts, including but not limited to, the face patch, the retinotopic polar angle tuning, the retinotopic eccentricity preference, as well as the motion sensitivity. Although a pixel's retinotopic preference had the same definition throughout all these panels (F, K, P, and R), the visualization of all contrasts within the same image can be busy that the clarity of some contrasts might be reduced by the presence of others. For example, a pixel that prefers the contra-lower visual field should show a greenish color, since the polar angle tuning is encoded in the "HUE" channel of the HSV space. However, if the same pixel also prefers the center of the visual field, this greenish color will transition into a whitish color, as the eccentricity is encoded by the "SATURATION" channel (saturated color: visual periphery, unsaturated white: visual center). This color "masking" effect in the assembled map (F, K, and P) may explain the reviewer's impression of the "inconsistency" and "altered definition" between these panels and the panel R.

To avoid this confusion, we separated and extended panel R as Fig. S11. In this newly added supplementary figure, the face patch in each subject was visualized with a single other contrast at a time, as with the polar angle tuning in panel B, eccentricity preference in panel D, and motion sensitivity in panel F. The histogram of face patch pixels under each contrast was also shown in the corresponding panel to the right (panels C, E, and G) following the same visualization "definition" in the tuning map (panels B, D, and F). In each of the subjects, the PD face patch was located within the retinotopic region that prefers the contralateral-lower visual field (panels B and C). Moreover, the face patch overlapped with part of the retinotopic region that preferred the visual field center (panels D and E). Furthermore, the face patch also preferred moving dots over static dots in each of the subjects (panels F and G). Together, these panels in the newly added Fig. S11 further clarify the results we have shown in the original Fig. 9F, K, and P, with a single other functional contrast visualized at a time.

The reviewer also mentioned that the colors in the color bars in FigS9F, K, P were not filled. We assume the reviewer meant the color wheel legend was not filled in the center. This is an intentional design to show retinotopic polar angle tuning (encoded in the hue channel of the HSV space) together with the retinotopic eccentricity preference (encoded in the saturation channel, i.e., the “whiteness”). The same visualization encoding scheme was also used in Fig. S6l and S11H (copied above). We have added an additional legend to further clarify the design of this multi-dimensional HSV encoding.

(p.14 in rebuttal) “eccentricity tuning ... can only be estimated in a fashion of center vs periphery preference from our experiment. We thus cannot directly apply the algorithmic method in Sereno et al., 1994”

From the explanation provided, it is not clear why the method in Sereno et al. cannot be applied. The eccentricity stimulus employed by the current study appears, from the description provided, to be ideally suited for this method. Perhaps the imaging result was not adequate for this method? For example, the spatial gradient for the eccentricity (Fig5F) might not be smooth, which is necessary for Sereno's method. If this was the case, please explain why the gradient was not continuous in space despite the eccentricity stimulus changing smoothly in time. In any case, the paper should at least acknowledge this as something to be attempted in future studies, and what conditions need to be improved to allow this (relative to the methods used in the present data collection).

The reviewer pointed out that for Sereno's method to be applied, the spatial gradient for the eccentricity must be smooth. As the data showed (e.g., Fig 5F), this requirement was not met in our eccentricity experiment. This discontinuity in the imaging results was mainly due to our stimulus design rather than other factors. We explain the reasons in detail below.

To resolve a continuous gradient with the Fourier-based experimental paradigm (Kalastsky & Stryker *Neuron* 2003), the stimulus generally needs to be changed smoothly in **only one direction** within each session. This was the case in our polar angle measurements, as the polar angle center swept either clockwise (Fig. 5G, H) or counterclockwise (Fig. 5I, J) in every single session. This smooth, unidirectional sweeping would result in a phase-locking at the trial-repeating frequency in the signal if the pixel is responsive. The polar angle tuning can thus be estimated based on the signal phase at the trial-repeating frequency in a continuous fashion. Conversely, our eccentricity stimulus, although being smoothly changed in time, has bidirectional changes within each trial cycle. This design can only result in a bipolar preference estimation, rather than a smooth gradient. For example, any pixel that is eccentrically tuned to greater than 5°-radius, no matter it's 7° or 14°, would show activation at the trial-repeating frequency with a relatively fixed phase near the time zero of the trial (encoded as the reddish colors in Fig. 5F). The same for any pixel that is eccentrically tuned to less than 5°-radius, but with the phase relatively fixed near the middle of the trial (encoded as the cyan-like colors in Fig. 5F). In contrast, for a pixel that is eccentrically tuned to exactly 5°-radius with a near-zero bandwidth, it would be activated all the time during our eccentricity stimulus with minimal modulation at the trial-repeating frequency. The pixel would thus be dim with uncertain color in our eccentricity map. This is evident as a thin and dimmed gap between the reddish and cyan-like regions in Fig. 5F, presumed corresponding to the sensitivity to 5°-radius eccentricity.

We followed the reviewer's suggestion to discuss this limitation in the revised manuscript (p12): "Moreover, the retinotopic eccentricity was only mapped as center/periphery preference under our current stimuli. A future improvement in stimulus design can be attempted when absolute tuning values are required, as by some retinotopy analysis⁴⁵."

(p.15 in rebuttal) "The subjects M126D and M15E had their centroids of the PD face patch separated by 2.3mm ...the cores of the face patch in these subjects were largely non-overlap with each other"

This raises a concern about the reliability of the location of the core of the face patch, including the possibility that they might reside in the different cortical areas depending on the subject. Moreover, the spatial distribution of the face patch depends on the subject, particularly when compared with other object categories (FigS9E, J, O): it is one quasi-gaussian patch in one animal (panel E), two patches in another animal (panel J) and yet one elongated region in the other animal (panel O). With the currently provided data, readers are left unclear whether this level of inter-subject variability is actual or incurring from the imaging technique employed. As the author speculated (rebuttal page 16), the success of XINTRINSIC could be affected by the optical properties of the skull surface. To disambiguate these possibilities, it is advisable that the location and the distribution of the face patch be confirmed with other measurement techniques, such as single-unit electrophysiology or functional MRI. It is also advisable how the functional signal would be impacted by optical properties of the skull, such as inhomogeneous thickness or different chemical compositions across the imaging field. The paper needs to better acknowledge these uncertainties, and discuss what could be done to disambiguate among these possibilities.

The reviewer raised several technical concerns regarding the reliability of our imaging results, particularly in the case of the location of the PD face patch. These concerns include (i) the possibility that the PD face patch may reside in different cortical areas depending on the subject; (2) whether the inter-subject variability, especially the shape of the PD face patch, is actual or can also be affected by the imaging technique; and (3) how the optical properties of the skull, such as inhomogeneous thickness, can impact the functional signal. We'll try to address these concerns below.

Regarding in which cortical area the PD face patch resides, we have followed the reviewer's suggestion in the last round to provide structural MRI-based analysis to align the PD face patch according to the "architecture" category for parcellating cortical areas in subject M126D. The results (Fig. 7E, F) showed the PD face patch largely resides within the cortical area FST in the subject regardless of template choice. We added further discussions on individual variability in the text (p14-15): "...These "architectural" registrations, together with the "functional" and "topographic" evidence that this face patch is also motion-sensitive and prefers the lower-center of the visual field in all three tested subjects (Supplementary Fig. 9 and 11), support the idea that this face patch resides in the same area (FST) across subjects. Nevertheless, other possibilities, such as the PD patch may reside in cortical areas other than FST depending on individuals, cannot be excluded. A cortical area is typically distinguished from its neighbors by one or more neurobiological properties from four basic categories: "function", "architecture", "connectivity", and "topographic organization" (also referred to as the "FACT" categories⁸⁵). Evidence from more categories, especially cross-validations by multiple

techniques on an individual-subject basis, may help better disambiguate these possibilities in the future.” It is also worth noting that even in a more extensively studied model (e.g., macaques), to date, the same concern raised by the reviewer remains largely unresolved for any of the face patches. Our current study has provided one of the most detailed datasets relevant to determining whether a face patch resides in the same cortical area across subjects or not. Nevertheless, A systematic answer to this question may require more emphasis on “connectivity” and “anatomical” analyses based on individual subjects in future studies.

The shape of the PD face patch varied across our tested individuals, notably at the periphery of the patch. Similar shape variations of face patches have also been reported in other studies (e.g., Fig. S1 in Tsao et al *PNAS* 2008). Although the XINTRINSIC signal can be affected by the optical properties near the skull surface. This effect, by mechanism, mostly alters the details in results at scales near or smaller than the claimed resolution (0.6-0.7 mm). The size of the face patch, especially when counting at its periphery, is much larger than this scale. The individual variation in our results would thus unlikely be explained by this effect. Nevertheless, we agree with the reviewer that future cross-validations would help disambiguate the possibilities for explaining the individual variation. We added the acknowledgment to this in the text (p15) “...Evidence from more categories, especially cross-validations by multiple techniques on an individual-subject basis, may help better disambiguate these possibilities in the future.” We have also discussed a related limitation for neuroimaging on hemodynamic signals in the text (p16) “the intrinsic signal of the brain is inherently constrained by hemodynamic change, which may not be fully predicted by local neural signals⁶⁰.”

Regarding optical properties of the skull and their variations, we have added the following sentence in the text (p16) “The optical properties of the skull, which are thickness- and composition-dependent and may have variations, would also affect the imaging resolution.”

Related to the point above, the actual size of the face patch remains elusive as well. In Fig6, the face patch diameter is ~5mm (against scrambled faces) or ~2mm (against other objects). Considering the spatial resolution of functional measures of XINTRINSIC (Fig4), it seems likely that the actual size of the face patch is smaller than this. To better interpret the XINTRINSIC signal, a reader will find an estimate of the actual size of the face patch to be informative. This is important when the XINTRINSIC result is used to guide further studies at cellular resolution. One way to do this could be done with spatial deconvolution or filtering – an idea similar to the one used for Fig4J. A similar concern holds for the facial- and oral-coding regions in the somatosensory cortex (figs7). Should a reader interpret the imaging result showing the region encoding both the face and the mouth (a yellow region in Fig7F)? Or these two modalities are entirely segregated, as postulated in Fig7C?

The reviewer suggested the possibility of applying deconvolution to XINTRINSIC signals, in hope of seeing better the actual size of a functional region, e.g., a face patch, or the oral, or the facial somatosensory regions. We appreciate this suggestion but are also cautious about this procedure. We'll explain our rationale below.

The suggested deconvolution procedure may “enhance” the resolution in some optical imaging paradigms. However, the results would heavily depend on the assumed “deconvolution kernel” to maintain artifact-free. Unlike some other microscopy methods (e.g., confocal), in which the kernel can be well pre-determined and maintained relatively constant across the FOV, the “kernel” for XINTRINSIC may deviate from our numerical estimate (Fig. 11) and vary across the FOV, since the thickness of the skull can deviate from the nominal thickness (0.5 mm) and vary across subjects and sites (the reviewer had also suggested the similar possibilities in the last point). These variations may leave the low-spatial-frequency signals largely immune to deconvolution but can generate artifacts when deconvolution is applied with an ill-assumed kernel, particularly at high-spatial-frequencies. This kernel uncertainty would impose a limit in practice on whether a deconvoluted sharp transition should be interpreted as a true neural function or an artifact that is possibly due to the deconvolution procedures. Currently, we are not comfortable pushing the resolution limit of XINTRINSIC computationally beyond what we have claimed, ~0.6-0.7mm, by procedures like deconvolution. The search for finer functional details may require other more invasive methods than XINTRINSIC.

It is also possible that some functional regions are naturally “elusive” at their borders. Some neural presentations along the cortical surface may have sharp transitions, in which a precise border can be clearly defined; whereas others may have slower and noisier transitions between functions, in which an explicit border might not be obvious to define. In the latter case, a deconvolution procedure can hardly further help. Indeed, fMRI-guided high-density neurophysiology in macaques (Aparicio, Issa et al *J Neurosci* 2016) have demonstrated that the latter way of organization might be the case for face patches - The selectivity for faces was high near the center of the face patch. The selectivity gradually fell from the peak to a background level that depended on the measure and criterion for “face-selectivity” that was used, as well as whether single-units or multi-units were tested. Our XINTRINSIC results are consistent with this study that the selectivity for face slowly decreases from the patch center. Therefore, computational procedures like deconvolution may not help reveal further information about the size of the face patch. We also want to clarify that the face patch in our study, among many others, was strictly defined by the response contrast between faces and multiple other object categories. Scrambling in the visual form generally highlights the entire ventral stream, presumably most regions underlying visual-form processing (Tsao et al *Nat Neurosci* 2003) and the resulted regions are not necessary to be of the same sizes as the face patches contrasted by faces vs. other objects.

For the somatosensory orofacial mapping, we interpreted the results as in the figure legend (p81) “Every subject showed an orofacial somatotopic gradient with the oral component at the more anteroventral side, and the major facial component at the more posterodorsal side, with minimal overlap between them.” It is possible that the underlying representations are entirely segregated or have some overlaps in between. A careful answer to this question requires high-density recordings with much better resolution that are generally beyond the capacity of XINTRINSIC.

To sum up, the spatial features that the reviewer specifically looked for here may have to be answered ultimately by other recording techniques such as single-unit recording, two-photon imaging, etc. XINTRINSIC with deconvolution may not have the sufficient deterministic power to disambiguate possibilities in functional details for the reasons mentioned above. Nevertheless, mesoscopic functional guidance using hemodynamic

signals (e.g., fMRI) has been found widely helpful for determining the location of neurophysiological recordings at cellular resolution, despite its limited resolution and relatively elusive boundaries in many cases (e.g., Tsao et al *Science* 2006, Vanduffel et al *Neuron* 2014). The same may also apply to our XINTRINSIC imaging, which has an estimated resolution of ~0.6-0.7 mm.

(p.16, reply to our comment 3 in rebuttal)

Considering the spatial resolution of functional measures of XINTRINSIC to be 0.6-0.7mm (Fig4), structures smaller than this resolution are deemed an artefactual signal. Such artefactual signals are pervasive anterior to the lateral sulcus (figs4, figs7), and this needs to be resolved. Otherwise, interpretation of the imaging result would be confusing. The authors reasoned this could be due to unconstrained orofacial movements, which potentially drove somatosensation. However, this explanation does not itself explain why there are artefactual signals smaller than 0.6-0.7mm. We speculate the unconstrained orofacial motion could cause brain or skull motion, which in turn caused the artefactual signals below the resolution limit. Yet, this explanation would be somewhat surprising considering the fact that the same head fixation scheme is used for patch recording, which requires extreme stability. To further clarify the potential source of the artefactual signals, we suggest the following analysis: 1) Relate orofacial motion to imaging signals. If the orofacial motion causes the artefacts, then the artefacts are pronounced during the orofacial motion. One way to demonstrate this is to align the imaging signal by the time of orofacial motion events. 2) Register each imaging frame to a reference frame. If the orofacial motion incurs brain or skull motion as we suspect, this procedure should reduce the magnitude of the artefactual signals smaller than 0.6-0.7mm. Please report the size of the deviation in each experiment to show whether the brain/skull motion is pronounced in the somatosensory stimulation experiments.

(p.16 in rebuttal) "The optical properties along this (coronal) suture line could differ from other areas ..."

Whether to see if this is the case, please superimpose the trace of the coronal suture in each animal of FigS4. This explanation appears to be more persuasive than the other, as the artefact appears not only in response to somatosensory stimulation but also in response to auditory stimulation (FigS4B), and sometimes even response to visual stimulation (M8E and M117B of FigS4C), suggesting that the optical property of the skull can be a critical determinant of for the success of INTRINSIC. This may impose a constraint on the location of the imaging window.

We'd like to address the above two points together since they are related to the same concern regarding some minor features of the images, presumably, artifacts, that appeared anterior to the lateral sulcus in subjects M15E, M8E, and M7E (Fig. S4). In the rebuttal letter of the last revision, we suspected potential sources of these features as somatosensation driven by unconstrained orofacial movements, as well as the condition of the skull along the coronal suture.

To better disambiguate these possibilities, we performed further examinations, including terminal and postmortem procedures in some subjects, and checked on the skull condition, particularly along the coronal suture. We didn't observe any noticeable

variation in skull condition at most locations. However, in subjects that showed these anterior artifacts, signs of moderate osteoporosis and vascular proliferation were seen near the coronal suture, for reasons that we have not fully understood yet. We also followed the reviewer’s suggestion to superimpose the traces of the coronal suture in Fig. S4 (dashed white lines). These lines also correlated with the locations that showed the small features with non-cyan colors in the functional mapping results. After these examinations, we agree with the reviewer that the variation in coronal suture condition is a more plausible explanation for the small artifacts anterior to the lateral sulcus in some of the subjects in Fig. S4. This imposes a subject-dependent constraint on the location of the through-skull imaging in terms of imaging quality. We have added the following sentence in the text (p16-17): “Moreover, the condition of the coronal suture may vary across subjects and can affect the imaging quality over the local area (Supplementary Fig. 4).”

The reviewer also raised another concern on whether the unconstrained orofacial motion could cause brain or skull motion, which in turn caused the artefactual signals below the resolution limit. We followed one suggestion and registered each individual frame to a reference frame to correct the motion in subjects M15E, M8E, and M7E (subjects that were affected most by this type of artifact). After the motion correction (Pnevmatikakis & Giovannucci *J Neurosci Meth* 2017) was applied, the amounts of x- and y-displacements in individual frames were estimated at sub-pixel scale in each of the three subjects (size of recorded pixel: 46.9 μ m). This scale is much smaller than the features seen in the functional maps.

We further re-generated the functional maps on the motion-corrected data. We did not observe any reduction in the magnitude of the artifactual signals smaller than 0.6-0.7mm. As the reviewer commented: *“If the orofacial motion incurs brain or skull motion as we suspect, this procedure should reduce the magnitude of the artifactual signals smaller than 0.6-0.7mm.”* The data shown here are consistent with the idea that the unconstrained orofacial motion generated very little motion artifact, if any, and may not have the capacity to explain the local artifacts here.

(p.21 in rebuttal) "The difference in best diffusion scales has been reduced after this update"

This is an interesting observation, but its interpretation needs attention. It is not clear whether the convergence is due to using the same number of trials or due to innate variability of the measured signal across trials. Please show how variable this estimate is.

With 20 trials sampled from the total 100 trials, it should be possible to draw at least five lines.

We performed the suggested analysis and separated the 100-cycles through-window data into five 20-cycles each. The same analysis was then repeated five times and the results are shown in the panels below. The best diffusion scales based on response amplitude were estimated as 0.70, 0.62, 0.74, 0.58, 0.62 mm, respectively, with a mean value of 0.62 mm, whereas the best diffusion scales based on tuning were 0.60, 0.38, 0.18, 0.38, 0.22 mm, respectively, with a mean value of 0.35 mm. The difference between the two estimates was the smallest when based on the first 20-cycles of the through-window data.

There is certain variability among the five correlation curves within each measure, especially for the tuning-based measure. Yet, our current data cannot conclude on the reasons underlying this variability. The current five 20-cycles through-window data were artificially cut from the continuously recorded 100-cycle sessions that each lasted for more than 30 minutes. These long sessions were recorded with a repeating fixed trial that lacked randomness in the stimuli variety. The auditory system is known for its adaptation to repeating stimuli (Ulanovsky et al *J Neurosci* 2004). The behavioral state of the subject may also vary after the initial trial cycles due to the prolonged sessions. Therefore, it is hard to disambiguate among possibilities like the adaptation of the auditory system, innate variability in neural representations, change in the behavioral state such as arousal, or imbalanced experimental design between the through-skull and through-window conditions. Nevertheless, the first 20-cycles of the through-window experiment should be in good balance with the through-skull data that were also acquired as freshly started 20-cycles.

(p.23 in rebuttal) “to produce a lateral blurring more than half of this scale, a target needs to be placed at least more than 5mm away from the focal plane”

Please explain how this number (5mm) was estimated.

Our objective features a 0.03 numerical aperture (NA, $NA = n \cdot \sin \theta$, where n is the refractive index of the working medium, θ is the maximal half-angle of the cone of light that can enter the objective).

The NA is the determining factor for how fast the sample is defocused when placed at a depth deviated from the focal plane. The geometry of this effect can also be illustrated in panels (B) and (C) below. When the sample is placed at a distance of h away from the focal plane, for the same camera pixel that receives light from the focal point in (A) would now receive light over an area on the sample surface with a diameter of $D = 2h \cdot \tan \theta$,

As we claimed in the rebuttal letter of the last revision, “to produce a lateral blurring more than half of this scale [FWHM=0.6mm]”, which would be half of 0.6mm, thus 0.3 mm on D , the corresponding h is, $h = D/2 / \tan \theta = D/2 / \tan(\sin^{-1}(NA/n))$. In air, $n = 1$. For $D = 0.3mm$, the corresponding h is $h = 4.9977mm \approx 5mm$. That is how we claimed (p26): “..., a target needs to be placed at least more than 5mm away from the focal plane ($h \geq D \cdot n / (2 \cdot NA)$, where h is the focal plane depth from the nominal plane, D is the maximal blurring diameter, n is the refractive index, and NA is the numerical aperture)”.

(FigS6I) Colors in the color bar are not filled.

We are not quite sure what the reviewer meant by “colors in the color bar are not filled”. But since only the retinotopic color wheel has color among “color bars” in Fig S6I, we assume the reviewer meant the retinotopic color wheel is not “filled” but with a white center. This white color is intentionally designed to represent the visual field center by encoding eccentricity preference through the saturation channel of the HSV color space. As we have described in the figure legends under Fig. S6I (see also Methods), “(I) The summary map by combining the retinotopic maps and the motion sensitivity map. These maps are encoded into separate channels in the HSV color space. The retinotopic polar angle tuning (F) is encoded in the hue (color) channel, whereas the retinotopic eccentricity tuning (G) is encoded in the saturation (chroma) channel. A visual field color wheel incorporating these 2 features is shown at the lower left (for left hemispheres). A pixel sensitive to the visual field center (<5° radius) would be shown in white, whereas a pixel sensitive to the periphery (>5° radius) would be shown in a vivid color according to

its polar angle tuning. Additionally, the motion sensitivity (H) is encoded in the value (brightness) channel. Thus, the more motion sensitivity a pixel exhibited to moving dots, the brighter it appears in the map.”

To further clarify this color wheel, we have added a legend inset in the figure panel to show how this 2D retinotopic color wheel is encoded.

Fig S6I

https://en.wikipedia.org/wiki/HSL_and_HSV

2D
Retinotopy

=

Polar angle
tuning encoded
in the **"Hue"**
channel of the
HSV color space

×

Eccentricity
preference encoded
in the **"Saturation"**
channel of the
HSV color space

REVIEWERS' COMMENTS

Reviewer #2 (Remarks to the Author):

The authors have replied successfully to the last set of technical points I raised. I apologise for being a little bit picky, but hope that these comments will be found to have improved the paper.

Reviewer #4 (Remarks to the Author):

The paper is greatly improved, and there remains a few points that deserve consideration.

(p.9 in rebuttal) “We ...registered each individual frame to a reference frame to correct the motion in subjects ... The data shown here are consistent with the idea that the unconstrained orofacial motion generated very little motion artifact, if any, and may not have the capacity to explain the local artifacts here”

The analysis and its result may be worth a position in the manuscript, as they provide essential information that the animal’s motion is unlikely the source of the artefact.

(p.12 in rebuttal) Our previous remark on the colorbar. This problem seems to be a software-dependent issue. The same pdf viewed with Foxit PDF reader correctly shows a filled colorbar, whereas the same pdf viewed with adobe Acrobat Pro shows an empty colorbar as attached. The same phenomenon happens in SFig9.

i

F Cortical functional landscape imaged through-skull (subject M126D)

Responses to Reviewers' Comments (MS# NCOMMS-21-27211B)

We thank the reviewers for their constructive and helpful comments. In the revised manuscript, we have carefully addressed all concerns raised by the reviewers. We hope our responses and revision have adequately addressed all concerns by the reviewers. The following is a point-by-point list of all changes made in response to the comments by the reviewers. All page and figure numbers referred to are those of the revised manuscript. The reviewers' comments are cited (marked by **BLUE font**), followed by our response.

List of changes in figures or panels:

Supplementary Fig. 4: Motion corrected results were added as panels (**f-i**).

Supplementary Fig. 7: renumbered from the original Supplementary Fig. 8

Supplementary Fig. 8: renumbered from the original Supplementary Fig. 7

Supplementary Fig. 10: renumbered from the original Supplementary Fig. 11

Supplementary Fig. 11: renumbered from the original Supplementary Fig. 10

This figure renumbering was according to the editorial suggestion.

Reviewer #2 (Remarks to the Author):

The authors have replied successfully to the last set of technical points I raised. I apologise for being a little bit picky, but hope that these comments will be found to have improved the paper.

We thank the reviewer for the comments and wish our revised work could better introduce our through-skull imaging approach to the field.

Reviewer #4 (Remarks to the Author):

The paper is greatly improved, and there remains a few points that deserve consideration.

(p.9 in rebuttal) "We ...registered each individual frame to a reference frame to correct the motion in subjects ... The data shown here are consistent with the idea that the unconstrained orofacial motion generated very little motion artifact, if any, and may not have the capacity to explain the local artifacts here"

The analysis and its result may be worth a position in the manuscript, as they provide essential information that the animal's motion is unlikely the source of the artefact.

We have added this analysis and results in Supplementary Fig. 4 as panels (**f-i**) and related information in the Methods.

(p.12 in rebuttal) Our previous remark on the colorbar. This problem seems to be a software-dependent issue. The same pdf viewed with Foxit PDF reader correctly shows a filled colorbar, whereas the same pdf viewed with adobe Acrobat Pro shows an empty colorbar as attached. The same phenomenon happens in SFig9.

We thank the reviewer for noticing this compatibility issue. We have made sure these colorbars are compatible with Adobe Acrobat Reader in our resubmitted supplementary figure file.